# Individualized prescriptive inference in ischaemic stroke

**Dominic Giles** [1] ✉, **Chris Foulon** [1], **Guilherme Pombo**[1], **James K. Ruffle** [1], **Tianbo Xu**[1], **H. Rolf Jäger**[1], **Jorge Cardoso** [2], **Sebastien Ourselin** [2], **Geraint Rees** [1], **Ashwani Jha** [1] & **Parashkev Nachev** [1] ✉

The gold standard in the treatment of ischaemic stroke is set by evidence from randomized controlled trials, typically using simple estimands of presumptively homogeneous populations. Yet the manifest complexity of the brain's functional, connective, and vascular architectures introduces heterogeneities that violate the underlying statistical premises, potentially leading to substantial errors at both individual and population levels. The counterfactual nature of interventional inference renders quantifying the impact of this defect difficult. Here we conduct a comprehensive series of semi-synthetic, biologically plausible, virtual interventional trials across 100M+ distinct simulations. We generate empirically grounded virtual trial data from large-scale meta-analytic connective, functional, genetic expression, and receptor distribution data, with high-resolution maps of 4K+ acute ischaemic lesions. Within each trial, we estimate treatment effects using models varying in complexity, in the presence of increasingly confounded outcomes and noisy treatment responses. Individualized prescriptions inferred from simple models, fitted to unconfounded data, are less accurate than those from complex models, even when fitted to confounded data. Our results indicate that complex modelling with richly represented lesion data may substantively enhance individualized prescriptive inference in ischaemic stroke.

Marked individual variability in response to treatment is near-universal across medicine. However plausible the therapeutic mechanism, estimating the efficacy of an intervention—for a specific patient and across the population—requires inference from empirical observations of treatment outcomes. The established inferential approach, enshrined in the randomized controlled trial (RCT), traditionally seeks to estimate the average treatment effect (ATE) across populations described simply enough to permit modelling with standard statistical methods under randomized treatment allocation policy[1]. The critical underlying assumption—random inter-subject variability—is secure where the underlying biological mechanisms are homogeneous across the population, but violated where they are heterogeneous, undermining any inference that rests on it: for both the individual patient and the whole population, unless more complex analytic methods are employed[2]. For example, in the presence of two sub-populations of equal size and diametrically opposite responses—one negative, one positive—the average treatment effect will tend towards zero, regardless of the statistical power of the study, the quality of the data, and the magnitude of subpopulation-specific effects (Fig. 1). Crucially, heterogeneity need not be detectable within the prespecified trial evaluation itself, nor necessarily in any subsequent subgroup analysis based on the same trial specification: richer descriptions, more flexible models, and less reductive estimands are required[3–6]. The heterogeneity must be explicitly exposed by a new attribute—the differentiation between ischaemic and haemorrhagic stroke in thrombolysis offers a striking example[7]—or hitherto unmodelled relations between

---

[1]UCL Queen Square Institute of Neurology, University College London, London, UK. [2]School of Biomedical Engineering & Imaging Sciences, King's College London, London, UK. ✉e-mail: dominic.giles@ucl.ac.uk; p.nachev@ucl.ac.uk

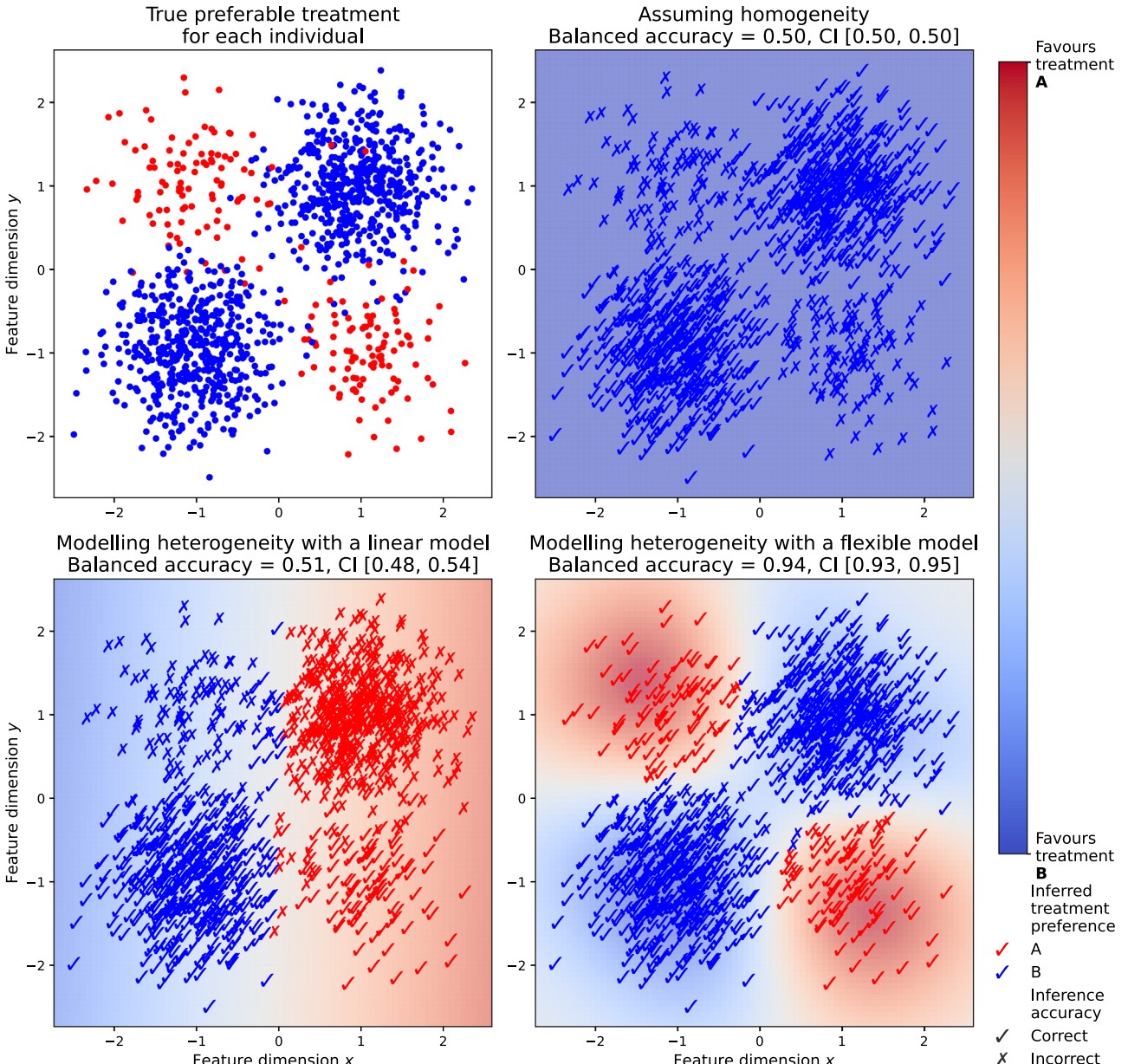

**Fig. 1 | Inferring treatment effects in the setting of heterogeneity.** Consider a hypothetical scenario where the members of a population described on continuous dimensions (*x* & *y*) systematically differ in their responsiveness to two treatments (A (red) & B (blue)). The two populations differ in size (A is smaller), and are non-linearly distinguished by their features (occupying diametrically opposed quadrants of the feature space). A statistical model of the optimal treatment that ignores the heterogeneity entirely, unconditionally averaging across all patients, inevitably favours B wholesale as the larger proportion, achieving chance level balanced accuracy (upper right). A model accounting for heterogeneity in simple, linear terms (a linear support vector machine) achieves only slightly better accuracy, for the distinguishing features are not linearly separable (lower left). By contrast, a model flexible enough to absorb complex, non-linearly defined heterogeneity (a support vector machine with a radial basis function kernel) achieves near perfect balanced accuracy (lower right). Note the differences in fidelity here can only be entrenched by higher quality or larger scale data, for they are a consequence not of the data or its sampling but of the mismatch between the complexity of the data and the flexibility of the statistical model.

attributes. Moreover, it may be so complex as to be impossible to capture within the simple, rigid statistical practice traditional RCTs conventionally employ.

This cardinal problem has three major consequences. First, an intervention may be mistakenly judged to be ineffective for all when it benefits an identifiable many, prematurely dismissing a valuable treatment, wasting the cost of its development, and discouraging further research into its mechanism and application[8]. Second, an intervention may be avoidably prescribed to those who do not respond or would be harmed by it, degrading clinical outcomes, exposing patients to unnecessary risk, and promoting systematic

inequalities in care. Third, since RCTs, with ATE as the outcome measure, have been placed on the highest pedestal, at the apex of the hierarchy of clinical evidence, neither argument nor data outside the established framework is permitted to gainsay them, indefinitely perpetuating both the errors and their consequences[9].

Arguably nowhere is this defect more important than in stroke, worldwide the second commonest cause of death and the commonest cause of adult neurological disability[10,11]. Though the aetiology is comparatively simple—vascular rupture, occlusion, or both—and the immediate treatment commensurately focused[12,13], the functional deficits subsequent management must address may be as complex as

the organization of the brain itself[14]. The observed patterns of disability, responses to treatment, and long-term recovery all suggest marked heterogeneity[15], explicitly violating the assumptions on which conventional interventional inference rests. Indeed, what we know of the functional anatomy of the brain[16], and the topology of ischaemic lesions, makes it inevitable that their interaction in stroke should exhibit a complex but nonetheless organized structure that may neither be dismissed as random noise, nor adequately represented by simple phenotypic markers[17–19]. A 'one-size-fits-all' assumption is not only biologically implausible here: it is directly contradicted by the facts[2].

While studies may account for simply described forms of heterogeneity, such as large vs small vessel occlusion, without a means of capturing complex heterogeneity historically, we have had no option but to ignore it[20] or model it in an oversimplified manner[21]. But the combination of machine learning with large-scale data now renders statistical models of sufficient expressivity and flexibility computationally tractable and deployable in real world practice[4–6]. Inferences about individuals may now be informed by the local subpopulation to which they belong—jointly defined by many clinical, physiological, and anatomical characteristics—estimating treatment effects much closer to the level of the individual than is possible with conventional approaches. Within such individualized prescriptive inference, the underlying statistical model seeks to determine not the ATE across the population, but the conditional average treatment effect (CATE) informed by a wide array of the patient's distinctive features, resulting in estimates that are more personalized, with reduced systematic variability across the population[22]. Just as the now familiar heterogeneity of ischaemic vs haemorrhagic stroke was resolved by brain imaging, so as yet unknown heterogeneities defined by multiple factors may be resolved by complex modelling of rich data[15,23–25].

How do we realize such an individualized approach? Where available, observations of responses to unconfounded treatment allocation—such as from a traditional RCT—may be combined with established low[26] or, new high-dimensional[19] multivariable modelling of conditional outcomes[4–6]. Randomization remains a powerful strategy, the best available means of protection from hidden confounding, and its vital historical role in advancing stroke practice will continue in the future. But in the setting of unknown heterogeneity defined by many interacting factors, the necessary data scale becomes infeasibly large, forcing reliance on routine clinical data streams with limited control over randomization. We must therefore determine the comparative importance of minimizing confounding due to non-random treatment allocation policies *vs* maximizing model expressivity and flexibility in inferring the prescription for each patient. If fidelity is now shown to be more sensitive to the former than the latter, then a RCT remains the optimal approach. Conversely, if model expressivity and flexibility are shown to be decisive, then achieving the data scales machine learning requires—and only routine clinical streams could plausibly provide—becomes paramount. In an ideal world, randomization would be combined with highly expressive models[6]: the question is where the compromise should lie when real-world constraints place the two in conflict, and the ideal is infeasible.

Examining the largest set of non-linearly registered acute stroke lesion data ever assembled—4119 lesions—here we perform a comprehensive comparison of traditional interventional inference—randomized allocation policy with simple covariate adjustment—vs a new approach—flexible counterfactual modelling of non-randomized data based on deep representation learning of richly defined anatomical patterns of damage. Our analysis tests the hypothesis that, under real-world constraints, where response to treatment is sensitive to the functional anatomy of the brain, the fidelity of inferred individual prescriptions depends less on unconfounded treatment allocation policy than on capturing treatment response heterogeneity with expressive generative models combined with flexible causal models.

Formally, the null hypothesis is that CATE-based inference with expressive representations in the presence of confounded allocation is not superior to inference with simple representations in the presence of random allocation, across the full spectrum of potential treatment effects, representational expressivity, model flexibility, allocation confoundedness, and functional domains. On the rejection or acceptance of this null hypothesis rests the optimal inferential policy for evaluating treatments in acute stroke sensitive to the functional organization of the brain, under real-world constraints on the scale of available data.

Since counterfactual outcomes of treatment are unobserved, this comparison requires a potential outcomes modelling framework[27,28]. Crucially, we cannot rely on empirical evidence of treatment efficacy from standard RCTs, for we are concerned with the setting of heterogeneity where their assumptions are definitionally violated. In the absence of a ground truth, constitutive of the inferential problem here, we are compelled to use a semi-synthetic counterfactual framework[29–33]. To promote real-world generalizability, our simulations are informed by large-scale, multi-modal empirical data: high-resolution lesion anatomy, and meta-analytic functional[34], gene transcription[35], and neurotransmitter receptor[36] maps. Moreover, we traverse a wide space of plausible real-world settings—systematically manipulating the richness of lesion parameterization, the magnitude and variability of treatment effects, the extent of treatment–outcome confounding, and the biological nature of the underlying deficit and treatment responsiveness—across the largest set of simulations ever performed with brain lesion data of any kind.

Following well-established principles of causal inference[37–39], our approach relies on a causal model, represented by a directed acyclic graph, of the spatial relationship between brain lesions and functional anatomy—a map of disrupted functions—and between treatment receipt and physiological anatomy—a map of treatment responsiveness—that explains observable patient outcomes (Fig. 2). This formulation allows us to quantify the impact on the efficacy of individualized prescription of both observable and unobservable treatment–outcome confounding, across the full space of plausible real-world scenarios, including a wide range of treatment and recovery effects. We can thus directly address the question of the relative contribution of randomization—the advantage of RCTs—and greater model expressivity and flexibility—the advantage of complex models of large-scale observational data, where real-world constraints force a compromise between the two approaches.

## Results

To test our hypothesis, we must first establish a semi-synthetic framework for empirically informed virtual interventional trials (Fig. 3). We cannot use a real trial, for the success or failure of a treatment has no ground truth against which the fidelity of the inference can be quantified. Since the disability caused by a stroke is directly related to the underlying disrupted functional anatomy, it is natural to posit lesion–deficit relations in terms of observed, potentially disrupted, neural patterns of functional organization, even if other factors will inevitably bear on the outcome. Equally, responsiveness to treatments acting downstream of the vascular disruption—e.g., tissue preservation, modulation, or regeneration—is plausibly sensitive to the biological properties of the damaged or threatened substrate. Our semi-synthetic simulation framework therefore combines lesion–deficit ground truth maps from functional anatomical data with treatment responsiveness ground truth maps from physiological, receptor distribution and genetic expression data. Both sets of simulated ground truths are derived from comprehensive, large-scale, meta-analytic data in the public domain. Finally, we need a large, maximally inclusive collection of acute stroke lesion maps that is plausibly representative of the heterogeneity of the population, registered into a common anatomical space to allow faithful comparison between individuals. A

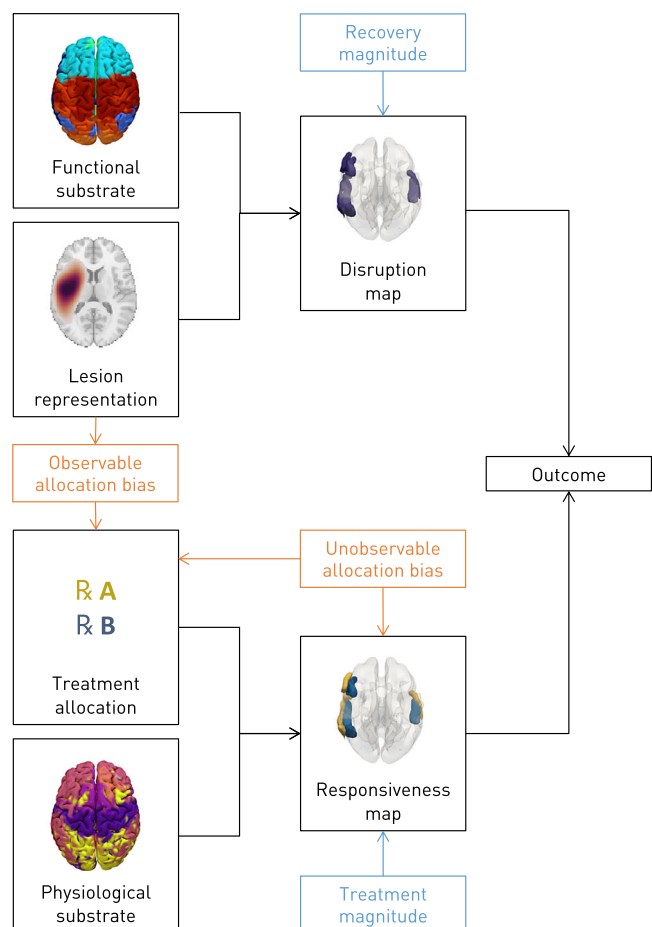

**Fig. 2 | Causal modelling of treatment—conditional outcomes in stroke.**
Directed acyclic graph of the modelled causal relations between brain lesions and functional anatomy—a map of disrupted functions—and between treatment receipt and physiological anatomy—a map of treatment responsiveness—in generating the observed outcome. The lesion anatomical representation (lesion or disconnectome) interacts with the functional substrate to generate a data-driven functional disruption map. In parallel, the treatment allocation policy interacts with the physiological substrate (microarray RNA transcription or neurotransmitter receptor data) to generate a data-driven treatment responsiveness map. These two spatial maps intersect to generate the patient outcome, modulated by the magnitudes of recovery and treatment effects. The relationships are modelled to be corrupted by varying degrees of observable or unobservable confounding due to non-random treatment allocation, enabling quantification of their impact on the fidelity of inferred optimal treatment.

small sample biased by selective recruitment would inevitably lead to under-estimates of observed heterogeneity and our capacity to model it.

## A data-driven map of functional networks in the human brain
To construct a set of ground truth lesion—deficit relations maximally grounded in real-world evidence, we derived a hierarchical grey matter functional parcellation of the human brain from a comprehensive analysis of NeuroQuery meta-analytic functional imaging and associated textual data[34]. Hierarchical voxel-based clustering of individual behavioural and cognitive terms based on the similarity of corresponding neural activation patterns yielded 16 distinct functional networks covering the full range of imaged behaviours (Fig. 4; Supplementary Figs. 1–2). Remarkably, though the clustering was driven solely by these activation patterns, strong commonalities in the keyword terms belonging to each cluster emerged—across the network archetypes of hearing, language, introspection, cognition, mood,

memory, aversion, co-ordination, interoception, sleep, reward, visual recognition, visual perception, spatial reasoning, motor, and somatosensory—corroborating the fidelity of the mapping approach (Supplementary Fig. 3). Each of these 16 archetypal functional networks provided a plausible ground truth for the critical neural substrates of the associated group of behaviours, replicating the likely spatial form and complexity of actual lesion—deficit relations, across the entire behavioural landscape.

## Genetic expression and receptor-determined treatment responsiveness
To create biologically plausible patterns of treatment response heterogeneity, individual variation in response was determined by further subdivision of the functional networks guided by physiological, whole-brain genetic expression (transcriptome) maps[35] or by neurotransmitter receptor distributions[36]. A deficit associated with a given functional network was thus designated as preferentially responsive to one of two treatments, dependent on which genetic expression or receptor-directed subdivision (Supplementary Figs. 11–74) is primarily disrupted by a given lesion. Crucially, treatment responsiveness, like the deficit itself, is thereby rendered sensitive to the anatomy of the lesion in its interaction with plausibly material biological features. Again, the objective is to replicate the likely form and complexity of anatomically organized determinants of responsiveness, not to identify them for any specific function.

## Lesion-deficit and treatment response simulation
A set of 4119 non-linearly registered diffusion-weighted magnetic resonance images (DWI) from 2830 unique unselected patients with confirmed acute ischaemic stroke was used to create a set of empirically informed, simulated lesion—deficit relations (Fig. 3, upper panel; Fig. 5). Though a proportion were drawn from the same patient, no images from within the same 30-day period from the same individual were retained in the dataset, as visualized in Supplementary Fig. 4.

Lesion binary masks automatically segmented from each DWI and transformed into standard MNI stereotactic space were intersected with each of the 16 functional grey-matter networks, yielding 4119 lesion—deficit relations across the full range of lesions and behaviours. A lesion intersecting with ≥5% of a subnetwork was designated as causing a corresponding deficit. To capture the broader impact of lesions on grey matter-defined functional areas through white matter damage, 'disconnectome' representations were also generated from each lesion mask[40,41], extending each lesion to remote grey matter areas disconnected by it, weighted in proportion to the extent of connectivity (Fig. 5). Disconnectome deficits were analogously designated, using a threshold of 0.5 to binarize the representation.

For each lesion and each functional network affected by it, the intersection with the responsiveness maps (see Fig. 2) determines whether the individual is included in the trial and, if they are, their responsiveness to the two treatments. The precise mechanism with which the intersection determines inclusion and response is described in the Methods section *Ground truth modelling*.

## Prescriptive inference with virtual interventional trials
Having generated a ground truth, we conducted a series of virtual interventional trials, where each patient event—defined by empirical lesion anatomy, the corresponding simulated deficit, and simulated treatment responsiveness—was allocated to receive one of two treatments, differing in their individual-level effects as outlined above (Fig. 3, lower panel). The task is to fit probabilistic models of individual outcomes to the trial data, from which we can then infer individualized prescriptions. Unlike a real trial, however, here we can quantify the fidelity of the inference against the ground truth, where the outcome for each individual, given each treatment, is known. This enables us to

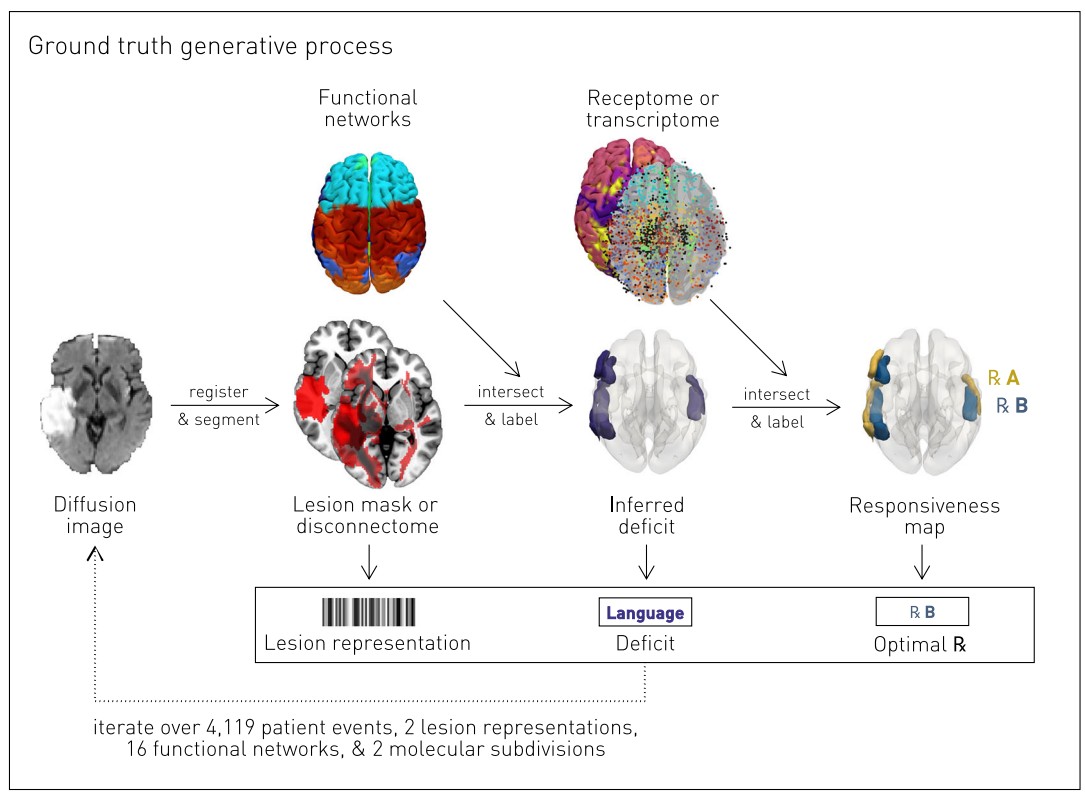

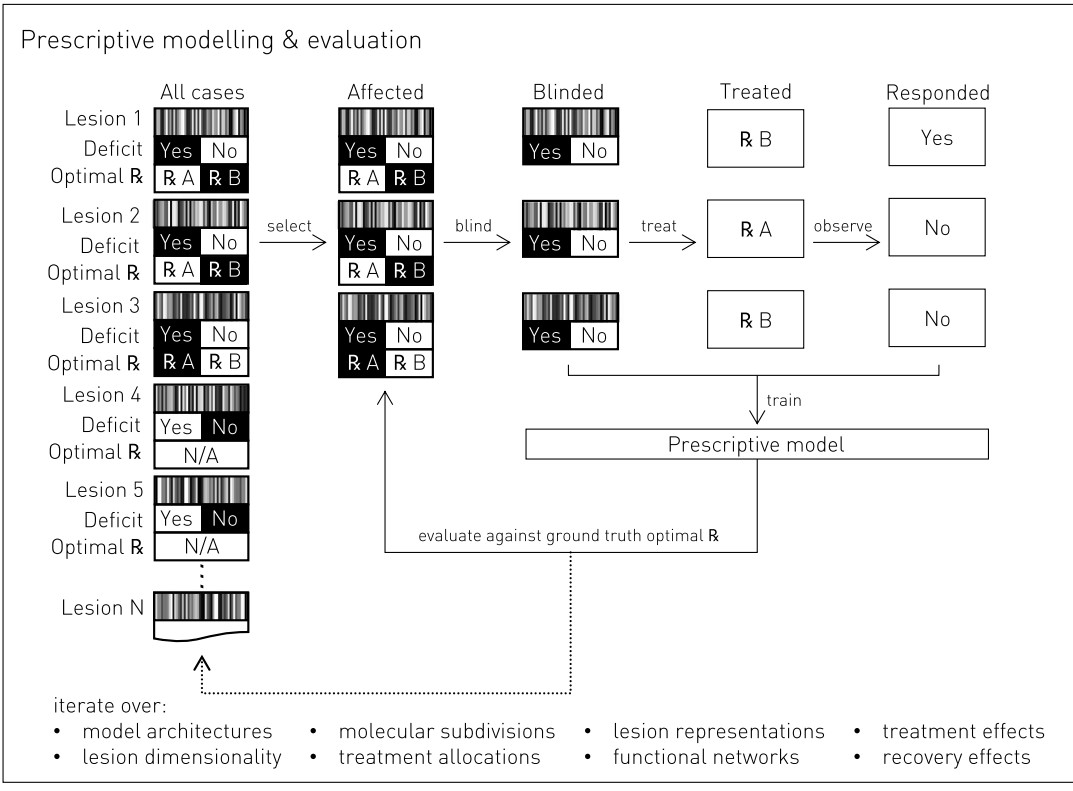

compare the performance of different models and experimental settings objectively[27].

To capture the full space of possible treatment and response scenarios, we varied both treatment and recovery effects across their entire possible range within independently conducted trials. Similarly, to generate treatment–outcome confounds, we vary the degree to which treatment allocation policy depends on lesion location, from not at all (random/unconfounded), to fully deterministic (non-random/confounded). In the process, we observe the effect on prescriptive inference as the assumptions underlying causal inference (see Methods section *Prescriptive inference*) are stretched to breaking point[42].

An alternative non-random allocation regime based on unobserved covariates was additionally implemented as a test of hidden

**Fig. 3 | Semi-synthetic ground truth synthesis (upper panel).** Schematic of the process of semi-synthetic ground truth generation from lesions or disconnectomes, given meta-analytic maps of brain functional organization and physiology. Diffusion weighted magnetic resonance images were obtained from patients with confirmed acute ischaemic stroke. Each image volume was non-linearly transformed into Montreal Neurological Institute (MNI) template space, facilitating voxel-based comparisons between different lesioned images and with meta-analytic functional and physiological maps. A binary ischaemic mask was automatically segmented from each image, and used to simulate real-world individual-level variability of functional deficits and treatment responses. The overlap between each lesion and each of 16 meta-analytic functional grey matter networks, transformed into MNI space, was used to generate plausible synthetic functional deficits (lesion−deficit simulation). The anatomical territory associated with each functional network was further subdivided by transcriptome or receptome distribution data into two subregions, variably responsive to different interventions, furnishing treatment effect heterogeneity. A plausible anatomically determined ground truth for treatment responsiveness is thereby established, enabling explicit quantification of the fidelity in inferring the optimal treatment. The resulting semi-synthetic data comprise stroke phenotype representations, associated functional deficits, and the designated optimal treatment for each patient simulation. **Virtual interventional trials (lower panel).** We used Monte Carlo methods to simulate a large number of virtual trials across the full set of modelled lesion−deficit relationships, treatment responsiveness, and treatment allocation policies. For each functional network, all patients with corresponding deficits were recruited to virtual trials in which they receive one of two treatments. At least one of these was determined to be truly effective. Non-random treatment allocation as may occur in an observational setting was simulated using a confounding hyperparameter, ranging from 0 (randomized) to 1 (treatment selection based strongly upon patient phenotype). A patient's response to a given treatment was dependent upon two sources of probabilistic noise: treatment effect, $\mathbb{P}(response \mid truly\ susceptible)$, and recovery effect, $\mathbb{P}(spontaneous\ response)$, independent of treatment received and responsiveness. These sources of noise are always present to some degree in any trial, randomized or observational. Within 10-fold cross-validation, prescriptive models were fitted using the training set, with quantified noise and confounding, for the objective of retrieving the optimal treatment, to each individual patient. The model was evaluated using a held-out test set, for lesion representations, disconnectome representations, functional networks, genetic expression-derived treatment responsiveness subnetworks, and receptor distribution-derived treatment responsiveness subnetworks, across the ranges of confounding, treatment effect and recovery effect.

confounding. The complete combination of these parameters−functional network, treatment responsiveness, lesion representation, treatment effect, recovery effect, and allocation policy−yielded 22,528 separate virtual clinical trial conditions, of which a 10-fold cross-validation scheme was implemented and results evaluated over 460 representation−prescriptive configuration combinations, leading to a total of 103,628,800 simulations.

Each trial used machine learning to model the distribution of conditional outcomes. We used the following model architectures: random forest[43], extremely randomized trees[44], XGBoost[45], logistic regression[46] and Gaussian process[47]. Here, the models were trained using cross-validation to predict outcomes from treatment allocation and representations of lesion anatomy. The treatment to prescribe is then inferred to be the one that has the greatest relative probability of causing a favourable outcome. The efficacy of this prescription is evaluated using balanced accuracy (Figs. 6 and 7); see also Methods section *Prescriptive inference* and Supplementary Tables 4–8 and Supplementary Figs. 75–84 for other evaluations.

To quantify the value of capturing the anatomical detail of lesions with expressive representational models, each stroke lesion representation (binary lesion mask or disconnectome) was embedded into a low-dimensional space, using a volume variational deep auto-encoder (VAE)[48], volume non-variational deep auto-encoder (AE), non-negative matrix factorization (NMF)[49] and principal component analysis (PCA)[50] to latent embeddings of varying length up to 50 components[17]. Independent models employing progressively higher dimensional representations were then compared against the baseline low-dimensional lesion representations typically employed in conventional stroke trials, vascular territory (anterior circulation ischaemic stroke or posterior circulation ischaemic stroke), or 6 specific arterial divisions, adapted from the atlases presented by Liu et al.[51]. Each trial was thus evaluated with 46 representations: two low-dimensional baselines and eleven incrementally higher dimensional ones across four representational rationales.

### Prescriptive inference is limited less by allocation policy than by model dimensionality

Under statistically ideal conditions−large treatment effects, infrequent spontaneous recovery, random allocation−prescriptions based on richly expressive representations were significantly more accurate than simple vascular baseline models overall (Fig. 6, upper left corners of each panel; Fig. 7, upper row of each panel).

With lesion masks as the input, balanced accuracy for inferring the optimal treatment from the 50-dimensional VAE representation, averaged across all 16 functional networks, was 0.793 (95% CI [0.75, 0.84]), vs 0.523 (95% CI [0.50, 0.55]) for a simple vascular baseline ($t = 10.5$, $p = 1.66 \times 10^{-18}$, Cohen's $d = 1.96$). A similarly striking difference was revealed by precision-in-estimation-of-heterogeneous-treatment-effects (PEHE, lower is better), yielding 0.466 (95% CI [0.44, 0.49]) for the 50-dimensional VAE representation vs 0.676 (95% CI [0.63, 0.72]) for the simple vascular baseline ($t = -8.12$, $p = 3.92 \times 10^{-13}$, Cohen's $d = -1.44$), indicating superiority of the richly expressive representation across both metrics.

With disconnectomes as the input, balanced accuracy from the 50-dimensional VAE representation was 0.875 (95% CI [0.83, 0.92]), vs 0.546 (95% CI [0.52, 0.57]) for the simple vascular baseline ($t = 13.5$, $p = 4.28 \times 10^{-24}$; Cohen's $d = 2.62$); by PEHE, 50-dimensional VAE yielded 0.349 (95% CI [0.33, 0.37]) vs 0.592 (95% CI [0.56, 0.63]) for the simple vascular baseline ($t = -10.2$, $p = 2.86 \times 10^{-17}$, Cohen's $d = -2.13$), similarly indicating broad superiority of the richly expressive representation.

With increasing noise−small treatment effects, frequent spontaneous recovery−the advantage of the richly expressive representations remained significant across most of the parameter space (white filled circles in Fig. 6, right column; Fig. 7, upper right panel), except for the noisiest settings where no statistical distinction could be made. Crucially, this advantage persisted even under strong corruption by treatment−outcome confounding from allocation bias (white filled circles in Fig. 6, right column; Fig. 7, lower right panel), generally yielding greater prescriptive fidelity even from a markedly confounded and noisy trial modelled with a richly expressive representation than from a fully randomized trial at the same noise level modelled with a simple one.

The observed patterns of comparative performance were broadly invariant when averaged across the criteria of treatment responsiveness−transcriptome or receptome (Fig. 6, upper two panel rows); lesion anatomical representation−lesion masks or disconnectomes (Fig. 6, central two panel rows); and observability of treatment allocation confounding−location-based or unobservable (Fig. 6, lower two panel rows). A high-dimensional representation, even in the setting of strongly confounded allocation, was almost always at least non-inferior to the RCT-like setup, of low-dimensional models and randomization, and in the vast majority substantially superior. These findings were consistently replicated across most functional deficit simulations (Fig. 7), based on the functional−anatomical networks visualized in Supplementary Figs. 11–74. An identical analysis based on PEHE demonstrated the same overall patterns (Supplementary Tables 4–8 and Supplementary Figs. 75–84).

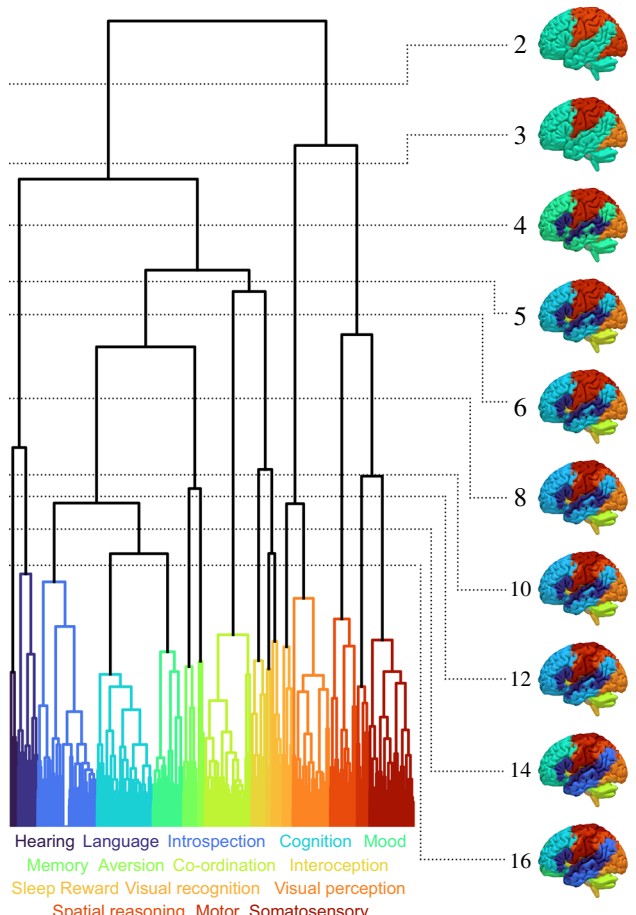

**Fig. 4 | Functional ground truths.** Functional grey matter parcellation dendrogram, showing agglomerative voxel-based clustering based upon association distances between voxels in their functional distributions. Visualizations of functional groupings at various thresholds up to 16 are shown on the right, displaying the grey matter functional hierarchy. From the 16-network parcellation, the archetypal terms associated with each group are shown below the dendrogram, with consistency of colours throughout.

These results provide grounds for rejecting the null hypothesis stated in the Introduction. We have shown that, across the range of modelled conditions, complex representations are always non-inferior and almost always superior to simple representations, even in the presence of non-random treatment allocation.

## Discussion

We address a cardinal question in the domain of ischaemic stroke: the optimal approach, under real-world constraints, to evaluating a treatment where the deficit and treatment responsiveness may exhibit sensitivity to anatomical patterns of functional activation, neurotransmitter receptors, genetic expression, or other neural properties of similar spatial structure. This is a question no-one has previously examined. Yet the unusually large number of interventions discarded at the trial stage[52], and the protracted history of those now persuasively shown to be beneficial[53], cast doubt on the felicity of the established approach founded on simple estimands, reductive descriptions of patients, and inflexible statistical models. Moreover, the picture of the brain emerging at every level—from the molecular to the physiological—indicates a finely granular, replicable organization likely to result in treatment heterogeneity no simple approach could adequately capture[8]. Given the sheer scale of stroke worldwide, many years of disability may be lost not because we do not have the good treatments but because we may have failed to

mitigate the impact of heterogeneity on the determinability of their effects.

Crucially, this is not a question that can be answered with conventional empirical studies alone. When—as almost always—only one of many treatments can be offered to a given patient, the question of which is optimal is counterfactual, to be settled by inference across a population of patients for which there can be no hard ground truth. A failure of inference here cannot be distinguished from a failure of the intervention, and if the inferential approach is never questioned, replication will not correct the error but typically entrench it. It is striking that whereas absence of evidence is commonly understood not to constitute evidence of absence, negative RCTs are widely interpreted as conclusive proof of a lack of meaningful therapeutic effect[54], for provided the trial has been correctly conducted, no other outcome is assumed to be possible[9].

Addressing this urgent question requires semi-synthetic simulations where the space of hypothetical treatment–outcome relationships is comprehensively surveyed with as much biological constraint as the problem permits. If we can show that one inferential approach is superior to another—across the broadest accessible space of hypothetical possibility—we need not have evaluated any specific relationship for our conclusions plausibly to generalize across the field. Here, we therefore combine the largest and most diverse collection of stroke lesion anatomical maps ever assembled with functional[34], genetic expression[35], and receptor distribution[36] maps drawn from the largest accessible data scales, and evaluate the treatment–outcome relationships across the full range of magnitudes of treatment and recovery effects. Our analysis spans 103,628,800 prescriptive simulations, involving >72,000 CPU-core hours of computation on a high-performance cluster. Evaluating any specific treatment–outcome relationship—however finely described and empirically corroborated—would not be sufficient, because generalization across arbitrary relationships would remain untested.

Our analysis shows that the expressivity of the representational models used to infer treatment effects has a substantial impact on their fidelity, across the surveyed space of hypothetical possibility, irrespective of the representation of the lesion, the functional domain of the deficit, or the source—receptor or genetic expression—of the determinants of treatment responsiveness. The use—widespread in the field—of simple, highly reductive lesion descriptors, such as crude vascular territory, is revealed to be suboptimal across the conditions that we have simulated. Now that advances in machine learning have made rich representation of stroke lesions realizable, choosing not to model lesion architecture with the granularity the underlying anatomical relationships demand is difficult to justify.

The superiority of inference with complex models extends to the setting of confounded treatment allocation such as observational studies inevitably risk. We show that richly expressive models of non-randomized data are typically superior to simple models of fully randomized data even where strong allocation confounding is present. Though randomization itself is always desirable, and provides the only theoretical guarantees of unconfoundedness, it is shown to matter *less* than representational expressivity across the modelled scenarios. Where real-world resourcing or other constraints force a choice between achieving the data scales complex models demand in an observational context vs conducting a RCT with simple methods, the latter may no longer be assumed to be preferable. Of course, where real-world feasibility permits it, complex modelling is ideally combined with randomization; equally, an effect drawn from observational data may be subsequently evaluated with randomized, prospectively acquired data from subpopulations rendered plausibly homogeneous by stratification with a sufficiently expressive model. Both approaches, including their combination, need to be considered in the light of real-world constraints, and their comparative fidelity tested in the same manner.

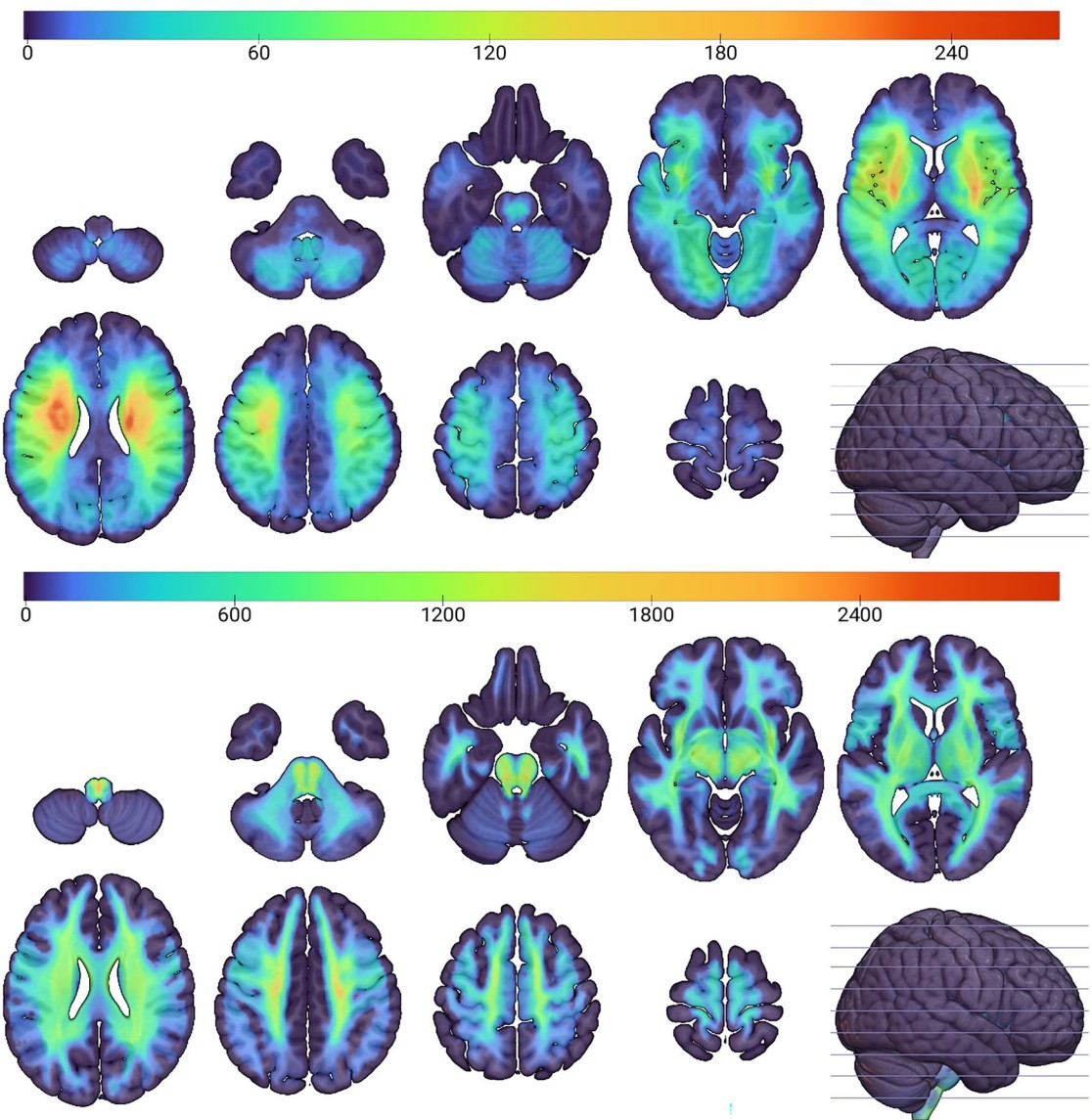

**Fig. 5 | Lesion (upper panel) and disconnectome (lower panel) anatomical distributions.** Axial slices showing the coverage in voxel-based summation of binary ischaemic maps of lesions (upper), and the summation of binarized disconnectome distributions ($p > 0.5$) associated with each of the lesions (lower), showing coverage across the brain. For both representations, $N = 4119$.

That there are bound to be aspects of treatment—revascularization, for example—where a therapeutic mechanism may be simple and general enough to be captured with conventional inferential methods does not imply that alternatives should be pursued only where the mechanism is plausibly complex. The critical outcome measure in stroke—as in medicine generally—is the impact on the patient's life, which is here expressed through the complex interaction between the anatomical patterns of damage and the functional organization of the underlying neural substrate. If one link in the chain of causation requires it, a more flexible inferential approach is desirable overall. Even revascularization requires reference to the structure of the vascular tree, whose intricate organization entails the possibility of complex heterogeneity. In any event, our mechanistic knowledge is rarely, if ever, sufficiently secure to place hard bounds on the requisite model flexibility.

Equally, though we may use a highly compressed representation—a two-dimensional embedding of lesion anatomy, for example—to stratify patients upstream of randomization within an otherwise conventional RCT, the loss of fidelity in the compression must be balanced against the gain in enabling randomization. Representation learning dependent on large-scale data would be required anyway, for the quality of the embedding is likely to be at least as dependent on data scale as a discriminative model of comparable flexibility. It is unlikely that overt—rather than latent—low-dimensional descriptors can commonly be found, for no (say) blood-borne signal could plausibly capture anatomically distributed information.

Though model compactness should be encouraged, the objective to be maximized here is generalizable prescriptive fidelity, not interpretability. The representational machinery we use permits the visualization of latent lesion archetypes (Supplementary Figs. 5–8), grounding prescriptive recommendations in an anatomically meaningful 'lexicon'. Our extremely randomized trees-based prescriptive models, though highly flexible, also offer feature importances, explicitly quantifying the influence that each lesion component exerts on the prescription. Note there is no clear ethical basis for preferring a low-fidelity, readily interpretable model over a higher fidelity, comparatively opaque one. In any event, the matter cannot be settled without soliciting the views of patients and the public at large, to whom this choice has never been presented.

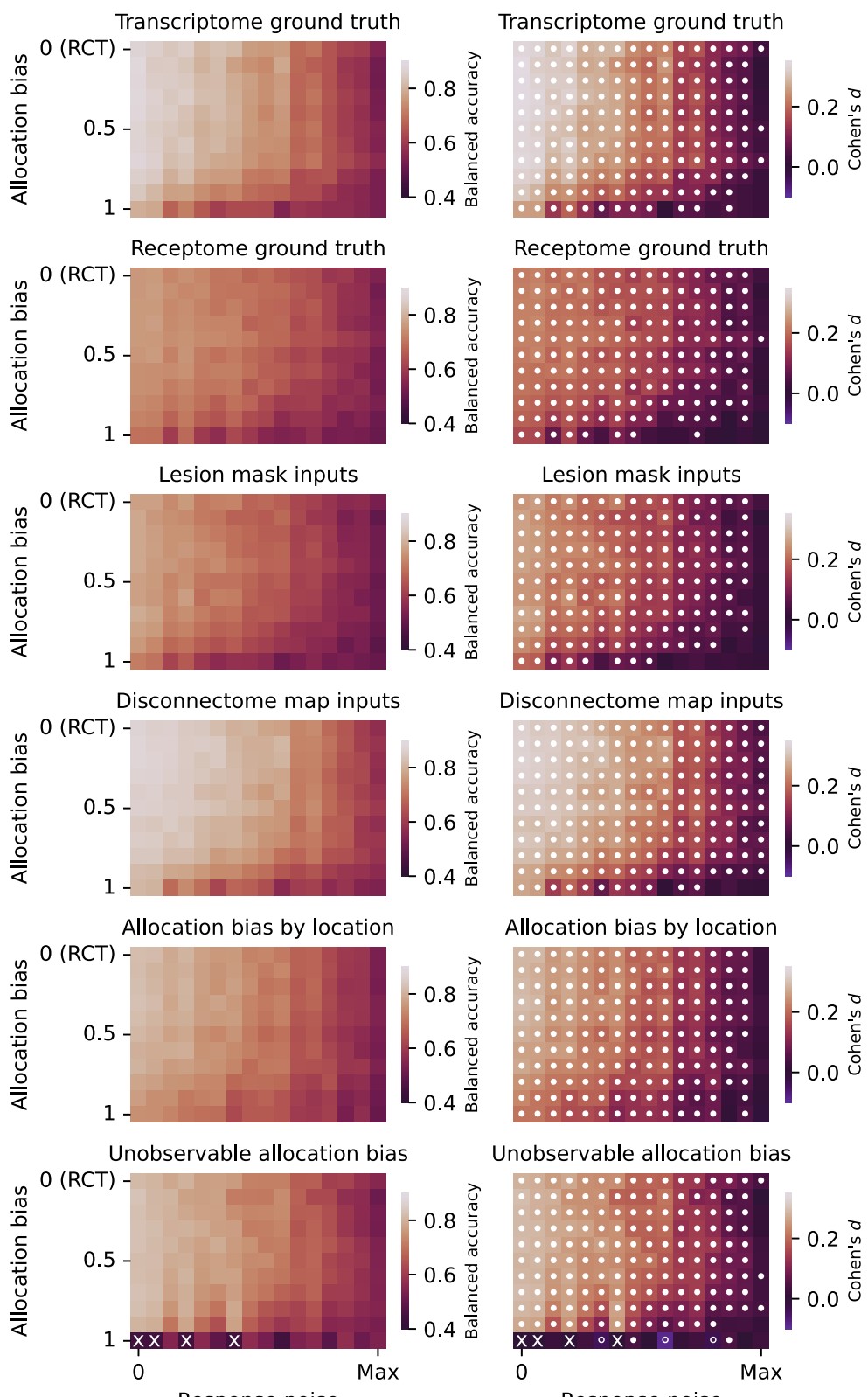

Unmodelled heterogeneity is not merely a statistical nuisance, limiting our power to infer the prescription associated with the greater probability of a favourable outcome at the individual level. It has consequences for medical equity, for where a failure of inference affects a distinct subpopulation, systematic epistemic inequity is introduced[55]. That the disadvantaged may here be identified by multiple interacting factors−creating diverse, intersectional groups−and that the defining

factors may lie outside recognized protected characteristics, does not make the resultant inequity any less significant. Medicine has a duty to maintain equity across all, regardless of the basis for their identity.

Medicine also has a duty to make the maximum use of the knowledge it draws from individual patients, often at some discomfort to them, and always at a cost. For treatments already in widespread use, individually prescriptive systems can be built on−and delivered

**Fig. 6 | Prescriptive performance with richly expressive lesion representations.** Balanced accuracy of treatment recommendation (left panel column, see Supplementary Tables 4–8 and Supplementary Figs. 75–82 for full descriptions), and statistical comparison against a simple vascular baseline, anterior or posterior vascular territories (right panel column), using a two-sided independent sample $t$-test. A filled circle indicates significantly higher performance for the expressive representation, and an unfilled circle significantly higher performance for the baseline model (using a Benjamini–Hochberg corrected critical value for 0.05 significance level). An 'x' indicates simulation conditions with insufficient class balance

to permit prescriptive model fitting (e.g., because there are no non-responders). The upper two panel rows show performance averaged across the criteria for treatment responsiveness (transcriptome or receptome); the central two panel rows across lesion input representation type (lesion masks or disconnectomes); and the lower two panel rows across allocation confounding observability (location-based or unobservable). The optimal expressive representation is shown to be non-inferior to simple vascular territories at informing prescription across almost the full landscape of observational conditions, and superior in the vast majority. See Supplementary Fig. 83 for respective PEHE plots.

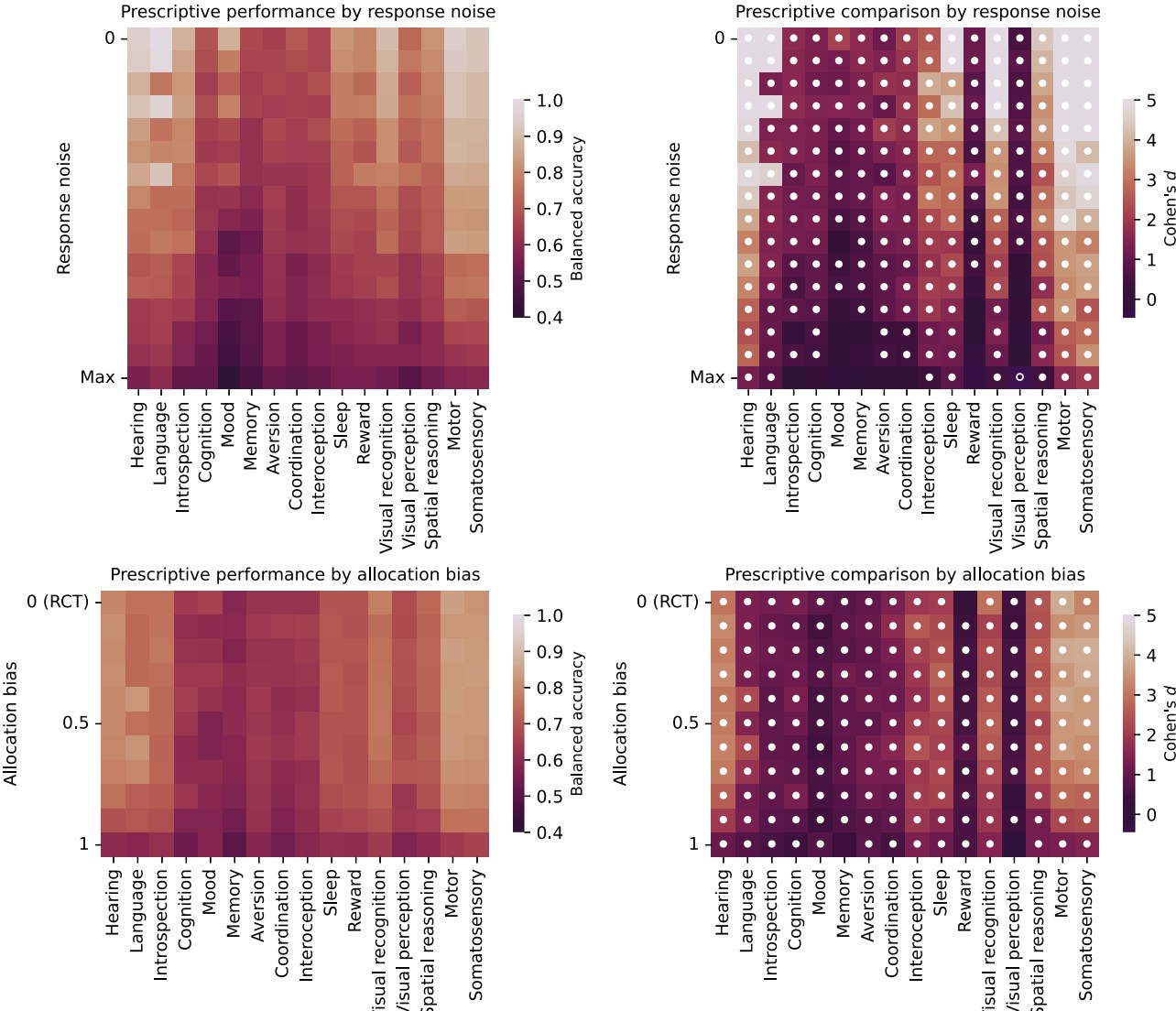

**Fig. 7 | Prescriptive performance with richly expressive lesion representations, stratified according to functional deficit category.** Balanced accuracy of treatment recommendation, stratified according to the modelled functional deficit (see Supplementary Tables 4–8 & Supplementary Figs. 75–82 for full descriptions), and averaged across all simulation conditions. The right column shows Cohen's $d$-effect size when comparing prescriptive performance using the optimal representation against a simple vascular baseline. A filled circle indicates superior performance for the optimal representation, according to a two-sided independent sample $t$-test,

beyond the Benjamini–Hochberg corrected critical value for 0.05 significance level; an unfilled circle indicates superior performance for the simple vascular baseline correspondingly. The full anatomical–physiological modelling framework is visualized in Supplementary Figs. 11–74 for receptome and transcriptome, respectively. The advantage in prescriptive performance for the richly expressive approach is shown to generalize across modelled functional deficits as well as data conditions defined by outcome response noise and treatment allocation confounding. See Supplementary Fig. 84 for respective PEHE plot.

through–established clinical streams. Not only may such systems render the process of individual prescription objective and scalable beyond specialist centres, they enable fine-tuning to local populations, including in response to dynamic changes over time. Note that the relevant features of ischaemic stroke–discrete spatial signals with high

contrast-to-noise ratio on routine diffusion-weighted imaging–are comparatively invariant to the imaging instrument, and easy to project into standard stereotactic space, rendering the modelling pipeline readily implementable. Indeed, our study exclusively uses clinical data collected in the course of routine practice, drawn from a wide diversity

of scanners and image sequence types. Nonetheless, though feasible, neither the universal use of highly expressive imaging nor its automated analysis are as yet routine, and requires motivation of the kind this study, amongst others, provides.

Though stroke is caused by anatomically organized damage to the brain, its outcomes are not exclusively determined by anatomy. A wide array of additional factors contribute to treatment outcome heterogeneity, placing varying demands on the expressivity of the models needed to capture them. The same approach can, and should, be extended beyond the anatomical domain. How readily any factor may be captured will, of course, vary, and there will inevitably be residual variation inaccessible to any practicable model. But the question of whether a factor is material to outcomes, and modellable within a given regime, requires investigation within the kind of framework presented here. Indeed, the central role of clinical outcomes in clinical evaluation and therapeutic decision-making demands careful evaluation of this aspect in future studies.

Our results must be interpreted in the light of an array of limitations. All model-based inference is inevitably constrained by the quality and availability of data, both for model development and clinical application. Though brain imaging is widely mandated in acute stroke, policies on the use of highly expressive modalities such as DWI vary, and the extraction of high-resolution lesion maps from CT remains challenging, if not infeasible[56–61]. But widespread recognition of the need for better characterization of the acutely threatened brain is driving the adoption of high-quality imaging, and the development of techniques, such as low field MR[62], with greater clinical accessibility. Our cohort was obtained from a routine clinical pathway with comprehensive MR imaging excluded only where contraindicated or not tolerated by the patient, yielding a plausibly representative population enrolled solely on the criterion of radiologically confirmed ischaemic stroke in the absence of major potentially confounding additional pathology or image artefact that precluded image analysis. As in any study, there will nonetheless be minority subpopulations with comparative undersampling for whom the generalizability of the findings cannot be assured. But where material heterogeneities have been identified in the majority population, the existence of under-characterized subpopulations does not remove the need to account for them but rather provides grounds for insisting on better coverage. And if such subpopulations are defined by complex characteristics, including lesion anatomy, their existence strengthens the case for more expressive models. Another source of heterogeneity, not modelled here, is variation in the timing of imaging in relation to what is inevitably an evolving pathological process. Though early DWI correlates well with subsequent persistent changes[63–70], expressive models permit the incorporation of temporal information in the lesion representation. Equally, the approach we outline is readily extensible to modelling the interactions between lesion and clinical characteristics within joint representations: since neither class of characteristic can plausibly be expected to render the other redundant, optimal joint models may be expected to be at least as expressive as each aspect demands.

The maximum achievable expressivity will always be bounded by the quality, size, breadth, and inclusivity of available data, and the nature and extent of treatment allocation confounding in any observational setting will tend to vary. It is not possible to model all forms of confounding in the observational setting, just as it is not possible to anticipate all forms of heterogeneity in the RCT setting. Individualized selection between equipoised approved treatments is one clinical context in which adequate data scale and comparatively low confounding may be expected; post-market surveillance is another. But if pivotal trials are rendered insensitive by unmodelled heterogeneity, consideration should be given to employing expressive representational models derived from observational data to encode trial data with the succinctness the applicable comparatively small-scale data

regime demands. Though higher individual-level fidelity naturally translates to higher population-level fidelity, an analysis of the same kind can be conducted in the context of population-based trials. Here highly compressed, latent representations of complex attributes can be used as baseline covariates or in treatment-by-covariate interaction terms in the trial statistical model, reconciling the demands of expressivity and compact description within the standard statistical framework or RCTs. The approach is akin to the use of composite biomarkers that distil high-dimensional variation into a low-dimensional phenotype, and its potential value can be quantified by analogous analyses. Note that since deriving expressive latent representations of any complex signal involves device-assisted models, direct clinical use for directing specific treatments would require a regulatorily approved device.

Finally, in common with all inference, our framework depends on foundational assumptions about the nature of the underlying causal relationships and their accessibility to imperfect observation. Inference here, as elsewhere, must always be qualified, and no model may be assumed to be the best, only more or less credibly better than another. Our approach leaves a great deal of room for further optimization it is the task of future studies to explore. But if the complexity of the brain, in both health and disease, is now undeniable, so ought to be the direction in which prescriptive models must evolve to deliver personalized, equitable care.

## Methods

This study was performed under ethical approval by the West London & GTAC research ethics committee for consentless use of fully anonymized data.

Indexing individuals by $i = 1, \ldots, n$, each participant in our virtual trials is represented by a tuple $(\mathbf{x}_i, w_i, y_i)$ defined as follows.

1. The phenotype characterization is denoted by $\mathbf{x}_i \in \mathbb{R}^d$, for the range of dimensionalities, $d$, from 2 to 50.
2. The allocated binary treatment is denoted by $w_i \in \{A, B\}$.
3. The observed binary outcome is denoted by $y_i \in \{0, 1\}$. Here, 1 is the favourable outcome and 0 is the unfavourable outcome.

This constitutes a biologically plausible observational dataset, $\{(\mathbf{x}_i, w_i, y_i)\}_{i=1}^{n}$, with manipulable values of allocation confoundedness, and observational noise in the form of fixed treatment and recovery effects. Each $(\mathbf{x}_i, w_i, y_i)$ is assumed to be sampled i.i.d. from some distribution $\mathbb{P}(X, W, Y)$. Here, and henceforth, we model each of these random variables respectively as belonging to the same probability distribution, but with different parameters (for this reason, there are no subscripts on $X$, $W$ or $Y$).

We follow the Neyman–Rubin causal modelling framework[37,71]. Each individual has two potential outcomes, $Y_i^{(A)}$ and $Y_i^{(B)}$, caused by receiving treatment A or B, respectively. Given wholly empirical data, only one potential outcome is observed and therefore the individual treatment effect, $\tau_i := Y_i^{(A)} - Y_i^{(B)}$, cannot be evaluated. The optimal treatment, $w_i^*$, cannot be evaluated either. The advantage of our semi-synthetic virtual trials is that each of these quantities is known, enabling evaluation of the prescriptive inference. In the following sections, we describe the data-generating process for the virtual trial data.

### Patient population

Patients attending the University College London Hospitals NHS Foundation Trust hyperacute stroke service undergo diffusion weighted imaging (DWI) within 24 h of presentation where feasible as part of the clinical routine. We used irrevocably anonymized magnetic resonance imaging data from unselected patients with an admission diagnosis—both clinical and radiological—of acute ischaemic stroke. This dataset comprises 4,119 irrevocably anonymized DWI from 2830 unique patients. Where multiple images of a patient were available, only those ≥ 30 days apart where used, ensuring—given the duration of

DWI positivity[72]—that each image represented a new lesion. Supplementary Fig. 4 shows a CONSORT-2010 flow chart of the recruitment process. The age (mean 67 years) and sex (56% male; 44% female) distributions are shown in Supplementary Fig. 10. Beyond basic demographics, no other characteristics, or observed outcomes, were collected or modelled, for we are concerned with modelling counterfactual, unobserved outcomes given virtual, yet to be identified, interventions. A proportion of the data reported here has been used in previous studies[17–19,41,73].

## Image processing and lesion segmentation

The UNETR transformer segmentation model[74], a variant of the U-Net convolutional neural network that has been tailored to volume segmentation, was used to automatically segment the lesions in each DWI. The architecture comprises a stack of transformers as the encoder, connected to the decoder through skip connections. The training dataset contained 1803 DWI with b1000, chosen for the high contrast between acute stroke lesions and the surrounding brain tissues. From this dataset, 1256 images comprised a group for manual validation, a subset documented in the published study by Mah et al.[18] and 547 images were segmented using a simpler U-Net model and then manually corrected by a neurologist (PN) experienced in the task. All (b0, b1000) pairs were non-linearly registered to the MNI template, where the b1000 was co-registered to the b0, and resliced to 2 mm$^3$ resolution[73]. The resultant volumes were $91 \times 109 \times 91$. Five UNETR models were trained, each time using a different division of the dataset into a training set (80% of the data) and a validation set (20%). The best-performing model was selected for further use. Both the Dice similarity coefficient and binary cross-entropy were used to train the model. The Dice similarity coefficient alone was used to evaluate the performance on the validation set.

The signal intensities were normalized to the range 0–1. CoordConv[75] channels were concatenated to each image, to provide spatial context to the convolutional filters. Random data augmentation functions were used to improve the generalization of the model further. These functions included histogram transformations, small affine and elastic transformations, and midline reflections.

## Disconnectome representation

Ischaemic lesions do not substantially discriminate between tissue classes. Their impact on grey matter functional parcellations is ideally quantified not only by grey matter damage but also by the grey matter disconnections white matter damage incurs. For each lesion in the dataset, a disconnectome representation was therefore generated, providing a probabilistic distribution of affected tissue, including grey matter, resulting from focal ischaemia.

We employed the Disconnectome Map method from BCBtoolkit to achieve this[40]. Individual-level white matter connectivity is not available at the necessary data scale. Furthermore, the long acquisition times necessary for white matter tractography render it infeasible in the acute setting, and relying on it would introduce a bias against scanning-intolerant patients. In conformity with established practice[76,77] we therefore used a reference dataset of healthy participants' tractographies. For each lesion mask, segmented as described, the white matter fibres passing through the lesion were tracked for each of 178 healthy tractographies[41], computed from the Human Connectome Project 7T diffusion imaging dataset[78]. 178 binary disconnection masks were obtained, from which a probabilistic disconnectome map was derived for each patient. This map shows, for each voxel, the proportion of individuals in the healthy dataset that have white matter connections to the lesion.

## Compressed data-driven lesion representations

Image volumes, whether lesion masks or disconnectomes, contain many redundant variables arising from the characteristic distribution of ischaemic damage. Redundancy harms computational efficiency and model stability. Creating compressed, concise representations shorn of redundancy furthermore allows explicit control over the richness of the lesion representation, allowing us to quantify the value of its descriptive detail. Since the optimal method of representation for this data is unknown: we tested a range of methods, including principal component analysis (PCA)[50], non-negative matrix factorization (NMF)[49], and volume deep auto-encoders (standard (AE) and variational (VAE)[48]. We independently evaluated downstream models employing each method. A VAE was used in addition to an AE because the Bayesian prior has a regularizing effect on the representation.

Each model was trained and validated using 10-fold cross-validation. The 2830 lesion–disconnectome pairs, constituting each individual's earliest image acquisition, were divided randomly into 10 training: validation splits. All images were assigned once to a test set. The remaining images were then randomly distributed across the 10 training sets, ensuring that data from the same patient never appeared in both compartments of a training: validation split. Each representation method was evaluated with embedding dimensionalities 2, 5, 10, 15, 20, 25, 30, 35, 40, 45, 50.

**Principal component analysis.** The standard implementation of PCA from SciKit-Learn[79], with default parameters was used. PCA is limited to capturing linear relations, but admits a closed form solution. The principal components for ischaemic mask lesion archetypes are visualized in Supplementary Fig. 6. The principal components for disconnectome archetypes are visualized in Supplementary Fig. 8.

**Non-negative matrix factorization.** The use of NMF in this setting is well established[17]. The Nimfa library[80], with 'random vcol' initialization was used. Visualizations for lesion archetypes are shown in Supplementary Fig. 5 and for disconnectome archetypes in Supplementary Fig. 7.

**Deep auto-encoders.** The AE's encoder was composed of four layers of three-dimensional ResNet-type convolutional blocks, followed by fully connected layers. The decoder used the same architecture, in reverse. The VAE's architecture was the same, except the encoder produced the mean and standard deviation of the $m$-dimensional posterior latent distribution. Samples were drawn from the VAE's posterior with the reparameterization trick[48]. A batch size of 10 was used. We trained for a minimum of 16 epochs, and a maximum of 32. If the loss failed to improve for 4 consecutive epochs then training was terminated early.

## Atlas-based lesion representations

We created a set of baseline representations from simple vascular territories of the kind commonly used in stroke studies[81]. Liu et al. describe atlases of vascular territories with varying levels of arterial tree specificity[51]. The territories indexed by the simplest arterial map were supplied by the anterior or posterior arterial systems[82]. They also provide[51] a specific atlas including the territories supplied by specific major arteries: anterior cerebral artery, middle cerebral artery, posterior cerebral artery and vertebrobasilar artery. The baseline representations were computed by identifying the occluded artery, as the lateralized vascular territory that overlapped the most with the lesion representation, as measured by Dice.

## Functional deficit modelling

Our virtual trials begin with recruitment, where participants are selected based on the overlap between their ischaemic stroke lesion segmentation and a functional–anatomical network. These networks are parcellated into anatomical regions such that, if sufficiently lesioned, a corresponding functional deficit is modelled to be present. The deficit itself was derived by agglomerative clustering of open

source meta-analytic data from the neurological and neuroscientific literature, compiled by NeuroQuery[34], as visualized in Fig. 4. The process is outlined below.

**Functional parcellation.** The NeuroQuery[34] dataset consists of predicted neural activation maps, where voxel intensities are z-scores describing the association at that point in MNI space with a specific n-gram drawn from a large corpus of neuroscientific papers. Each image volume comprised $46 \times 55 \times 46$ voxels at $4\,mm^3$ resolution in MNI space. Volumes associated with n-grams that could not be plausibly associated with cognitive and behavioural functions were manually excluded, leaving 2,095 n-grams for analysis. A binary grey matter mask was generated by thresholding the grey matter tissue probability map[83] at 0.2. To match the template space of the functional data, nearest neighbour resampling was used to reshape this $1.5\,mm^3$ $121 \times 145 \times 121$ mask to $46 \times 55 \times 46$ voxels, leaving 21,410 nonzero voxels. Grey matter was selected to isolate neural substrates plausibly associated with functional deficits caused by lesions. White matter (dis)connectivity was incorporated downstream using disconnectome analysis to infer probabilistic estimates of additional disrupted grey matter resulting from each lesion[41]. The grey matter voxels from the 2095 selected NeuroQuery volumes were reshaped into a $2095 \times 21410$ matrix. Each column represented a grey matter voxel, and each row quantified the functional association with each of the neuroscientific terms.

We applied agglomerative clustering to the 21,410 columns[84], grouping voxels hierarchically based on their functional dependence. Distance between clusters was correlated with functional dissimilarity. Agglomeration of voxels was then performed by adjusting the critical distance threshold, forming clusters of voxels according to their functional similarity (Fig. 4), without consideration of spatial relations. A custom programme was developed for rapid post-hoc manipulation of the distance threshold. Voxels from discrete connected components of volumes $<4^3$ voxels were reclassified using K-nearest neighbours ($K = 5$) to mitigate the risks of potential low-effect size artefacts resulting from underlying noise. Figure 4 shows the functional parcellation at ten different levels of the hierarchy. Supplementary Figs. 1 and 2 show further visualizations of the 16-group functional parcellation. The parcellation is available in Neuroimaging Informatics Technology Initiative (NIfTI) format.

The n-grams associated with each of the 16 functional networks respectively were recovered by computing similarity metrics with the NeuroQuery volumes[34], giving a ranking of functional similarity for each network. For each network, the n-grams ranked in the top ten had a coherent functional theme (Supplementary Fig. 3). A summary functional theme for each was manually determined on inspection of this list. Where functional–anatomical associations were expected–given the neuroscientific literature e.g., memory–hippocampi–these were corroborated by the automatic n-gram rankings.

**Ground truth modelling**
The availability of both factual and counterfactual outcomes in our virtual trials enables us to fully evaluate the prescriptive inference. With real data alone this is impossible due to the fundamental problem of causal inference[27,37]. Our virtual trial data combines real patient data with empirically-informed patterns of treatment responsiveness. Given this semi-synthetic ground truth, we can simulate both random and non-random treatment allocation policies, with corresponding observational outcomes. An individualized prescription can then be inferred from probabilistic models of outcomes conditioned on treatment. These prescriptions can then be evaluated using the virtual ground truth, which would otherwise have been partially unobserved.

Each network in the 16-group functional parcellation is divided into two subnetworks. The subdivision, described in the following section, is based on neurotransmitter receptomes[36] or genetic

transcriptomes[35] (Supplementary Figs. 11–74). This yields two targets for treatment that could plausibly be differentially modulated by our virtual treatments. The modelling of treatment targets in this manner allows us to synthesize virtual trial data, where each individual's responsiveness to a virtual treatment can be determined by the overlap of these subnetworks with the lesion anatomy. Note the treatment targets were based on plausible anatomical–physiological structure, rather than any specific established therapeutic process or molecular identity.

The probabilistic disconnectome representations were binarized at 0.5. The voxel-wise dot-product between each binary lesion representation and each of the 32 responsive-determining subnetworks was computed. If this measure of overlap exceeded 5% of the volume of each subnetwork, the lesion was designated as responsive to the corresponding treatment. This threshold is within the range of critical damage thresholds considered by Mah et al.[18]. In this way, we generate the data for the virtual trials from the 16 functional networks, with both lesion masks or disconnectomes, and responsiveness informed by meta-analytic neurotransmitter receptor or genetic transcription distributions. The tables describing the sample sizes, separability and overlap are available in Supplementary Tables 1 and 2.

**Neurotransmitter receptome.** Hansen et al.[36] conduct a large-scale analysis of neurotransmitter organization within the human neocortex. Their analysis was based on positron emission tomography (PET) data, collected from >1200 healthy individuals. Radiotracers were administered that targeted 19 neurotransmitter receptors, across 9 systems: 5-hydroxytryptamine (5HT), dopamine, noradrenaline, histamine, acetylcholine (ACh), cannabinoid, opioid, glutamate and γ-aminobutyric acid (GABA).

To determine the preponderance of the receptor distributions within each functional network, we computed the Dice similarity coefficient between the binarization of each network and the group-average of PET image z-scores for each receptor. This was performed at both the level of the specific receptor subtypes, and at the level of the neurotransmitter system. For each functional network, voxels were assigned to whichever of the two most preponderant receptor types (Supplementary Table 3 and Supplementary Figs. 11–42) had the greatest z-score. This generates contrasting subnetwork pairs describing plausible treatment targets that can be differentially modulated.

**Genetic transcriptome.** The Allen Institute provides microarray ribonucleic acid (RNA) data with an 'all genes, all structures' approach, sampled across six cadaveric adult control brains[35]. Aggregating these data yielded 3702 samples, across 58,691 genes per sample. The spatial co-ordinates of each sample site were supplied in MNI space.

For each of the functional networks, samples located within each network mask were selected and then agglomeratively clustered into two groups according to their microarray RNA data[84]. This provides support for a subdivision based on molecular dissimilarity. The microarray data do not cover every grey matter voxel in the brain. With unobserved values set to 0, the volumes were convolved with a Gaussian kernel, to provide smooth estimates of the unobserved values. The standard deviation of the kernel was optimized to simultaneously balance the two classes while minimizing the number of connected components. The functional subnetworks acquired from this method are visualized in Supplementary Figs. 43–74. Note the objective here was to derive plausible spatial distributions of responsiveness, as far as available data allows.

**Treatment response heterogeneity.** The preceding steps yielded an array of functional networks spanning the entire brain, with two sets of anatomical–physiological subdivisions each, admitting plausible patterns of heterogeneous treatment responsiveness. Each lesion

representation that surpassed the 5% threshold with respect to at least one of the treatment-responsive subnetworks was included in virtual trial simulations. Some lesions were responsiveness to both virtual treatments, and therefore had no 'optimal' treatment. These lesions were excluded from balanced accuracy computations, but not precision-in-estimation-of-heterogeneous-effects, PEHE[32], because PEHE penalizes the model for inferring superiority given equivalent treatments.

The virtual trial data derived from transcriptome and receptome subnetworks enable us to evaluate the model in the presence of heterogeneous responses to treatment, within cohorts presenting with observable functional deficits. The evaluation uses semi-synthetic observational data, with non-random treatment allocation and response noise specified using hyperparameters, as described in the following sections.

**Modelling treatment allocation policies.** Estimates of treatment outcomes can be degraded by biases in the data. RCTs mitigate treatment–outcome confounding. For example, if a patient is more likely to receive a treatment in the context of severe disease associated with worse outcomes, a naïve comparison against those who do not receive the treatment will be contaminated by the confounding effects of disease severity. Observational data is especially prone to treatment–outcome confounding, as treatments are typically guided by prior beliefs about their efficacy. We must therefore evaluate the impact of confounding—as well as the expressivity of the lesion representation—in our simulation framework, so that its effect on prescriptive inference can be estimated.

**Observable confounding.** We used the co-ordinates of the lesion centroid to simulate the treatment–outcome confounding effects, as virtual treatment responsiveness varies with location. For each pair ($A$, $B$) of response-determining subnetworks, we computed the centroid of A, ($c_{x,A}, c_{y,A}, c_{z,A}$), and the centroid of B, ($c_{x,B}, c_{y,B}, c_{z,B}$). The axis along which the two centroids differ the most is then calculated, as argmax($[|c_{x,A} - c_{x,B}|, |c_{y,A} - c_{y,B}|, |c_{z,A} - c_{z,B}|]$). For each pair ($A$, $B$), the lesion representations are then arranged in increasing order of the centroid co-ordinate along this axis.

For the $k$th lesion of the ordered list, the corresponding probability with which treatment B is allocated rather than treatment A is the $k$th element of the vector $\boldsymbol{\rho}$ = linspace($0.5 - b_{obs}, 0.5 + b_{obs}, N$). Here, $N$ is the number of participants, and the degree of confounding is controlled by the confounding hyperparameter $b_{obs} \in [0, 0.5]$. If $b_{obs} = 0$, each patient has an equal probability of allocation to treatment groups $A$ or $B$, effectively simulating a RCT. We allocate treatments differently for $b_{obs} \in (0.5, 1]$. In this case, the extremes of the ordered data are allocated deterministically depending on their relation to the training set median.

**Unobservable confounding.** An additional mechanism was implemented to model allocation to treatment by unobserved factors. By unobserved factors, we mean features not predictable from the phenotype characterization, $\mathbf{x}_i$. This mechanism makes treatment allocation depend on true treatment responsiveness, $w^*$—allocation is now sampled from a Bernoulli distribution, with probability $\mathbb{P}(W = w^*) = b_{un\_obs}$. Here, $b_{un\_obs}$ is the hyperparameter that determines the degree of unobservable confounding.

**Response noise with fixed treatment and recovery effects.** The causal inference literature distinguishes recipients of treatments whose outcome is independent of treatment, as 'doomed' and 'immune'[85]. In our experiments, we vary the probability that participants are designated as 'doomed', using a fixed treatment effect (TE) hyperparameter, where TE = $\mathbb{P}(Y = 1 | W = w^*)$. This is the probability that there is a favourable outcome, $Y = 1$, given an individually suitable

treatment allocation, $W = w^*$. We use an additional hyperparameter to designate participants as 'immune', using a fixed recovery effect (RE) hyperparameter, where RE = $\mathbb{P}(Y = 1)$. This is the probability of a favourable outcome. A synthetic ground truth with TE = 1 and RE = 0 would therefore have no response noise, with all patients responding according to whether their stroke phenotype is responsive to the intervention that they were allocated to receive.

### Prescriptive inference

**Problem setup.** It is common to replace the individualized treatment effect, $\tau_i$, with the conditional average treatment effect[86] (CATE). This is defined as $\hat{\tau}(\mathbf{x}_i) : = \mathbb{E}[Y_i^{(A)} - Y_i^{(B)} | \boldsymbol{X} = \mathbf{x}_i]$. We can estimate the CATE, based on the following set of assumptions: the interventions are sufficiently well-defined (consistency); the allocation to treatment depends only on observed covariates (conditional exchangeability); and the probability of receiving each available treatment is non-zero across all individuals (positivity)[39,87].

When these assumptions are satisfied, the CATE is referred to as identifiable, i.e., it can be estimated[86]. We experiment with values of the TE, RE and treatment–outcome confounding that stretch these assumptions to breaking point. The experiments involve simulated noise levels that exist in real healthcare data[88]. Simulating observable confounding illuminates the effect of stressing conditional exchangeability, while simulating unobservable confounding illuminates the effect of its violation. Extreme confounding weakens positivity/overlap, making some individuals highly likely to be allocated to one treatment over the other. When the hyperparameter that controls confoundedness is 1, allocation becomes deterministic, and we observe the effect of invalidating positivity.

Given these three assumptions, we can estimate the CATE as the difference between two probabilities:

$$
\begin{aligned}
\hat{\tau}(\mathbf{x}_i) &= \mathbb{E}[Y_i^{(A)} | \boldsymbol{X} = \mathbf{x}_i] - \mathbb{E}\left[Y_i^{(B)} | \boldsymbol{X} = \mathbf{x}_i\right], \text{ linearity,} \\
&= \mathbb{E}\left[Y_i^{(A)} | W = A, \boldsymbol{X} = \mathbf{x}_i\right] - \mathbb{E}\left[Y_i^{(B)} | W = B, \boldsymbol{X} = \mathbf{x}_i\right], \text{ cond.exchangeability,} \\
&= \mathbb{E}\left[Y_i | W = A, \boldsymbol{X} = \mathbf{x}_i\right] - \mathbb{E}\left[Y_i | W = B, \boldsymbol{X} = \mathbf{x}_i\right], \text{ SUTVA \& consistency,} \\
&= \mathbb{E}\left[Y | W = A, \boldsymbol{X} = \mathbf{x}_i\right] - \mathbb{E}\left[Y | W = B, \boldsymbol{X} = \mathbf{x}_i\right], \text{ equal distributions,} \\
&= \mathbb{P}\left(Y = 1 | W = A, \boldsymbol{X} = \mathbf{x}_i\right) - \mathbb{P}\left(Y = 1 | W = B, \boldsymbol{X} = \mathbf{x}_i\right), \text{ binary outcome.}
\end{aligned}
$$

Here, the final two probabilities are realized using supervised learning, e.g. decision trees, Gaussian processes and logistic regression.

**Assumptions in more detail.** Consistency means that, if $W_i = A$ then $Y_i = Y_i^{(A)}$, and if $W_i = B$ then $Y_i = Y_i^{(B)}$. It also implies that the treatment is always applied in the same manner (e.g., the same dose and route of drug administration). In some of the causal modelling literature, an assumption of stable unit treatment value assumption, or SUTVA, is stated, conveying the same condition. This further makes explicit the underlying assumption of independence between units: that one individual's receipt of a treatment has no effect on another individual's treatment allocation, also encapsulating the concept of no variation within defined treatments.

Conditional exchangeability guarantees the absence of unobserved confounders (sometimes referred to as unconfoundedness). The potential outcomes, $Y^{(A)}$ and $Y^{(B)}$ are independent of the treatment allocation, $W$, given the phenotype characterization, $\boldsymbol{X}$, i.e., $Y^{(A)} \perp W | \boldsymbol{X}$ and $Y^{(B)} \perp W | \boldsymbol{X}$. This is also referred to as ignorability. The randomization of treatment in RCTs promotes conditional exchangeability[89].

Positivity, or overlap, means that before allocation, both treatment allocation probabilities are non-zero, encouraging inclusion of data relating to both treatments across subpopulations: $0 < \mathbb{P}(W = w | \boldsymbol{X} = \mathbf{x}) < 1$ for all $w$, $\mathbf{x}$. This is sometimes referred to as strong ignorability.

**Prescriptive inference.** We infer treatment effects using two frameworks from the causal inference literature[29,31,90–92]. The one-model/SLearner[90] framework involves modelling $\mathbb{P}(Y = 1| W = w, \boldsymbol{X} = \mathbf{x})$ using a single outcome classifier, fitted to pairs $(\mathbf{x}, \delta_w^A)$, where $\delta_w^A$ is the Kronecker delta, which equals 1 when $w = A$ and 0 otherwise. The optimal treatment is then inferred to be the treatment with the greatest corresponding probability.

The two-model/TLearner[90] approach employs two separate outcome-predicting classifiers. The first classifier is fitted to the participants allocated to treatment A; the second to treatment B. For each unseen individual and each treatment, the probability of causing a favourable outcome is computed from the corresponding model. The optimal treatment is then inferred to be the treatment with the greatest corresponding probability.

**Models.** We used the following range of contrasting architectures for the one-model/SLearner and two-model/TLearner: extremely randomized trees[44], random forest[43], Gaussian process[47], XGBoost[45] and logistic regression[46], from SciKit-Learn[79] and the XGBoost[45] Python packages, each with default parameter settings.

**Model evaluation.** The fidelity of the prescriptive inference was assessed with balanced accuracy and PEHE. In all cases, we use 10-fold cross-validation and report validation set accuracies. The training: validation splits were selected randomly at the start, before any models were fitted. In particular, the splits were selected before the phenotype characterizations were computed. The prescriptive inference was evaluated on the validation sets only.

**Model evaluation: balanced accuracy.** Balanced accuracy is only computed for virtual trial participants who are responsive to treatment A or B, but not both. This was computed as the mean of the proportions where the preferable treatment was correctly inferred. We reported most of our performance results in terms of balanced accuracy because most of our virtual trials involved class imbalance.

**Model evaluation: precision in estimation of heterogeneous effects (PEHE).** The PEHE[32,93] is a continuous quantification of the CATE estimation. It was computed across all virtual trial participants in each validation set. The participants included those responsive to both treatments equally ($\tau_i = 0$). This allowed us to evaluate whether the model could indeed infer responsiveness to both treatments. The measure was computed as $\sqrt{\frac{1}{N_{val}}\sum_{i=1}^{N_{val}}(\hat{\tau}(\mathbf{x}_i) - \tau_i)^2}$, for $N_{val}$ validation set participants. The CATE, $\hat{\tau}$, was computed as the difference of two supervised learning model predictions, as described in *Problem setup*.

**Comparison.** While the PEHE is usefully a continuous measure of predictive performance, the balanced accuracy measures how closely the inferred prescription matches the optimal prescription. This is the question that motivates our work. For this reason, we include PEHE only for completeness.

Low-dimensional representations, $\mathbf{x}_i$, were used as a baseline in our experiments. They were derived from the affected vascular territories arising from the broad anterior or posterior arterial systems, as described in *Atlas-based lesion representations*. This distinction, between anterior and posterior vascular occlusion, is incorporated in the UK clinical guidance as a factor influencing the decision between the acute ischaemic stroke treatments of mechanical thrombectomy or intravenous thrombolysis[81]. This baseline was computed with unconfounded allocation, as a surrogate for RCTs. When comparisons were made to this baseline, treatment effect and recovery effect were kept equal. This is because randomization does not control for these features of interventions.

Prescriptive performance, with phenotype characterization, $\mathbf{x}_i$, derived from ischaemic lesion masks was presented separately from $\mathbf{x}_i$ derived from disconnectomes. These results were summarized using their aggregate mean, and contrasted using independent sample two-sided *t*-tests. Superiority was determined using the critical *p*-value, computed using the Benjamini–Hochberg method[94]. The false discovery rate reflected 0.05 significance level.

## Inclusion & Ethics Statement

This study was performed under ethical approval by the West London & GTAC research ethics committee for consentless use of fully anonymized data. The data is an unselected sample based on clinical diagnosis of acute ischaemic stroke; no other specific inclusion or exclusion criteria were used. It should therefore be proportionally representative of the acute ischaemic stroke patient population presenting to hospital in London, UK.

## Reporting summary

Further information on research design is available in the Nature Portfolio Reporting Summary linked to this article.

## Data availability

Data and code relating to this article are available publicly from https://github.com/high-dimensional/individualized_prescriptive_inference. Functional grey matter parcellations including their subdivided versions are available in the enclosed 'atlases' directory. Irrevocably anonymized lesion segmentations in template space, complete with, where available, age and sex, but no other information that could identify any individual, are available in the 'lesions.zip' file within this repository. The filenames are arbitrarily numbered in the format "lesion{arbitrary_id}_{age}_{sex}.nii.gz", with age and/or sex filled with NA when unavailable. All images are binary lesion segmentations in MNI space ((91, 109, 91) voxels of 2 mm³). Data used from external sources, including transcriptome and neurotransmitter receptor data, are publically available as described in each respective cited resource. Source data are provided with this paper.

## Code availability

The code relating to this article is available from https://github.com/high-dimensional/individualized_prescriptive_inference. The code for establishing the functional parcellation and its subdivisions is available publicly in the enclosed directory 'functional_parcellation' and the representational preprocessing and full prescriptive simulation setup codes are publicly available in correspondingly named scripts inside the directory 'software'.

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

## Acknowledgements

Funded by Wellcome (213038, P.N.), UCLH NIHR Biomedical Research Centre (NIHR-INF-0840, D.G.) and UKRI EPSRC (2252409, D.G.). Other funding sources were Medical Research Council (MR/X00046X/1, J.K.R.) and UKRI EPSRC (L016478, G.P.). The authors acknowledge the use of the UCL Myriad High Performance Computing Facility (Myriad@UCL), and associated support services, in the completion of this work.

## Author contributions

D.G. and P.N. conceptualized the study and wrote the article. D.G. wrote the software. C.F. and G.P. contributed towards the lesion (and disconnectome) dataset preprocessing and J.K.R. manually reviewed the lesion segmentations. T.X. and A.J. contributed towards the methods. H.R.J., J.C., S.O., G.R. and A.J. internally reviewed the article.

## Competing interests

D.G., J.C., S.O., G.R. and P.N. are affiliated with Hologen, a healthcare deep generative modelling company. C.F., G.P., J.K.R., T.X., H.R.J. and A.J. declare no competing interests.
