## [Peer Review File · Nature Communications]

Individualized prescriptive inference in ischaemic stroke

Corresponding Author: Dr Dominic Giles

Version 0:

Reviewer comments:

Reviewer #1

(Remarks to the Author)

This is an interesting project utilizing thorough design and data analysis, with the aim to compare the individual-level fidelity of the traditional interventional randomized allocation system versus a more flexible modelling of non-randomized data based on deep representation learning. The investigators make use of big data (using high-resolution maps of 4,119 acute ischaemic lesions), including anatomical patterns of functional activation, neurotransmitter receptors, gene expression, and other neural properties of spatial structure to evaluate treatment susceptibility. In the era of personalized medicine, this work is more relevant than ever.

1. The introduction is rather long. The authors should be able to address the main question of their work in less than 2 pages. More relevant information may be presented in the Discussion section.
2. I would suggest toning down a bit the statements relevant to RCTs, such as "Not only may we not assume that conventional RCTs will converge on the correct inference here, there are strong reasons for doubting the fundamental grounds on which they rest".

(Remarks on code availability)

Reviewer #2

(Remarks to the Author)

Giles et al. report the results of performing a comprehensive study of ascertaining therapeutic choices for ischemic stroke patients using a combination of state-of-the-art computational and statistical techniques, paired with a wide variety of data sources, including both high-resolution maps of patient acute ischemic lesions and meta-analytic information on connective, genetic, and functional expression. The advanced methodological and computational aspects of the work are justified by the aim of overcoming the limitations of methods that are considered as "the standard" in health/medical sciences research, and which have been demonstrated to be incapable of making use of expansive datasets and statistical tools now relatively widely available. The manuscript is, on the whole, well written, and it has the potential to be an important contribution in applied medical sciences research, as well as a notable contribution in applied statistics and/or machine learning. There are several points that the authors should consider improving, both in the presentation of the manuscript and in some of the simulation experiments performed, in order for the manuscript to have a broader impact and accessibility. Some specific comments follow.

* Since the main-text treatment of the causal inference ideas avoids the use of notation, some phrases can be confusing, e.g., "likely success of an intervention" (sentence 1), which could reference any of $E[Y(1) - Y(0)]$, the average treatment effect; $Y(1) - Y(0)$, an individual treatment effect; or $P\{E[Y(1) - Y(0)] > 0\}$ and/or $P\{Y(1) - Y(0) > 0\}$, which are probabilities of counterfactual quantities. Some use of notation, or greater care with language, would improve the presentation.

* In discussing the possible over-reliance on RCTs, the authors note that the "established inferential approach...assumes our knowledge of individuals is limited to the average of the population..." (sentence 2), without making clear what the "established inferential approach" is; this is confusing since it is less a criticism of RCTs, given that randomization is enough to identify the *full distribution* of potential outcomes, than it is of too much focus being placed on simple estimands, like the average treatment effect, without thought being spared for whether the estimand actually aligns with the underlying science or policy questions that motivate the research.

* In the first paragraph, the authors make it an important point, related to the above bullet, that the ATE may simply not be an interesting contrast; however, they then mistakenly claim that conditional ATEs (CATEs) cannot be learned from RCTs ("heterogeneity is not detectable within the trial evaluation itself, nor necessarily in any subsequent subgroup analysis based on the same description of the population"). This is not true, as has been detailed in the causal inference and (bio)statistics literature, e.g., VanderWeele et al. (2019), "Selecting optimal subgroups for treatment using many covariates" (Epidemiology); Luedtke & van der Laan (2017), "Evaluating the impact of treating the optimal subgroup" (Statistical Methods in Medical Research); Kessler et al. (2019), "Machine learning methods for developing precision treatment rules with observational data" (Behaviour Research and Therapy); etc. While there are many approaches to learning heterogeneous treatment effects, and the authors seem to use the "X-learner" framework, all such frameworks, including advances such as the "DR-learner", ought to be compatible with randomized trials (RCTs, A/B tests).

* The authors make a critical point for precision/personalized medicine, that "Inferences about individuals may now be informed by the local subpopulation to which they belong", yet then go on to conflate individual treatment effects (ITEs) $Y_i(1) - Y_i(0)$ with conditional average treatment effects (CATEs) $E[Y(1) - Y(0) | V = v]$ for some subgroup $V = v$. Since ITEs are never identified due to the fundamental theorem of causal inference, the focus on individualized effects (or ITEs) draws attention away from the more practical goal of pursuing inference about CATEs.

* Throughout, the manuscript refers to "unbiased treatment allocation", which is meant to be synonymous with "unconfounded" treatment allocation; the latter terminology is more standard, and it may be useful to avoid the term "unbiased", as it risks conflating randomization, which yields unconfoundedness in terms of counterfactuals, with the concept of "balance", which is the empirically testable analog that may be viewed as "bias".

* In comparing standard practice in evaluating RCTs with the proposed tact of flexible modeling (i.e., "the comparative importance of minimizing treatment allocation bias vs maximizing model expressivity and flexibility in inferring the optimal treatment for each individual patient"), the tension appears to be misplaced, as there is no apparent incompatibility between employing flexible modeling strategies (i.e., for an outcome regression) "within" an RCT, except for dogma about how inference ought to be performed (e.g., difference-in-means or parametric g-computation). The authors message would likely be better received if they were to note that the proposed strategy could actually be employed in RCTs with large enough sample size, as has recently been done outside of the literature on neurology/neuroimaging.

* Some of the authors' statements are too far-reaching, even if well-intentioned, e.g., "This comparison *can only be performed*", or "to use data within the RCT framework is to beg the question of what the right standard should be: our task here"; moreover, the latter of these is not even true, as data from an RCT could certainly be used to learn about treatment effect heterogeneity (as noted above), as long as key baseline covariates are collected as part of the trial. The false dichotomy of casting principled/flexible modeling approaches against randomized studies does a disservice to both fronts of scientific inquiry, i.e., surely the way forward is not to set aside RCTs but to learn to also incorporate flexible modeling and observational studies as part of the "scientific corpus".

* In the Methods section, standard assumptions for identification of causal effects (consistency, positivity, no unmeasured confounding) are presented, but it would help to give an example of an identification result (e.g., for the ATE) while also connecting this to the treatment effect heterogeneity parameter that is of interest, i.e., the CATE. Notably, the simulation studies include a "bias parameter" for treatment allocation, which can be set to 1 (deterministic treatment based on patient phenotype); this is only mentioned in the main text, without noting the underlying issue of resulting in non-identification of causal parameters, by violating the positivity assumption. While this fact is acknowledged in the methods, it may escape notice of some of the readership and is not an interesting case anyway, so it may be better to simply truncate the simulations at a bias parameter of 0.95, where one may encounter positivity violations in the data but not in the underlying data-generating mechanism, since, in the latter case, no causal inference is possible at all.

* The simulation studies appear well-constructed and thorough; however, some of the design decisions underlying them are left unexplained. For example, the authors note that "treatment allocation [was varied] from random (no bias) to directly determined by the location of the centroid of each lesion (high bias)", which seems to capture different scientific contexts, e.g., what would be the reason to pursue random treatment assignment across lesions types regardless of location? Presumably, this is not how medical practice actually works, as one would only randomize treatment strategies if a given strategy is completely novel and, even then, only for a subset of lesion locations/types and against standard-of-care. My suggestion would be to clarify the decision-making behind some of the simulation parameters and to place these in the appropriate context of a medical or computational study.

* I found the conclusion of the Results section, which states "rejecting the null hypothesis of no superiority of richly expressive representations under biased treatment allocation conditions vs low-dimensional models under randomization", to provide a helpful, clear framing of the goal of the study, which I would suggest be made as clear earlier. To this point, it is an interesting, practically useful computational study, but the exact goal is vaguely stated as comparing flexible modeling strategies with standard inferential practice in RCTs. Related to this, the authors make several strong statements in this section that ought to be toned down, including both "[a] high-dimensional representation, even in the setting of strongly biased allocation, was almost always at least non-inferior to a RCT employing low-dimensional models, and in the vast majority substantially superior" and "higher prescriptive fidelity even from a markedly biased and noisy trial modelled with a richly expressive representation than from a fully randomized trial at the same noise level modelled with a simple one", the latter of which may simply be over-interpretation/generalization of some technical artifacts of the authors' computational study. From theoretical arguments, one might expect a too-simple model to fail, even in an RCT, when the model cannot

capture the complexity underlying data well enough; however, simply using a more complex model in an RCT should improve this, as the RCT will protect against confounding and more flexible modeling strategies may succeed in picking up smaller treatment effects...

* Related to the above, there is some overly strong language used in the Discussion section as well, e.g., "there are strong reasons for doubting the fundamental grounds on which [RCTs] rest" (whereas, in reality, RCTs are simply not a panacea and, indeed, can be poorly performed -- so, more flexible modeling and openness to the use of observational studies would help to improve evidence-generation in modern medicine). On the other hand, some points -- like "A failure of inference here cannot be distinguished from a failure of the intervention, and if the inferential approach is never questioned, replication will not correct the error but typically entrench it." -- are very well put!

* The Methods section is also well-written, though some aspects could be improved, including a relative paucity of notation, some of which could go a long way to improving clarity (see a previous suggestion about working out an identification result to ground the arguments for pursuing inference about the CATE). In this section, some statements could be made more precise, e.g., "treatment-related confounding" (page 23, line 619) would be better stated as "treatment-outcome confounding". In this section, τ_i is said to be the individual treatment effect and $\hat{\tau}$ is then said to be an approximation of this, yet this is not, generally speaking, true. $\hat{\tau}$, which appears to be a conditional average treatment effect, approximates τ_i only as the support set $X = x$ grows to approximate all features that define "individuals" in the population, e.g., $\dim(X) \rightarrow \infty$. It might be better to rely on standard results to simply note that $\hat{\tau}$ is an identifiable causal parameter whose interpretation is closest to what might be desired in the context of personalized/precision medicine. Also, $\hat{\tau}$, if it is actually meant to be the CATE, can be better expressed as $\hat{\tau} = E[P(Y = 1 | W = A, X) - P(Y = 1 | W = B, X) | X = x]$, where the $X = x$ could also be written as $V = v$ for some V that is a subset of X , allowing for effect heterogeneity to be learned in subgroups that are not necessarily concordant with the full set of putative measured confounders X .

* In describing the "one-model" and "two-model" approaches to inference on heterogeneous treatment effects, it would be helpful to describe these in notation and to specifically refer to algorithmic frameworks that are employed. For example, the authors specifically cite distinct works by Kunzel et al. (2019), Sontag, and van de Schaar, but how these are exactly used is not elaborated upon. Some description of the techniques, e.g., X-learner, would help to clarify how these ideas are built upon or adapted in this work.

* Throughout, the authors note the use of the Benjamini-Hochberg (BH) correction, which controls the false discovery rate (FDR), for which the standard justification of tolerating some false discoveries is given. I would question the relevance of the FDR in this setting, as one would presumably prefer to err on the side of failing to discover personalized treatment strategies rather than take the more liberal view of discovering some personalized treatment strategies that may, in fact, be mistakes. Also, the authors mistakenly note that the BH correction controls the family-wise error rate (FWER) [page 27, line 781-782].

* Rather than using "individualized" throughout, I might recommend that the authors consider using "personalized" or "precision", which are both more vague but carry the advantage of not directly referring to "individualized treatment effects", which are not identifiable, generally speaking.

* The writing makes use of many specialized terms, including those that are not even standard jargon, e.g., "prescriptive inference", "therapeutic inference" (these from the abstract), etc. Since the research lies at an interface between statistics / causal inference, machine learning, and medical research, such terminology can be particularly confusing; for example, after years of teaching (and learning) causal inference, I've never run into such terms as the above though can figure out what they may refer to, e.g., dynamic/personalized treatment regimes or similar. Given the broad readership that ought to be the target, I expect that many readers will be confused at one point or another. Perhaps the authors could attempt to resolve some of this confusion by providing a glossary at the end of the main text that defines such terms as the above (and others like them) in terms of what might be more easily understood by a statistician, medical researcher, etc. Other such terms include, e.g., "semi-synthetic counterfactuals", by which the authors presumably mean regression quantities like $\hat{E}[Y(1)]$ or similar.

* Typo: "...for based on..." (page 9, line 222)

(Remarks on code availability)

Reviewer #3

(Remarks to the Author)

This paper discusses the importance of treatment effects' heterogeneity in reasoning about the efficacy of treatments, particularly in the context of acute stroke lesions. The paper seems to make two key arguments:

(1) RCTs with inadequate accounting of heterogeneity may be missing out on treatments that can benefit subsets of the population

(2) Observational data may be a better source for understanding treatment heterogeneity, and the observed and unobserved confounding biases in this data may have less impact on the optimal treatment rules than not accounting for heterogeneity.

While points (1) and (2) are obviously not new, the authors propose a form of sensitivity analysis to compare the relative importance of richer covariates in observational data in identifying the optimal treatment rules vs. the effect of biases in identifying wrong rules. They do so using semi-synthetic simulations of counterfactual outcomes based on a causal model that is informed by biological plausible mechanisms in the context of stroke lesion data.

Overall, I had a hard time reading this paper and understanding the authors' contributions. The writing is extremely verbose and convoluted to the extent that I think the paper should be completely rewritten to improve clarity and conciseness if accepted.

In addition I have the following major technical comments:

1- The authors build a semi-synthetic model to simulate the potential outcomes of each subject in the study and use that model to conduct a "virtual" interventional trial using rich observational data. The key premise of their proposed method is that this model can be used to assess the relative importance of randomization vs. rich covariate adjustment using modalities only available in observational data. However, I am a bit confused about the logic behind this approach. Wouldn't the validity of your inferences still be limited by the correctness of the synthetic model which is itself going to be affected by confounding biases? That is, you would still need to randomize the intervention in a new trial that only includes patients hypothesized to benefit from the treatment based on your simulation. But isn't this the goal of any observational study for which we can use any ML method for estimating conditional average effects without running a simulation?

2- What is "treatment susceptibility"? Does this mean a propensity score/treatment assignment probability? This is not a standard terminology as far as I am concerned.

3- The key result of the paper is on page 13 where the authors reject the null hypothesis of "no superiority of richly expressive representations under biased treatment allocation conditions vs low-dimensional models under randomization". However, I am not sure I understand the grounds based on which the authors reject the null hypothesis. This seems to be based on ground-truth data from semi-synthetic simulations, but doesn't this result only tell us that "there exists a synthetic model of treatment outcomes where this null hypothesis is false"? How can this be the grounds for reasoning about actual treatment rules in a real-world scenario?

4- I think that the authors proposed idea (a sensitivity analysis of effect of randomization vs. richness of covariates) should be framed as a formal framework with theoretical conditions on the synthetic models for valid inferences. The authors propose a general direction for thinking about treatments heterogeneity but then propose one very specific synthetic model with empirical results with unclear utility. Some relevant literature on sensitivity analysis based on marginal sensitivity models for individual treatment effects also seems to be ignored, though these frameworks try to achieve a very similar objective.

I can't comment on the biological plausibility of the proposed model since it's beyond the scope of my expertise.

(Remarks on code availability)

Reviewer #4

(Remarks to the Author)

The authors describe a complex modelling and simulation exercise to explore how heterogeneity of brain lesions might influence hypothetical treatment responses in acute stroke, based on a model of functional connectivity and receptor distribution, and a large dataset of around 4000 acute ischaemic stroke lesions. They illustrate superior performance of their complex model compared to a simple classification system in predicting the effect of simulated interventions. It is unclear what outcomes – if any – were included in the modelling exercise. It appears that no standard clinical measures of outcome were included and the modelling fidelity was assessed only against modelled effects on receptor maps, connectome maps or similar. This could be clarified.

The comparator approach was to allocate patients on the basis of arterial lesion location. This however is seldom done in clinical trials or practice, beyond the selection of large artery occlusions for thrombectomy. For other types of intervention, vascular occlusion site is of very limited relevance and since identifying one is dependent on imaging interpretation, it is uncommon for this to be used as a prognostic variable at time of randomisation.

Superior performance of a more detailed model is to be expected when compared against much simpler models, not least because the model incorporates a detailed brain lesion that, at least for acute trials, represents an outcome rather than a baseline feature. The reality of acute trials is that typically only a few key features are reliably known at the time of randomisation and treatment allocation.

The authors argue that traditional clinical trial approaches cannot deal with heterogeneity, in the context of stroke, and propose that this can only be modelled by an almost individualised approach.

The argument in places seems to propose that a truly individualised treatment selection is the ultimate aim; indeed, that results from randomised trials should be disregarded since they potentially discard treatments that might be effective for only a specific individual. This goes beyond the precision medicine, or even individualised medicine, concept since these approaches would at least use treatments that have an established evidence base and effect profile based on individual patient features to maximise benefits, minimise adverse effects, or avoid treatments for which there is no biological target. The argument put forward here appears to go significantly beyond these goals and even to advocate for testing therapeutic

interventions solely on the basis of complex modelling without any traditional trial.

A more balanced perspective would consider more cautiously the role for complex modelling and the limitations of this approach. The model assumes availability of complex data – often not available at all, or only at later time points of clinical course – and that averaged models of receptor distribution or connectivity normalised into averaged anatomical space can be applied to individuals. When could such a model be applied in the clinical journey? What types of intervention might be suitable targets for investigation (and which would not be)? How will the real-world performance of this complex model be evaluated?

It seems intrinsically unlikely that a modelling exercise, no matter how complex, would ever replace randomised controlled trials. The generalised critiques that are offered are based on hypotheses rather than clear evidence: it may indeed be the case that effective treatments have been discarded through inadequate characterisation of patient physiology, but this is a proposal with no current evidence to support it.

Available – highly effective - treatments for stroke are based on logical selection criteria (eg exclusion of haemorrhage from trials testing treatments for ischaemic stroke) applied in clinical trials undertaken in large and representative populations, at least at phase 3 stage. Phase 2 trials have generally recruited narrower populations based on reasonable a priori evidence of being potential responders to treatment. There are many examples where treatment choices are informed by characterisation of patient subgroups based on traditional phase 3 trial data and where it is possible to infer different treatment by subgroup interactions that are relevant to patient care (eg age group and severity against treatment effect for intravenous thrombolysis). While a more efficient means of identifying a responder population for phase 2 trials would certainly be advantageous, it is unclear if this approach offers this functionality. Stroke trials usually adjust their analyses for known prognostic markers and plan subgroup analyses around biomarkers that identify important subgroups, including clinical variables such as age, severity, onset to treatment time, or imaging variables such as the volume of ischaemic tissue, site of an arterial occlusion, or extent of established ischaemic damage. Clinical investigators recognise that individual treatment effects are inherently unpredictable given the heterogeneity of age, brain health, comorbidities, medication and individual capacity for recovery, but nonetheless base coherent treatment decisions on reasonably well-informed average outcomes from clinical trials, nuanced by subgroups.

The underlying assumption that the brain injury is the dominant determinant of disability (indeed the only one modelled – the authors contend that “...the disability caused by a stroke is directly related to the underlying disrupted functional anatomy”) - is an oversimplification. Stroke outcome is certainly determined in part by lesion size and location, but the dominant factor in clinical studies is the initial severity measured clinically, which typically renders measures such as ischaemic lesion volume non-significant in multivariable models. The functional outcome is also influenced by pre-stroke levels of disability, comorbidities, age, occurrence of complications (including recurrent stroke) and social support. Clinical trials also have to balance risks and benefits of treatments to estimate risk-benefit balance, which is not all related solely to brain effects (eg systemic bleeding complications).

(Remarks on code availability)

Version 1:

Reviewer comments:

Reviewer #1

(Remarks to the Author)

The authors have adequately addressed all issues raised in the first round of review.
No further comments exist on my behalf.

(Remarks on code availability)

-

Reviewer #2

(Remarks to the Author)

My comments have been sufficiently addressed.

(Remarks on code availability)

Reviewer #4

(Remarks to the Author)

The authors have offered a detailed rebuttal and updated their text.

The manuscript overall continues to lack balance. Broad statements continue to dismiss RCTs and fail to acknowledge the limitations of the present investigation into complex modelling, or acknowledge the current study's own inherent biases. The text overstates the importance of the current investigation in somewhat grandiose statements. It would be appropriate to more clearly acknowledge the limitations and biases of the modelling exercise and consider where and how the hypothesis that has been generated here could be tested.

From the perspective of a clinician and clinical trialist, modelling is involved in current trial designs, and the complexity of the modelling is limited by the availability of relevant data. While the authors argue that a demonstration of superiority of

complex multidimensional modelling might drive acquisition of richer (imaging) data, this argument fails to recognise many practical limitations, discussed further below, which inevitably bias the subgroup in whom suitable complex data can be obtained. Further, the anatomical modelling approach is based on forcing diverse anatomy into an average brain volume, mapped onto averaged maps for receptor distribution, gene expression etc. Individual anatomy is more heterogeneous than the model systems, even less so when the extreme complexity of underlying disease (old infarcts, disruption of white matter integrity etc) is considered.

The text argues that observational data should be considered on the basis of the simulation exercise as having superiority over RCT data. This fails to acknowledge the many serious concerns about observational data, which are often highly biased and misleading even when a seemingly compelling hypothesis can be built around them (aggressive control of hyperglycaemia in ICU patients might be a pertinent example of the dangers of reliance on observational data). Again, the manuscript could better recognise the limitations of the current study and place this (currently very strong) recommendation in more appropriate context.

The Abstract states "randomized controlled trials, [are] based on simple descriptions of presumptively homogeneous populations," which I think is a considerable over-simplification. Similar comments appear elsewhere in the text. RCTs involve many stages of investigation, from first-in-man drug trials, through proof of concept, to generalisable efficacy, to real world implementation and may include the full range of interventions from simple molecules through to complex care protocols. In trials that are seeking to establish the effect of an intervention at an early stage (typically phase 2 trials) the risk-benefit balance of the intervention is not sufficiently characterised, and there are ethical and practical barriers to recruiting large numbers of participants. Strategies to reduce heterogeneity are typically employed through a range of means including selection criteria to focus on a (predicted) responder population with low risk of adverse effects, including clinical and brain imaging approaches. The descriptions of populations are typically not simple and extend to a wide variety of demographic features, physiological variables, medical and medication history, stroke features and imaging. The text states that estimands are always simplified and lack nuance, but this is in large part driven by the constraints of sample size and practical barriers to obtaining all the desirable information in all participants. It may be the case that complex modelling could further assist patient selection for phase 2 trials but this is a hypothesis that requires to be tested, and the constraints over availability of complex anatomical imaging for such an approach need to be recognised as this biases the population and inevitably leads to losses to follow-up. In large phase 3 (or later) trials, heterogeneity is inherent, and is often embraced in the trial design; generalisability of treatment effects that are suggested in early phase trials is important, as is a more realistic estimate of average effect size, and the larger sample sizes that are involved in phase 3 trials allow more reliable subgroup analyses. How complex modelling could assist should be considered more critically. A major limitation of phase 3 trials – particularly in the stroke field, historically - is that sample size is insufficient to allow reliable statistical analysis of subgroups. Unless complex modelling allows sample size to be reduced while retaining robust statistical analyses this approach might not provide any advantage.

The model data set is based on MRIs that were acquired in a hospitalised acute stroke population. Typically MRI in UK hospitals is undertaken in the subacute period 24-48h (or later) after onset. The main paper gives no detail on the population from which these scans were acquired at all, and the supplementary data presents only an age histogram. There is no information at all that would allow a reader to judge how representative a population has been studied – demographics, medical history, stroke onset and presentation, treatment, timings for these, clinical severity and outcomes are all absent. Selection bias is not acknowledged by the authors but is plain from the flow chart (Fig S3) which illustrates that even from the subset of patients for whom imaging was available to review, >20% of scans were excluded due to technical failure or presence of other pathology. Since tolerance of subacute MRI depends on severity of stroke, many severe patients are excluded, as of course are all those who do not survive for long enough to have such imaging. Intolerance of MRI due to agitation, claustrophobia or contraindications such as ferromagnetic foreign bodies or pacemakers further reduces the representation of the stroke population. The study scans may exhibit heterogeneity of anatomical locations, but this dataset is far from representative of all strokes. The argument that acquisition of complex imaging data will be feasible and superior compared with conventional data is undermined by the figure legend that notes that 16% of these datasets lacked the most basic clinical information (age or sex). The current model has therefore selected imaging data alone, and has neither sought nor compared conventional clinical data at either baseline or at follow-up. Having access to a large number of MRIs does not equate to generalisability.

Ischaemic lesions are dynamic in the first days after stroke, and a DWI lesion at 6h offers very limited insight into the extent or topography of the lesion at 24h or 48h. This is an important caveat to the dataset used here, and should temper the language used to support the applicability of the modelling approach. We do not know whether complex modelling in other circumstances would yield similar (synthetic) results; nor do we know what imaging modalities (or other data) would act as a reliable input to a similar multidimensional modelling exercise. It is likely that each imaging modality will have its own limitations and characteristics. Many important acute modalities have limited resolution and anatomical delineation of lesions is much less clear than with DWI.

It should be further emphasised that there were no clinical outcomes used in the modelling exercise. Again, very broad conclusions are being drawn from an exercise that does not even include basic clinical evaluations that could be understood by patients or families, or clinicians.

I would regard the current study as no more than hypothesis-generating. The hypothesis that complex modelling provides more robust insight and predictive value than simpler modelling should be investigated, but a modelling exercise involving simulations based on a highly selected dataset subjected to complex image analysis is not a basis for generalised conclusions.

In summary I feel that the present analysis has merit in demonstrating an approach to complex modelling of MRI-defined brain anatomical lesions that might yield better predictive value for individual cases in whom imaging is available compared with simpler modelling, but that the requirements for appropriate imaging inputs to the model have not been explored with respect to modality or timing, no clinical outcomes have been assessed, and the complex model has not been compared with the predictive value of simpler models based on conventionally available variables. Replication of the findings in another dataset and exploration of the robustness of this approach with respect to other imaging modalities or timings is

necessary. The present approach could be evaluated in the context of a phase 2 clinical trial that uses clinical outcomes, and compared against current best standards. The approach is more likely suited to a subacute population and therefore to rehabilitation or regenerative therapy strategies and more work is required to investigate potential applicability to the acute environment.

“Our results indicate that complex modelling with richly represented lesion data is critical to individualized prescriptive inference in ischaemic stroke.” – I would suggest “may offer an approach to improve upon” rather than “critical.”

Discussion p17 “this question... requires semi-synthetic simulations,” – I think the question of whether there are subgroups for whom a treatment might be effective can be explored with simulations, but this is only one approach. There are many examples of subgroup analyses (the great majority of which have been misleading) in the conventional trial world, and whether the simulation approach brings any useful insights is at best a testable hypothesis, but would still require confirmation in a conventional RCT.

(Remarks on code availability)

Version 2:

Reviewer comments:

Reviewer #4

(Remarks to the Author)

Based largely on the rebuttal text, my understanding is that the authors position includes the following:

- Their existing data are sufficiently robust, and no amount of additional data would, or indeed could, mathematically, modify their conclusions. Concerns that bias of MRI acquisition to younger, fitter patients; the supposed stability of lesions over time (not what is observed clinically in the acute stroke setting); and reliance on single centre data; are essentially dismissed.
- The conclusions cannot be validated or tested since clinical trials are inherently unable to identify true treatment effects
- All measures of clinical outcome are unreliable and should not be considered necessary to model
- Any imaging modality that yields structural imaging data can be used to derive models robustly and without being tested (despite inherently lower resolution, poorer discrimination of acute from chronic changes, etc)

The text fundamentally starts from the position that treatment effects of interventions in stroke cannot be considered reliable because stroke is heterogeneous. That the clinically-defined diagnosis of stroke reflects a wide range of underlying pathologies is not at issue: the apparently dogmatic approach of the authors however appears to disregard more than 30 years of substantial progress in treatment that has been successful largely because, in recognition of this heterogeneity, clinical trials selected participants based on more precise phenotypic characterisation of patients (much of it based on imaging).

The modelling approach uses only brain imaging data, with no other clinical information, and the model outcomes are based on anatomical correlates of imaging. Under these specific conditions the conclusion is that a complex model of anatomy outperforms a simple randomisation scheme: in itself this is unsurprising, although it is reassuring to see this hypothesis confirmed in a large dataset. Qualitatively it reaches the same conclusion as previous, much simpler, critiques of clinical trial design in stroke that have addressed the issue of heterogeneity and potential use of imaging to more precisely phenotype trial participants.

The extrapolation by the authors is very far-reaching. The text barely acknowledges the very considerable practical barriers to obtaining complex data in an emergency setting, assumes that inherently different imaging modalities will be readily adapted to the model scheme, and dismisses the limited correspondence of clinically meaningful outcomes such as disability to brain anatomy. There is no clear practical suggestion of how complex modelling might be reconciled with clinical trials, or consider whether there are constraints on how it might be applied. Indeed the text continues to express the view that observational data can be considered superior to randomised trials.

These are very big, generalised claims from a computer modelling exercise. Some recognition of the limitations and much more cautious conclusions would, in my view, remain appropriate.

(Remarks on code availability)

Reviewer #1 (Remarks to the Author):

This is an interesting project utilizing thorough design and data analysis, with the aim to compare the individual-level fidelity of the traditional interventional randomized allocation system versus a more flexible modelling of non-randomized data based on deep representation learning. The investigators make use of big data (using high-resolution maps of 4,119 acute ischaemic lesions), including anatomical patterns of functional activation, neurotransmitter receptors, gene expression, and other neural properties of spatial structure to evaluate treatment susceptibility. In the era of personalized medicine, this work is more relevant than ever.

Thank you for these appreciative remarks.

1. The introduction is rather long. The authors should be able to address the main question of their work in less than 2 pages. More relevant information may be presented in the Discussion section.

Thank you for this suggestion. We appreciate the value of brevity, and have tried to be as succinct as possible in our revision, but the complexity, multi-disciplinarity, and relative novelty of the material demand a gentler introduction than is usual if intelligibility to a wide readership is to be achieved. We also require some space to provide clarifications requested by the other reviewers.

2. I would suggest toning down a bit the statements relevant to RCTs, such as “Not only may we not assume that conventional RCTs will converge on the correct inference here, there are strong reasons for doubting the fundamental grounds on which they rest”.

Thank you for this suggestion: we have carefully revised our criticisms of RCTs to clarify their precise nature and scope (lines 64–67: “Where available, observations of responses to unconfounded treatment allocation—such as from a traditional RCT—may be combined with established low-²⁶ or novel high-dimensional¹⁹ multivariable modelling of conditional outcomes^{4–6}. Randomization remains a powerful strategy, the best available means of protection from hidden confounding.”; lines 75–76: “In an ideal world, randomization would be combined with highly expressive models⁶: the question is where the compromise should lie when real-world constraints place the two in opposition.”; lines 371–374: “Where real-world feasibility permits it, complex modelling is ideally combined with randomization; equally, an effect drawn from observational data may be subsequently evaluated with randomized, prospectively acquired data from plausibly homogeneous subpopulations. Both approaches, including their combination, need to be considered in the light of real-world constraints”).

Reviewer #2 (Remarks to the Author):

Giles et al. report the results of performing a comprehensive study of ascertaining therapeutic choices for ischemic stroke patients using a combination of state-of-the-art computational and statistical techniques, paired with a wide variety of data sources, including both high-resolution maps of patient acute ischemic lesions and meta-analytic information on connective, genetic, and functional expression. The advanced methodological and computational aspects of the work are justified by the aim of overcoming the limitations of methods that are considered as "the standard" in health/medical sciences research, and which have been demonstrated to be incapable of making use of expansive datasets and statistical tools now relatively widely available. The manuscript is, on the whole, well written, and it has the potential to be an important contribution in applied medical sciences research, as well as a notable contribution in applied statistics and/or machine learning. There are several points that the authors should consider improving, both in the presentation of the manuscript and in some of the simulation experiments performed, in order for the manuscript to have a broader impact and accessibility. Some specific comments follow.

Thank you for this summary of our work and appreciative remarks on its potential value. We are grateful for your detailed comments on presentational and experimental aspects, and are glad to address them to improve broader impact and accessibility.

** Since the main-text treatment of the causal inference ideas avoids the use of notation, some phrases can be confusing, e.g., "likely success of an intervention" (sentence 1), which could reference any of $E[Y(1) - Y(0)]$, the average treatment effect; $Y(1) - Y(0)$, an individual treatment effect; or $P\{E[Y(1) - Y(0)] > 0\}$ and/or $P\{Y(1) - Y(0) > 0\}$, which are probabilities of counterfactual quantities. Some use of notation, or greater care with language, would improve the presentation.*

Thank you for this point. Introducing these ideas precisely without notation—which we feel to be necessary to reach the widest possible audience—is challenging, and we have now carefully revised the main text to remove any residual ambiguities (e.g. line 16: "estimating the efficacy of an intervention", instead of "likely success of an intervention"; lines 57–60: "Within such individualized prescriptive inference, the underlying statistical model seeks to determine not the ATE across the population, but the conditional average treatment effect (CATE) informed by a wide array of the patient's distinctive features, resulting in estimates that are much more personalized"). In critical places of the main text, we also refer the reader to the methodological sections (lines 217, 232, 244), where formal exposition is presented, and which we have revised throughout for maximum precision, clarity and completeness.

** In discussing the possible over-reliance on RCTs, the authors note that the "established inferential approach...assumes our knowledge of individuals is limited to the average of the population..." (sentence 2), without making clear what the "established inferential approach" is; this is confusing since it is less a criticism of RCTs, given that randomization is enough to identify the *full distribution* of potential outcomes, than it is of too much focus being placed on simple estimands, like the average treatment effect, without thought being spared for whether the estimand actually aligns with the underlying science or policy questions that motivate the research.*

Thank you for this recommendation: we have adjusted our wording to clarify what we mean by the established inferential approach (lines 17–20: “*The established inferential approach, enshrined in the randomized controlled trial (RCT), seeks to estimate the average treatment effect (ATE) across populations described simply enough to permit modelling with standard statistical methods under randomized treatment allocation policy¹.*”), and emphasize that the fundamental problem arises not from randomization itself, but from the widespread use of simple estimands, over-reductive descriptions of the population, and inflexible statistical models (lines 26–29: “*heterogeneity is not detectable within the prespecified trial evaluation itself, nor necessarily in any subsequent subgroup analysis based on the same trial specification: richer descriptions, more flexible models, and less reductive estimands are required^{3–6}*”). We also draw attention to the practical obstacles of conducting RCTs where, as in the case of focal brain injury, the heterogeneity is so high-dimensional, adequate stratification would likely require infeasibly large samples (lines 67–69: “*But in the setting of unknown heterogeneity defined by many interacting factors, the necessary data scale becomes infeasibly large, forcing reliance on routine clinical data streams with limited control over randomization.*”; lines 173–175: “*A small sample biased by selective recruitment would inevitably lead to under-estimates of observed heterogeneity and our capacity to model it*”).

* *In the first paragraph, the authors make it an important point, related to the above bullet, that the ATE may simply not be an interesting contrast; however, they then mistakenly claim that conditional ATEs (CATEs) cannot be learned from RCTs (“heterogeneity is not detectable within the trial evaluation itself, nor necessarily in any subsequent subgroup analysis based on the same description of the population”). This is not true, as has been detailed in the causal inference and (bio)statistics literature, e.g., VanderWeele et al. (2019), “Selecting optimal subgroups for treatment using many covariates” (Epidemiology); Luedtke & van der Laan (2017), “Evaluating the impact of treating the optimal subgroup” (Statistical Methods in Medical Research); Kessler et al. (2019), “Machine learning methods for developing precision treatment rules with observational data” (Behaviour Research and Therapy); etc. While there are many approaches to learning heterogeneous treatment effects, and the authors seem to use the “X-learner” framework, all such frameworks, including advances such as the “DR-learner”, ought to be compatible with randomized trials (RCTs, A/B tests).*

Thank you for this comment, and apologies for the unclarity of our statement, now revised to make the point absolutely clear (lines 26–29: “*heterogeneity is not detectable within the prespecified trial evaluation itself, nor necessarily in any subsequent subgroup analysis based on the same trial specification: richer descriptions, more flexible models, and less reductive estimands are required^{3–6}*”). There is, of course, no reason randomization would make estimation of conditional ATEs impossible, indeed it could only help it (lines 64–67: “*Where available, observations of responses to unconfounded treatment allocation—such as from a traditional RCT—may be combined with established low-²⁶ or novel high-dimensional¹⁹ multivariable modelling of conditional outcomes^{4–6}. Randomization remains a powerful strategy, the best available means of protection from hidden confounding.*”). Our point is that within the usual statistical framework of RCTs—highly reductive patient descriptions modelled with simple, inflexible architectures—trial failure caused by inadequately captured treatment response heterogeneity is indistinguishable from that caused by a lack of any therapeutic effect. Reanalysing the same data—within the same statistical framework but with CATE as the estimand—need not be sufficient, for the problem may arise from inadequate expressivity of the input and/or model. Critically, by far the commonest outcome from a failed RCT is a conclusion of no therapeutic effect, not inadequate handling of heterogeneity, and the resultant

policy is typically to stop development, not to rerun the trial with a better description of the patient and/or a more flexible model. Where heterogeneity is very high-dimensional, the scale of the data necessary for estimating CATEs accurately may in any event be infeasible outside the observational setting (lines 75–76: *“In an ideal world, randomization would be combined with highly expressive models⁶: the question is where the compromise should lie when real-world constraints place the two in opposition.”*; lines 371–374: *“Where real-world feasibility permits it, complex modelling is ideally combined with randomization; equally, an effect drawn from observational data may be subsequently evaluated with randomized, prospectively acquired data from plausibly homogeneous subpopulations. Both approaches, including their combination, need to be considered in the light of real-world constraints”*).

** The authors make a critical point for precision/personalized medicine, that “Inferences about individuals may now be informed by the local subpopulation to which they belong”, yet then go on to conflate individual treatment effects (ITEs) $Y_{i(1)} - Y_{i(0)}$ with conditional average treatment effects (CATEs) $E[Y(1) - Y(0) | V = v]$ for some subgroup $V = v$. Since ITEs are never identified due to the fundamental theorem of causal inference, the focus on individualized effects (or ITEs) draws attention away from the more practical goal of pursuing inference about CATEs.*

Thank you for this point. The difficulty here is that we wish to distinguish between conditioning on a feature *simpliciter*, and conditioning on a sufficient array of features (and with sufficient flexibility) to approximate the counterfactual outcome of the individual as closely as the data allows. Conditioning on a single, simple feature such as (say) age does not plausibly provide much scope for individualization; conditioning on a high-dimensional set of features by contrast does, given sufficient data and the right model architecture. We have thoroughly revised our use of individualized. As recommended, we have modified the text to clarify that the CATE is the focus of our inference (lines 57–60: *“Within such individualized prescriptive inference, the underlying statistical model seeks to determine not the ATE across the population, but the conditional average treatment effect (CATE) informed by a wide array of the patient’s distinctive features, resulting in estimates that are much more personalized”*), and included slightly more formal exposition in the Methods (lines 658–691).

** Throughout, the manuscript refers to “unbiased treatment allocation”, which is meant to be synonymous with *unconfounded* treatment allocation; the latter terminology is more standard, and it may be useful to avoid the term “unbiased”, as it risks conflating randomization, which yields unconfoundedness in terms of counterfactuals, with the concept of “balance”, which is the empirically testable analog that may be viewed as “bias”.*

Thank you for this excellent suggestion: we have replaced *unbiased* with *unconfounded* throughout. We originally chose to use *unbiased* for clearer relation to treatment allocation bias, but accept that *unconfounded* would be less ambiguous in this context.

** In comparing standard practice in evaluating RCTs with the proposed tact of flexible modeling (i.e., “the comparative importance of minimizing treatment allocation bias vs maximizing model expressivity and flexibility in inferring the optimal treatment for each individual patient”), the tension appears to be misplaced, as there is no apparent incompatibility between employing flexible*

*modeling strategies (i.e., for an outcome regression) *within* an RCT, except for dogma about how inference ought to be performed (e.g., difference-in-means or parametric g-computation). The authors message would likely be better received if they were to note that the proposed strategy could actually be employed in RCTs with large enough sample size, as has recently been done outside of the literature on neurology/neuroimaging.*

Many thanks for your valuable comments: we have adapted the text to acknowledge the compatibility between our methods and randomized trials (lines 64–67: *“observations of responses to unconfounded treatment allocation—such as from a traditional RCT—may be combined with established low-²⁶ or novel high-dimensional¹⁹ multivariable modelling of conditional outcomes⁴⁻⁶. Randomization remains a powerful strategy, the best available means of protection from hidden confounding”*; lines 75–76: *“In an ideal world, randomization would be combined with highly expressive models⁶: the question is where the compromise should lie when real-world constraints place the two in opposition.”*). We further emphasize that practical feasibility is a critical constraint here: where a RCT of the size needed to support the model flexibility the task demands is impossible in practice, inference from observational data may be the only realistic solution (lines 67–69: *“But in the setting of unknown heterogeneity defined by many interacting factors, the necessary data scale becomes infeasibly large, forcing reliance on routine clinical data streams with limited control over randomization.”*).

** Some of the authors' statements are too far-reaching, even if well-intentioned, e.g., "This comparison *can only be performed*", or "to use data within the RCT framework is to beg the question of what the right standard should be: our task here"; moreover, the latter of these is not even true, as data from an RCT could certainly be used to learn about treatment effect heterogeneity (as noted above), as long as key baseline covariates are collected as part of the trial. The false dichotomy of casting principled/flexible modeling approaches against randomized studies does a disservice to both fronts of scientific inquiry, i.e., surely the way forward is not to set aside RCTs but to learn to also incorporate flexible modeling and observational studies as part of the "scientific corpus".*

Thank you for this comment. We have revised these statements to clarify that the target of our criticism is not randomization itself, but the widespread belief—at least in the clinical domain—that it is a panacea for a wide variety of ills, many of which—such as including *all* relevant baseline covariates—are not remediable in practice within the logistical constraints of RCTs, at least where the target system is as complex as the neurological (lines 17–20: *“established inferential approach, enshrined in the randomized controlled trial (RCT), seeks to estimate the average treatment effect (ATE) across populations described simply enough to permit modelling with standard statistical methods under randomized treatment allocation policy¹”*; line 38: *“RCTs, with ATE as the outcome measure, have been placed on the highest pedestal”*; lines 67–69: *“But in the setting of unknown heterogeneity defined by many interacting factors, the necessary data scale becomes infeasibly large, forcing reliance on routine clinical data streams with limited control over randomization.”*). It is widely held, again at least in the clinical domain, that RCTs set the “gold standard”, and that any conclusions *not* derived from RCTs are to be viewed with scepticism. We now set out clearly how we believe inference from these two settings may be best conducted and interpreted in optimally guiding patient management: the ultimate aim here (lines 371–374: *“Where real-world feasibility permits it, complex modelling is ideally combined with randomization; equally, an effect drawn from observational data may be subsequently evaluated with randomized, prospectively acquired data from plausibly*

homogeneous subpopulations. Both approaches, including their combination, need to be considered in the light of real-world constraints.”).

** In the Methods section, standard assumptions for identification of causal effects (consistency, positivity, no unmeasured confounding) are presented, but it would help to give an example of an identification result (e.g., for the ATE) while also connecting this to the treatment effect heterogeneity parameter that is of interest, i.e., the CATE. Notably, the simulation studies include a "bias parameter" for treatment allocation, which can be set to 1 (deterministic treatment based on patient phenotype); this is only mentioned in the main text, without noting the underlying issue of resulting in non-identification of causal parameters, by violating the positivity assumption. While this fact is acknowledged in the methods, it may escape notice of some of the readership and is not an interesting case anyway, so it may be better to simply truncate the simulations at a bias parameter of 0.95, where one may encounter positivity violations in the data but not in the underlying data-generating mechanism, since, in the latter case, no causal inference is possible at all.*

Thank you for this comment. We now include an identifiability result for the CATE in the methods, with a short derivation (lines 663–677). We have also made this section slightly more formal and included more mathematical notation. The violation of the positivity assumption when setting the “bias parameter” (now “confounding hyperparameter”) to 1 was intentional: we wanted to assess the implications for treatment selection where this violation occurs. We have now explicitly described in the main text that we model it under the weakening of the assumption to the point where it is violated (lines 231–232: *“we observe the effect on prescriptive inference as the assumptions underlying causal inference (see Methods section Prescriptive inference) are stretched to breaking point⁴²”*; lines 668–669: *“When the hyperparameter that controls confoundedness is 1, allocation becomes deterministic, and we observe the effect of invalidating positivity.”*). Also, we have the experimental data, so we felt we should include it.

** The simulation studies appear well-constructed and thorough; however, some of the design decisions underlying them are left unexplained. For example, the authors note that “treatment allocation [was varied] from random (no bias) to directly determined by the location of the centroid of each lesion (high bias)”, which seems to capture different scientific contexts, e.g., what would be the reason to pursue random treatment assignment across lesions types regardless of location? Presumably, this is not how medical practice actually works, as one would only randomize treatment strategies if a given strategy is completely novel and, even then, only for a subset of lesion locations/types and against standard-of-care. My suggestion would be to clarify the decision-making behind some of the simulation parameters and to place these in the appropriate context of a medical or computational study.*

Thank you for this comment: we have now clarified this aspect of the simulations (lines 385–391: *“Equally, though we may use a highly compressed representation—a two-dimensional embedding of lesion anatomy, for example—to stratify patients upstream of randomization within an otherwise conventional RCT, the loss of fidelity in the compression must be balanced against the gain in enabling randomization. Representation learning dependent on large-scale data would be required anyway, for the quality of the embedding is likely to be at least as dependent on data scale as a discriminative model of comparable flexibility. It is highly unlikely that overt—rather than latent—low-dimensional descriptors can commonly be found, for no (say) blood-borne signal could plausibly capture anatomically distributed information.”*; lines 578–583: *“If this measure of overlap exceeded*

5% of the volume of each subnetwork, the lesion was designated as responsive to the corresponding treatment. This threshold is within the range of critical damage thresholds considered by Mah et al¹⁸. In this way, we generate the data for the virtual trials from the 16 functional networks, with both lesion masks or disconnectomes, and responsiveness informed by meta-analytic neurotransmitter receptor or genetic transcription distributions.”; lines 629–645; lines 663–667: “We experiment with values of the TE, RE and treatment–outcome confounding that stretch these assumptions to breaking point. The experiments involve simulated noise levels that exist in real healthcare data⁷³. Simulating observable confounding illuminates the effect of stressing conditional exchangeability, while simulating unobservable confounding illuminates the effect of its violation.”). Ischaemic stroke is typically managed indifferently to the detailed anatomical pattern of damage, i.e. anatomical variations will be random with respect to treatment. In certain circumstances—domain-specific rehabilitation, for example—patients may, however, be phenotyped by their functional deficit. To account for this scenario, we define subgroups by predicted functional deficit, and randomize only within the patterns of damage that plausibly produce the specific deficit, mirroring clinical practice. Therefore, even when selected at random, both treatments are modelled to be reasonable for the observed stroke presentation, and it is the task of the prescriptive inferential system to be able to disentangle the patterns of heterogeneity to be able to infer the treatment, $w \in \{A, B\}$, with the greatest probability of causing a favourable outcome, $\mathbb{P}(Y = 1 \mid W = w, \mathbf{X} = \mathbf{x})$.

** I found the conclusion of the Results section, which states "rejecting the null hypothesis of no superiority of richly expressive representations under biased treatment allocation conditions vs low-dimensional models under randomization", to provide a helpful, clear framing of the goal of the study, which I would suggest be made as clear earlier. To this point, it is an interesting, practically useful computational study, but the exact goal is vaguely stated as comparing flexible modeling strategies with standard inferential practice in RCTs. Related to this, the authors make several strong statements in this section that ought to be toned down, including both "[a] high-dimensional representation, even in the setting of strongly biased allocation, was almost always at least non-inferior to a RCT employing low-dimensional models, and in the vast majority substantially superior" and "higher prescriptive fidelity even from a markedly biased and noisy trial modelled with a richly expressive representation than from a fully randomized trial at the same noise level modelled with a simple one", the latter of which may simply be over-interpretation/generalization of some technical artifacts of the authors' computational study. From theoretical arguments, one might expect a too-simple model to fail, even in an RCT, when the model cannot capture the complexity underlying data well enough; however, simply using a more complex model in an RCT should improve this, as the RCT will protect against confounding and more flexible modeling strategies may succeed in picking up smaller treatment effects...*

Thank you, we have clarified the hypothesis of the study earlier in the manuscript (lines 80–88: “Our analysis tests the hypothesis that, under real-world constraints, the fidelity of inferred individual prescriptions depends less on unconfounded treatment allocation policy than on capturing treatment response heterogeneity with expressive generative models combined with flexible causal models. Our null hypothesis concerns CATE-based inference in the context of varying degrees of treatment allocation, and representations that vary from narrowly-defined (effectively ATE-based inference), to richly expressive: complex representations plus confounded allocation is not superior to simple representations plus random allocation. On the rejection or otherwise of the hypothesis rests the optimal inferential policy for evaluating treatments in acute stroke sensitive to the functional

organization of the brain.”). We absolutely agree and have acknowledged the compatibility between RCTs and flexible modelling (lines 64–67: “observations of responses to unconfounded treatment allocation—such as from a traditional RCT—may be combined with established low-²⁶ or novel high-dimensional¹⁹ multivariable modelling of conditional outcomes⁴⁻⁶. Randomization remains a powerful strategy, the best available means of protection from hidden confounding”; lines 75–76: “In an ideal world, randomization would be combined with highly expressive models⁶: the question is where the compromise should lie when real-world constraints place the two in opposition.”), while also maintaining the practical benefits of using observational data in the accessibility of data of the required scale for inference under such high-dimensional, heterogeneous context (lines 67–69: “But in the setting of unknown heterogeneity defined by many interacting factors, the necessary data scale becomes infeasibly large, forcing reliance on routine clinical data streams with limited control over randomization.”). One point to emphasize is that we have taken great care to maximize the generalisability of the computational study, employing internationally amongst the largest collections of unselected, anatomically registered and segmented imaging of ischaemic stroke, deriving patterns of plausible deficits and treatment responsiveness from large-scale meta-analytic functional imaging data, and employing a wide range of representational and inferential model architectures.

** Related to the above, there is some overly strong language used in the Discussion section as well, e.g., “there are strong reasons for doubting the fundamental grounds on which [RCTs] rest” (whereas, in reality, RCTs are simply not a panacea and, indeed, can be poorly performed -- so, more flexible modeling and openness to the use of observational studies would help to improve evidence-generation in modern medicine). On the other hand, some points -- like “A failure of inference here cannot be distinguished from a failure of the intervention, and if the inferential approach is never questioned, replication will not correct the error but typically entrench it.” -- are very well put!*

Thank you for these comments. We have adjusted our language to acknowledge the compatibility for flexible modelling with randomization in response to the former point.

** The Methods section is also well-written, though some aspects could be improved, including a relative paucity of notation, some of which could go a long way to improving clarity (see a previous suggestion about working out an identification result to ground the arguments for pursuing inference about the CATE). In this section, some statements could be made more precise, e.g., “treatment-related confounding” (page 23, line 619) would be better stated as “treatment-outcome confounding”. In this section, τ_i is said to be the individual treatment effect and $\hat{\tau}$ is then said to be an approximation of this, yet this is not, generally speaking, true. $\hat{\tau}$, which appears to be a conditional average treatment effect, approximates τ_i only as the support set $X = x$ grows to approximate all features that define “individuals” in the population, e.g., $\dim(X) \rightarrow \infty$. It might be better to rely on standard results to simply note that $\hat{\tau}$ is an identifiable causal parameter whose interpretation is closest to what might be desired in the context of personalized/precision medicine. Also, $\hat{\tau}$, if it is actually meant to be the CATE, can be better expressed as $\hat{\tau} = E[P(Y = 1 | W = A, X) - P(Y = 1 | W = B, X) | X = x]$, where the $X = x$ could also be written as $V = v$ for some V that is a subset of X , allowing for effect heterogeneity to be learned in subgroups that are not necessarily concordant with the full set of putative measured confounders X .*

Thank you – we have adapted treatment-related confounding to treatment–outcome confounding throughout. As described, we avoided notation in the main text to make it accessible to the widest possible audience. We have carefully revised the notation in the Methods, indeed we have revised most of this section with a focus on clarity and mathematical rigour. We have revised any arguments involving ITEs to focus instead on the CATE, in line with your helpful suggestions (lines 57–60: “*Within such individualized prescriptive inference, the underlying statistical model seeks to determine not the ATE across the population, but the conditional average treatment effect (CATE) informed by a wide array of the patient’s distinctive features, resulting in estimates that are much more personalized*”; lines 658–691).

** In describing the "one-model" and "two-model" approaches to inference on heterogeneous treatment effects, it would be helpful to describe these in notation and to specifically refer to algorithmic frameworks that are employed. For example, the authors specifically cite distinct works by Kunzel et al. (2019), Sontag, and van de Schaar, but how these are exactly used is not elaborated upon. Some description of the techniques, e.g., X-learner, would help to clarify how these ideas are built upon or adapted in this work.*

Thank you, we have done this (lines 693–700: “*We infer treatment effects using two frameworks from the causal inference literature^{29,31,75–77}. The one-model/SLearner⁷⁵ framework involves modelling $\mathbb{P}(Y = 1 | W = w, \mathbf{X} = \mathbf{x})$ using a single outcome classifier, fitted to pairs (\mathbf{x}, δ_w^A) , where δ_w^A is the Kronecker delta, which equals 1 when $w = A$ and 0 otherwise. The optimal treatment is then inferred to be the treatment with the greatest corresponding probability. The two-model/TLearner⁷⁵ approach employs two separate outcome-predicting classifiers. The first classifier is fitted to the participants allocated to treatment A; the second to treatment B. For each unseen individual and each treatment, the probability of causing a favourable outcome is computed from the corresponding model. The optimal treatment is then inferred to be the treatment with the greatest corresponding probability.*”).

** Throughout, the authors note the use of the Benjamini-Hochberg (BH) correction, which controls the false discovery rate (FDR), for which the standard justification of tolerating some false discoveries is given. I would question the relevance of the FDR in this setting, as one would presumably prefer to err on the side of failing to discover personalized treatment strategies rather than take the more liberal view of discovering some personalized treatment strategies that may, in fact, be mistakes. Also, the authors mistakenly note that the BH correction controls the family-wise error rate (FWER) [page 27, line 781-782].*

Thank you for this comment. It seems to us that both implementing and not implementing a given strategy may result in poorer outcomes, for the choice here is not between action and inaction, but different strategies for positive action. Apologies for the erroneous reference to FWER: now corrected (line 735: “*The false discovery rate reflected 0.05 significance level*”).

** Rather than using "individualized" throughout, I might recommend that the authors consider using "personalized" or "precision", which are both more vague but carry the advantage of not directly referring to "individualized treatment effects", which are not identifiable, generally speaking.*

Thank you for this suggestion, the agreed vagueness of “personalized” and “precision” makes us reluctant to use them. We no longer refer to individualized treatment effects, having refocused on the CATE, and so we hope that there is no longer any risk of confusing individualized prescription with individualized treatment effects. Having said that, for clarity we have also added a statement to the *Main text* that contrasts individualized prescriptive inference, the CATE, increasingly detailed conditioning data and estimates that are tailored to the individual, where the latter is referred to as “personalized” (lines 57–60: “*Within such individualized prescriptive inference, the underlying statistical model seeks to determine not the ATE across the population, but the conditional average treatment effect (CATE) informed by a wide array of the patient’s distinctive features, resulting in estimates that are much more personalized*”). We have also revised each use of the word ‘individual/ized’ throughout to address any ambiguity.

** The writing makes use of many specialized terms, including those that are not even standard jargon, e.g., “prescriptive inference”, “therapeutic inference” (these from the abstract), etc. Since the research lies at an interface between statistics / causal inference, machine learning, and medical research, such terminology can be particularly confusing; for example, after years of teaching (and learning) causal inference, I’ve never run into such terms as the above though can figure out what they may refer to, e.g., dynamic/personalized treatment regimes or similar. Given the broad readership that ought to be the target, I expect that many readers will be confused at one point or another. Perhaps the authors could attempt to resolve some of this confusion by providing a glossary at the end of the main text that defines such terms as the above (and others like them) in terms of what might be more easily understood by a statistician, medical researcher, etc. Other such terms include, e.g., “semi-synthetic counterfactuals”, by which the authors presumably mean regression quantities like $\hat{E}[Y(1)]$ or similar.*

Thank you for this suggestion. We have carefully revised the manuscript to ensure all terms are clearly defined at the point of introduction and now include a comprehensive glossary (Supplementary S18). We are compelled to introduce a few unfamiliar terms to describe novel characteristics of our approach, such as a semi-synthetic framework for providing ground truths for the simulations.

** Typo: “...for based on...” (page 9, line 222)*

Resolved, thank you (lines 228–230: “*Similarly, to generate treatment–outcome confounds, we vary the degree to which treatment allocation policy depends on lesion location, from not at all (random/unconfounded), to fully deterministic (non-random/confounded).*”).

Reviewer #3 (Remarks to the Author):

This paper discusses the importance of treatment effects' heterogeneity in reasoning about the efficacy of treatments, particularly in the context of acute stroke lesions. The paper seems to make two key arguments:

(1) RCTs with inadequate accounting of heterogeneity may be missing out on treatments that can benefit subsets of the population

(2) Observational data may be a better source for understanding treatment heterogeneity, and the observed and unobserved confounding biases in this data may have less impact on the optimal treatment rules than not accounting for heterogeneity.

While points (1) and (2) are obviously not new, the authors propose a form of sensitivity analysis to compare the relative importance of richer covariates in observational data in identifying the optimal treatment rules vs. the effect of biases in identifying wrong rules. They do so using semi-synthetic simulations of counterfactual outcomes based on a causal model that is informed by biological plausible mechanisms in the context of stroke lesion data.

Overall, I had a hard time reading this paper and understanding the authors' contributions. The writing is extremely verbose and convoluted to the extent that I think the paper should be completely rewritten to improve clarity and conciseness if accepted.

Thank you for this summary. In the realm of stroke, indeed focal brain injury in general, treatment selection is at present almost exclusively informed by RCTs employing simple estimands, highly reductive descriptions of patients, and inflexible statistical models. Specifically, brain anatomy is either ignored or modelled far more crudely than our existing knowledge of its functional architecture could reasonably justify.

Our task is to evaluate, as quantitatively as we can, the consequences of this policy in a way that is intelligible and persuasive to the wide audience on which any change to the policy depends. This task requires not merely applying well-established principles of causal modelling of heterogeneous treatment effects with observational data, but also innovating in the representation learning of 3D lesion data, and establishing a framework for quantifying the impact of real-world lesion and neural substrate structure on the estimability of conditional average treatment effects under realistic RCT and observational regimes.

Without such a biologically informed framework, designed to cover the widest possible range of treatment and lesioned substrate parameters, it is impossible to know how important and urgent a change in interventional modelling policy in this domain might be. As far as we are aware, no currently registered Phase III trial in stroke, indeed in any other form of focal brain injury, departs from the standard, simple RCT paradigm, and nothing in the neurological literature compels anyone to consider alternatives to the standard practice, despite its well-established theoretical shortcomings. Theory alone clearly does not provide sufficient motivation for reconsidering the inferential framework here.

Now were focal brain injury a rare disease, our analysis would be of niche interest. But stroke alone is globally the second biggest killer and largest cause of adult neurological disability internationally

(Vos et al., 2020), so even modest improvements in its management have substantial, global significance. Demonstrating that a change in inferential approach—a comparatively easy, low-cost intervention—could make a major difference to outcomes here therefore seems to us an important contribution.

In addition I have the following major technical comments:

1- The authors build a semi-synthetic model to simulate the potential outcomes of each subject in the study and use that model to conduct a "virtual" interventional trial using rich observational data. The key premise of their proposed method is that this model can be used to assess the relative importance of randomization vs. rich covariate adjustment using modalities only available in observational data. However, I am a bit confused about the logic behind this approach. Wouldn't the validity of your inferences still be limited by the correctness of the synthetic model which is itself going to be affected by confounding biases? That is, you would still need to randomize the intervention in a new trial that only includes patients hypothesized to benefit from the treatment based on your simulation. But isn't this the goal of any observational study for which we can use any ML method for estimating conditional average effects without running a simulation?

Thank you for this comment. There are two elements to the answer:

First, the fundamental problem here is the absence of a ground truth: neither ATEs nor CATEs can be explicitly validated, either in an RCT or an observational study, for counterfactual outcomes are definitionally unobserved (Alaa and van der Schaar, 2018; Hernán and Robins, 2020; Holland, 1986; Imbens and Rubin, 2015; Pearl, 2009; Shalit et al., 2017). The credibility of the inference therefore rests on the integrity of the statistical assumptions and the coherence of the mathematical model. But that does not mean the argument becomes purely theoretical, for a change in inferential strategy has a cost: far larger data scales, with no reliable means of calculating power from small samples, and model architectures whose flexibility makes them challenging to handle. Crucially, if lesion–outcome relations exhibit no learnable complex structure, for example if outcomes are no more predictable from (say) lesion volume than from an expressive lesion representation, then there will be no benefit from conditioning on richly described lesion anatomy, and no reason to deviate from current statistical practice. Indeed, this is what the field—at least of stroke—assumes to be true. Note that simply showing that the intrinsic dimensionality of lesions is insufficient, for the variation may have no bearing on treatment outcomes. This is why we need semi-synthetic simulations that span the widest range of possible lesion morphologies and patterns of neural dependencies of the outcome, and seek to directly quantify observed differences under the two modelling regimes.

Second, the objective of the modelling exercise is not to identify a subset of patients that are hypothesized to benefit from a given treatment, but to quantify the difference in the fidelity of the inferred prescription, under the two modelling regimes, over the widest possible range of potential lesions and patterns of neural dependence. The differences we observe are generally not specific to any deficit or pattern of neural dependence, but span the entire sampled domain. It is, of course, conceivable that there are lesion–outcome relations for which our observations do not hold, but if the vast majority of scenarios are in agreement, it seems unlikely that there will be enough exceptions for the overall conclusion to be undermined. Note that the simulation exercise here is not only of the widest practicable breadth, it makes assumptions widely agreed to be plausible—that deficits and

outcomes depend on structural and functional anatomy—and examines the full range of possible treatment effect and spontaneous recovery magnitudes. The grounds for generalizability here are therefore greater than in any other study of this kind we are aware of.

2- What is "treatment susceptibility"? Does this mean a propensity score/treatment assignment probability? This is not a standard terminology as far as I am concerned.

Thank you for the opportunity to clarify this key point. By treatment susceptibility we mean the susceptibility of an individual to respond to a given treatment, which is what the prescriptive model seeks to infer. To avoid confusion, we have changed this term to 'treatment responsiveness', which we believe is more self-explanatory. The necessity for this concept arises here because it is a feature of the independent semi-synthetic data generating process, which allows us to set counterfactual ground truths. We have now included a definition of this in the glossary within the supplementary material (S18).

3- The key result of the paper is on page 13 where the authors reject the null hypothesis of "no superiority of richly expressive representations under biased treatment allocation conditions vs low-dimensional models under randomization". However, I am not sure I understand the grounds based on which the authors reject the null hypothesis. This seems to be based on ground-truth data from semi-synthetic simulations, but doesn't this result only tell us that "there exists a synthetic model of treatment outcomes where this null hypothesis is false"? How can this be the grounds for reasoning about actual treatment rules in a real-world scenario?

Thank you for this comment. Our results show that across the widest range of synthetic models practically evaluable, with amongst the largest collections of lesions and most comprehensive set of potential patterns of neural dependence, the null hypothesis of no superiority of richly expressive representations under confounded treatment allocation conditions vs low-dimensional models under randomization is rejected. Since in the absence of purely empirical ground truths, no other form of reasoning is possible, this seems to us the only way of informing treatment policies in the real world. Note that current policies are not supported by any kind of empirical validation: they simply assume—against all prior plausibility—that anatomical heterogeneity can be simply ignored.

For the avoidance of doubt (though we are sure that is not what the reviewer means), we are not proposing that a researcher should use a synthetic model of a real-world deficit to infer treatment effects in any specific scenario: our inference is about the preferable inferential regime in the domain of stroke where treatment responsiveness may be dependent on the anatomical characteristics of the lesion.

4- I think that the authors proposed idea (a sensitivity analysis of effect of randomization vs. richness of covariates) should be framed as a formal framework with theoretical conditions on the synthetic models for valid inferences. The authors propose a general direction for thinking about treatments heterogeneity but then propose one very specific synthetic model with empirical results with unclear utility. Some relevant literature on sensitivity analysis based on marginal sensitivity models for individual treatment effects also seems to be ignored, though these frameworks try to achieve a very similar objective.

In the Methods section, we have clarified the formal grounds for the inferences made in the virtual trials, and we have also included an identifiability result, with a short derivation (lines 658–691). We disagree that our synthetic model is “very specific”: on the contrary, we have explored a vast range of modelling conditions including lesion representations, physiological substrates, functional deficit presentations (across the whole brain’s grey matter), and across synthetic observational data conditions in noise and confoundedness to evaluate comprehensively the prescriptive fidelity of applying data with these methods. This is, to our knowledge, the most comprehensive modelling exercise in the domain of focal brain injury ever attempted, and it is rooted in assumptions about the underlying lesion–outcome relations implicit in a vast body of work in the neuroscientific domain.

I can't comment on the biological plausibility of the proposed model since it's beyond the scope of my expertise.

Thank you for your comments and suggestions.

Reviewer #4 (Remarks to the Author):

The authors describe a complex modelling and simulation exercise to explore how heterogeneity of brain lesions might influence hypothetical treatment responses in acute stroke, based on a model of functional connectivity and receptor distribution, and a large dataset of around 4000 acute ischaemic stroke lesions. They illustrate superior performance of their complex model compared to a simple classification system in predicting the effect of simulated interventions.

Thank you for this summary.

It is unclear what outcomes – if any – were included in the modelling exercise. It appears that no standard clinical measures of outcome were included and the modelling fidelity was assessed only against modelled effects on receptor maps, connectome maps or similar. This could be clarified.

Thank you for this suggestion: this crucial point is now clarified in the revision (lines 219–226: *“Having generated a ground truth, we conducted a series of virtual interventional trials, where each patient event—defined by empirical lesion anatomy, the corresponding simulated deficit, and simulated treatment responsiveness—was allocated to receive one of two treatments, differing in their individual-level effects as outlined above (Figure 3, lower panel). The task is to fit probabilistic models of individual outcomes to the trial data, from which we can then infer individualized prescriptions. Unlike a real trial, however, here we can quantify the fidelity of the inference against the ground truth, where the outcome for each individual, given each treatment, is known. This enables us to compare the performance of different models and experimental settings objectively²⁷.”*; lines 418–424: *“Though stroke is caused by anatomically organised damage to the brain, its outcomes are not exclusively determined by anatomy. A wide array of additional factors contribute to treatment outcome heterogeneity, placing varying demands on the expressivity of the models needed to capture them. The same approach can, and should, be extended beyond the anatomical domain. How readily any factor may be captured will, of course, vary, and there will inevitably be residual variation inaccessible to any practicable model. But the question of whether a factor is material to outcomes, and modellable within a given regime, requires investigation within the kind of framework demonstrated here.”*). The outcomes were simulated by the intersection of the lesion with the subnetwork (of the functional map, subdivided by gene expression or receptor maps), and with the Monte Carlo parameters of treatment effect and recovery effect, to model plausible observational data. We have revised the Methods section to clarify all of these points (e.g. lines 432–448; 563–654).

Clinical measures of outcome were not included because the task requires not merely actual but also counterfactual outcomes that purely empirical data cannot provide. This is not a defect but a necessary feature of the analysis.

The comparator approach was to allocate patients on the basis of arterial lesion location. This however is seldom done in clinical trials or practice, beyond the selection of large artery occlusions for thrombectomy. For other types of intervention, vascular occlusion site is of very limited relevance and since identifying one is dependent on imaging interpretation, it is uncommon for this to be used as a prognostic variable at time of randomisation.

Thank you: we entirely agree that anatomy is rarely modelled, even as crudely as broad vascular territory, but that is not because it could not be modelled in theory or in practice. Our objective is to investigate the impact of modelling lesion anatomy as a potential change to future clinical practice, and here use vascular territory as the simplest anatomically informed baseline. Indeed, it is only by conducting studies such as this that consideration of a potential change in practice can be prompted.

Superior performance of a more detailed model is to be expected when compared against much simpler models, not least because the model incorporates a detailed brain lesion that, at least for acute trials, represents an outcome rather than a baseline feature. The reality of acute trials is that typically only a few key features are reliably known at the time of randomisation and treatment allocation.

Thank you for this comment. First, we entirely agree that more expressive models can be expected to be superior, but the *margin* of superiority in the setting of interventional inference is unknown, and no attempt to quantify it from real-world data has previously been made. Without such quantification, data-driven changes to policy are not possible. Second, that detailed anatomical information is not *currently* incorporated into trials at the time of treatment allocation does not imply it cannot be. It is feasible—even if challenging—to obtain lesion maps from imaging in the hyperacute setting. Furthermore, though hyperacute management is time-pressured and often involves inexpressive imaging such as simple CT, subsequent acute and chronic management provides sufficient time for the application of richly expressive imaging and its automated analysis. Crucially, any change in investigational practice requires evidence of its potential benefit from studies such as the present one, so unless we project ourselves into a hypothetical—yet realistic—future we would be locked in an endless Catch 22, with no prospect either of change or the evidence to justify it.

The authors argue that traditional clinical trial approaches cannot deal with heterogeneity, in the context of stroke, and propose that this can only be modelled by an almost individualised approach. The argument in places seems to propose that a truly individualised treatment selection is the ultimate aim; indeed, that results from randomised trials should be disregarded since they potentially discard treatments that might be effective for only a specific individual. This goes beyond the precision medicine, or even individualised medicine, concept since these approaches would at least use treatments that have an established evidence base and effect profile based on individual patient features to maximise benefits, minimise adverse effects, or avoid treatments for which there is no biological target. The argument put forward here appears to go significantly beyond these goals and even to advocate for testing therapeutic interventions solely on the basis of complex modelling without any traditional trial.

Thank you for this comment, and the opportunity to clarify this crucial point. We do not say that “results from randomised trials should be disregarded since they potentially discard treatments that might be effective for only a specific individual” and have carefully revised the text to ensure the reader cannot be left with this impression (lines 64–69: “*Where available, observations of responses to unconfounded treatment allocation—such as from a traditional RCT—may be combined with established low-²⁶ or novel high-dimensional¹⁹ multivariable modelling of conditional outcomes⁴⁻⁶. Randomization remains a powerful strategy, the best available means of protection from hidden confounding. But in the setting of unknown heterogeneity defined by many interacting factors, the necessary data scale becomes infeasibly large, forcing reliance on routine clinical data streams with*

limited control over randomization.”; lines 75–76: “In an ideal world, randomization would be combined with highly expressive models⁶: the question is where the compromise should lie when real-world constraints place the two in opposition.”; lines 371–374: “Where real-world feasibility permits it, complex modelling is ideally combined with randomization; equally, an effect drawn from observational data may be subsequently evaluated with randomized, prospectively acquired data from plausibly homogeneous subpopulations. Both approaches, including their combination, need to be considered in the light of real-world constraints.”). No-one could reasonably reject a policy because it is not optimal for a single individual; equally, no policy can be learnt from a single individual (assuming a single encounter) alone. The optimal treatment for a given patient cannot be known with certainty, but only estimated from the analysis of populations, based on evidence from delivered interventions. The critical question is what form such evidence should take. Current practice relies on RCTs employing simple estimands, over-reductive descriptions of the population, and inflexible statistical models. Complex anatomical heterogeneities—distributed subpopulations of arbitrary size—are ignored, despite good *a priori* grounds that they may be material to treatment outcomes, and that the inference for those subpopulations may therefore be wrong. What we show, crudely phrased, is that by conditioning on the “neighbourhood” of similar patients, causal models of sufficient expressivity can provide better evidence for the optimal treatment of an individual than standard population average effects, and that the superiority is maintained even with non-random treatment allocation, allowing inferences to be made from observational data where the necessary data scale is more plausibly achievable than with RCTs. So our objective is squarely aligned with precision medicine, and concerns precisely the evidence base that supports it. We have now clarified this in the text (e.g. lines 57–60: “Within such individualized prescriptive inference, the underlying statistical model seeks to determine not the ATE across the population, but the conditional average treatment effect (CATE) informed by a wide array of the patient’s distinctive features, resulting in estimates that are much more personalized, with reduced systematic variability across the population²².”; lines 81–109: “Since counterfactual outcomes of treatment are unobserved, this comparison requires a potential outcomes modelling framework^{27,28}. Crucially, we cannot rely on empirical evidence of treatment efficacy from standard RCTs, for we are concerned with the setting of heterogeneity where their assumptions are definitionally violated. In the absence of a ground truth, constitutive of the inferential problem here, we are compelled to use a semi-synthetic counterfactual framework^{29–33}. To promote real-world generalizability, our simulations are informed by large-scale, multi-modal empirical data: high-resolution lesion anatomy, and meta-analytic functional³⁴, gene transcription³⁵, and neurotransmitter receptor³⁶ maps. Moreover, we traverse a wide space of plausible real-world settings—systematically manipulating the richness of lesion parameterization, the magnitude and variability of treatment effects, the extent of treatment–outcome confounding, and the biological nature of the underlying deficit and treatment responsiveness—across the largest set of simulations ever performed with brain lesion data of any kind. Following well-established principles of causal inference^{37–39}, our approach relies on a causal model, represented by a directed acyclic graph, of the spatial relationship between brain lesions and functional anatomy—a map of disrupted functions—and between treatment receipt and physiological anatomy—a map of treatment responsiveness—that explains observable patient outcomes (Figure 2). This formulation allows us to quantify the impact on the efficacy of individualized prescription of both observable and unobservable treatment–outcome confounding, across the full space of plausible real-world scenarios, including a wide range of treatment and recovery effects. We can thus directly address the question of the relative contribution of randomization—the advantage of RCTs—and greater model expressivity and flexibility—the advantage of complex models of large-scale observational data, where real-world constraints force a compromise between the two approaches.”).

A more balanced perspective would consider more cautiously the role for complex modelling and the limitations of this approach. The model assumes availability of complex data – often not available at all, or only at later time points of clinical course – and that averaged models of receptor distribution or connectivity normalised into averaged anatomical space can be applied to individuals.

Thank you for this point: we emphasize the need for imaging at the time point of treatment allocation (lines 413–424: *“Note that the relevant features of ischaemic stroke—discrete spatial signals with high contrast-to-noise ratio on routine diffusion-weighted imaging—are comparatively invariant to the imaging instrument, and easy to project into standard stereotactic space, rendering the modelling pipeline readily implementable. Indeed, our study exclusively uses clinical data collected in the course of routine practice, drawn from a wide diversity of scanners and image sequence types. Though stroke is caused by anatomically organised damage to the brain, its outcomes are not exclusively determined by anatomy. A wide array of additional factors contribute to treatment outcome heterogeneity, placing varying demands on the expressivity of the models needed to capture them. The same approach can, and should, be extended beyond the anatomical domain. How readily any factor may be captured will, of course, vary, and there will inevitably be residual variation inaccessible to any practicable model. But the question of whether a factor is material to outcomes, and modellable within a given regime, requires investigation within the kind of framework demonstrated here.”*). To be absolutely clear, our model assumes only the availability of individual lesion maps and derived image analytics, which are fully automated and can be rapidly performed. Receptor distributions and gene expression data are employed in the semi-synthetic simulation framework only.

When could such a model be applied in the clinical journey? What types of intervention might be suitable targets for investigation (and which would not be)? How will the real-world performance of this complex model be evaluated?

Thank you for these questions. The approach we are proposing is applicable at the point where the input data is available, and a treatment decision must be made. Since the modelled treatment outcome dependency here invokes parenchymal characteristics, the inferences are most directly relevant to interventions post-thrombolysis or thrombectomy, such as those aimed at neural tissue preservation, reorganization or regeneration.

The question of evaluating real-world performance lies at the very heart of this endeavour. It is crucial to appreciate that there are no means—logically no means—of quantifying the real-world performance of *any* causal model in this setting because actual and counterfactual outcomes cannot be jointly observed, eliminating the possibility of a wholly empirical ground truth. This applies *both* to complex *and* simple models, within RCTs and without. No-one has ever provided real-world empirical proof of the performance of simple models in RCTs: the outcome from a RCT is *inferred*, and the only possible guarantees are theoretical, conditional on assumptions we here show to be violated. So the best that anyone could do with respect to real-world validation is semi-synthetic simulations of the kind we are performing here. One could, nonetheless, use rich phenotypes from our models in a prospective RCT large enough for sufficiently expressive phenotyping to be tractable, combining expressivity with randomization. But here, as in any trial—simply or elaborately modelled—the proof of the inferential method cannot be in the outcome of the trial.

It seems intrinsically unlikely that a modelling exercise, no matter how complex, would ever replace randomised controlled trials. The generalised critiques that are offered are based on hypotheses rather than clear evidence: it may indeed be the case that effective treatments have been discarded through inadequate characterisation of patient physiology, but this is a proposal with no current evidence to support it.

We respectfully disagree: the reviewer's position is radically at odds with a large and well-established literature on causal inference from observational data (along with, for what it is worth, 2021's economics Nobel Prize). Since interventional inference here has no purely empirical ground truth, the evidence we offer is the only kind of empirical evidence anyone could provide. To repeat the point Reviewer 2 above highlights as well-put: *"A failure of inference here cannot be distinguished from a failure of the intervention, and if the inferential approach is never questioned, replication will not correct the error but typically entrench it."* There is no empirical evidence that conventionally conducted and analysed RCT's *must* produce the correct answer: their fidelity is simply assumed based on theoretical arguments. It is true that randomization offers the strongest available protection against hidden confounding, but confounding is only *one of many reasons* a model may yield an incorrect answer, and examination of the optimal modelling approach in any particular domain requires investigation, not dogmatic assumptions.

Available – highly effective - treatments for stroke are based on logical selection criteria (eg exclusion of haemorrhage from trials testing treatments for ischaemic stroke) applied in clinical trials undertaken in large and representative populations, at least at phase 3 stage. Phase 2 trials have generally recruited narrower populations based on reasonable a priori evidence of being potential responders to treatment. There are many examples where treatment choices are informed by characterisation of patient subgroups based on traditional phase 3 trial data and where it is possible to infer different treatment by subgroup interactions that are relevant to patient care (eg age group and severity against treatment effect for intravenous thrombolysis). While a more efficient means of identifying a responder population for phase 2 trials would certainly be advantageous, it is unclear if this approach offers this functionality. Stroke trials usually adjust their analyses for known prognostic markers and plan subgroup analyses around biomarkers that identify important subgroups, including clinical variables such as age, severity, onset to treatment time, or imaging variables such as the volume of ischaemic tissue, site of an arterial occlusion, or extent of established ischaemic damage. Clinical investigators recognise that individual treatment effects are inherently unpredictable given the heterogeneity of age, brain health, comorbidities, medication and individual capacity for recovery, but nonetheless base coherent treatment decisions on reasonably well-informed average outcomes from clinical trials, nuanced by subgroups.

Thank you for this comment. The question is precisely what is "reasonably well-informed" and how it is to be determined. It is *a priori* wholly unreasonable to assume that the evolution and treatment responsiveness of (say) a hemiparetic syndrome assessed at presentation *must* be invariant to the anatomical properties of the lesion (e.g. SMA vs corticospinal tract damage), or that the patient's engagement with rehabilitation *must* be invariant to the integrity to dorsomedial and ventrostriatal pathways implicated in subserving motivation. To make such assumptions is to fly in the face of nearly half a century of functional brain mapping data (with lesions and, more recently, functional imaging) that shows finely structured and reasonably consistent patterns of functional organisation in the

human brain. We entirely agree with the reviewer that trialists attempt to adjust for “known” prognostic factors, but the critical question is how a factor is to become “known” unless it is modelled, and how treatment-by-outcome interactions *specifically* are to be estimated unless within the kind of framework we provide here? If the need to condition on the simple variables the reviewer cites is conceded, then so is the need for *other* potentially material variables, including the anatomical variables that lie at the causal heart of stroke. Physiological reasoning aside, that so much outcome variance currently remains uncaptured in stroke trials tells us that there is a great deal of room for improvement: to assume we already know all that can be known seems to us to deny the possibility of innovation without adequate warrant. We have clarified these points in the revised text (lines 385–387: *“Equally, though we may use a highly compressed representation—a two-dimensional embedding of lesion anatomy, for example—to stratify patients upstream of randomization within an otherwise conventional RCT, the loss of fidelity in the compression must be balanced against the gain in enabling randomization.”*; lines 418–424: *“Though stroke is caused by anatomically organised damage to the brain, its outcomes are not exclusively determined by anatomy. A wide array of additional factors contribute to treatment outcome heterogeneity, placing varying demands on the expressivity of the models needed to capture them. The same approach can, and should, be extended beyond the anatomical domain. How readily any factor may be captured will, of course, vary, and there will inevitably be residual variation inaccessible to any practicable model. But the question of whether a factor is material to outcomes, and modellable within a given regime, requires investigation within the kind of framework demonstrated here.”*).

The underlying assumption that the brain injury is the dominant determinant of disability (indeed the only one modelled – the authors contend that “...the disability caused by a stroke is directly related to the underlying disrupted functional anatomy”) - is an oversimplification. Stroke outcome is certainly determined in part by lesion size and location, but the dominant factor in clinical studies is the initial severity measured clinically, which typically renders measures such as ischaemic lesion volume non-significant in multivariable models. The functional outcome is also influenced by pre-stroke levels of disability, comorbidities, age, occurrence of complications (including recurrent stroke) and social support. Clinical trials also have to balance risks and benefits of treatments to estimate risk-benefit balance, which is not all related solely to brain effects (eg systemic bleeding complications).

Thank you for this comment. To say that “the disability caused by a stroke is directly related to the underlying disrupted functional anatomy” is not to say that it is the sole determinant. Given that ischaemic stroke is definitionally caused by ischaemic damage to neural tissue, and the success of all highly effective treatments for ischaemic stroke is in proportion to the extent to which that damage is minimized, it would be hard to deny a direct relation. Equally, that crude measures such as lesion volume may be poorly predictive over initial severity does not mean that finer anatomical featurization could not be superior, as many extant lesion-deficit studies have already shown. This position implies a belief—that brain tissue is interchangeable—our knowledge of the architecture of the brain surely makes profoundly implausible. The question of the expressivity of lesion patterns need investigation of the kind we conduct here, and cannot be dismissed out of hand simply because current practice ignores it.

But—crucially—our investigation *does not* presume that the relation between damage and outcome is invariant: on the contrary, in modelling the full potential range of treatment and recovery effects we are explicitly modelling the influence of other factors, just as one would in real-world practice. We isolate the anatomical because anatomy is so central to the cause and consequences of stroke, and

because the functional anatomical organization is plausibly learnable by models of sufficient expressivity. Our conclusions therefore do not depend on any assumption of anatomical invariance.

Finally, to emphasize the need for modelling anatomical factors with greater expressivity is not to neglect other factors such as systemic bleeding. Indeed, moving to a more expressive modelling framework allows other factors, where structured, to be handled better, so their existence is an argument in favour of the approach we advocate, not against it.

Thank you for your useful feedback and time in reviewing our submission.

1. *The authors have offered a detailed rebuttal and updated their text.*

Thank you for your detailed further thoughts on our paper.

2. *The manuscript overall continues to lack balance. Broad statements continue to dismiss RCTs and fail to acknowledge the limitations of the present investigation into complex modelling, or acknowledge the current study's own inherent biases. The text overstates the importance of the current investigation in somewhat grandiose statements. It would be appropriate to more clearly acknowledge the limitations and biases of the modelling exercise and consider where and how the hypothesis that has been generated here could be tested.*

Thank you for this comment. We have carefully revised our text to ensure maximum clarity on what can and cannot be concluded from our analysis. Our task is to quantify the comparative impact of model expressivity and non-random treatment allocation in making individual-level inferences about therapeutic effects, and to report our results. Our analysis shows that across the widest range of conceivable empirical parameters ever evaluated in this context, model expressivity has a greater impact on fidelity than random treatment allocation. This analysis directly compares the fidelity under *both* low-dimensional randomized regimes (i.e. RCTs based on simple estimands) *and* high-dimensional observational regimes, and reports the difference. Note that since stroke RCTs have never been subjected to such an analysis before, it would not be valid to argue that their widespread use discounts the significance of the only large-scale, comprehensive, direct comparison ever conducted. Yes, there are limitations to any counterfactual testing framework—that is the fundamental problem of interventional inference, as we make clear—but the limitations apply to the evaluation of *all* inferential methods, both established and novel. An important aspect of our contribution is to raise awareness of this neglected crucial point: unless we conduct comparisons of this kind we will never know if any one approach—established or new—is better than another, and relying on historical practice is not a defensible position, as the other reviewers have acknowledged.

We should also be clear about what constitutes a valid test of *any* inferential method here. Running an actual empirical trial (RCT or observational) with vs without complex modelling is *not* a valid test, as we definitionally lack a ground truth in such a setup at the individual scale. In either case, a “significant” result may be correct or incorrect, with no *empirical* grounds on which to favour one over the other. There are *statistical* grounds, however, for doubting a simple model—the demonstrable presence of heterogeneous structure in the residuals—and the best we can do is to establish comparative fidelity with the kind of evaluation we have conducted here. Nonetheless, one could pursue a two-stage process, where treatment-related heterogeneities identified by a complex model of observational data are accounted for in a subsequent stratified or covariate adjusted RCT with comparatively simple inference. Whether or not this approach is better than a direct application of complex models itself needs validation of exactly the same kind as performed here, but theoretically adds protection against hidden confounding. We now suggest such future work in the revised text.

3. *From the perspective of a clinician and clinical trialist, modelling is involved in current trial designs, and the complexity of the modelling is limited by the availability of relevant data.*

Thank you for this comment. We entirely agree models are limited by data availability. But only a small subset of the data routinely collected during clinical care is commonly modelled. Almost every patient presenting to hospital with suspected stroke undergoes brain imaging—with CT and/or MRI—as a first-line investigation and there is no defensible *a priori* reason not to model it. Although MRI, in particular the DWI sequence, is more expressive of the anatomical patterns of injury, novel image analytic methods are increasingly making the extraction of detailed lesion

maps from CT possible¹⁻⁴. Even where, as typically with MRI, high-quality imaging is not available in advance of thrombolytic or thrombectic therapy, subsequent therapy—e.g. neuromodulation, neuroreorganization, and rehabilitation—may still be informed by it. In any event, it would be hard to deny that the trajectory is towards richer, more detailed investigations, and equally hard to deny that there is a need for more nuanced characterisation of patients given the limited prognostic power of current biomarkers. Our work is intended to be addressed to the (near) future as well as the present. We have included an additional description within a newly-included limitations segment of the Discussion reflecting on the breadth of acute ischaemic stroke captured within this dataset (e.g. lines 432–464).

4. *While the authors argue that a demonstration of superiority of complex multidimensional modelling might drive acquisition of richer (imaging) data, this argument fails to recognise many practical limitations, discussed further below, which inevitably bias the subgroup in whom suitable complex data can be obtained.*

Thank you for this comment. We entirely agree that *equity* of care—minimization of biases across diverse subpopulations—is a cardinal imperative in medicine. Variations in the availability and quality of investigational data may introduce such biases. But our approach neither introduces nor amplifies this risk, indeed it provides a powerful means of addressing inequity, for the following reasons.

First, imaging is widely agreed to be central to the investigation of acute stroke, and is performed in almost all patients. Indeed, all stroke management guidelines we are aware of mandate brain imaging. It could hardly be otherwise given that the nature and anatomical pattern of damage is widely agreed to be material to outcomes, for it very strongly influences the diagnosis and observed deficit. The anatomical information we rely on here is contained in routinely performed imaging: all the data used in our study comes from routine clinical care. Yes, extracting detailed anatomical descriptions is considerably harder from CT, but it is not impossible; equally, advances in other modalities, such as low field MRI, are increasingly widening access to more expressive imaging (e.g. lines 432–453). The direction of travel in the field is towards more expressive imaging, not less, and it would be remiss not to quantify its potential benefits in terms of what ultimately matters: treatment-conditional outcomes.

Second, where inequity arises across subpopulations defined by complex characteristics—e.g. specific anatomical patterns of injury—its identification and remedy depend on descriptions of commensurate *finesse*⁵. Any approach that ignores complex characteristics is blind to subpopulations defined by them and cannot—definitionally cannot—either identify or remedy associated unwarranted variation⁵. Subpopulations demarcated by complex, intersectional characteristics are no less important for being so defined; equally attending to them does not imply neglect of those more simply demarcated: this is not a zero-sum game. We have expanded on this point in the revised text (e.g. lines 441–447).

Third, there is a widely accepted moral imperative in medicine to act on all available information. If data contained in a routinely performed investigation are shown to be material to a patient's outcome, and—harnessed by an appropriate model—can helpfully inform clinical management, we have a moral duty to use them. If such data are available for subpopulation A but not subpopulation B, the morally correct response is to seek to extend data acquisition to B as far as possible, not to pretend that they are not available for A. Indeed, demonstrating a benefit for A is the strongest motivation for extending data acquisition to B (e.g. lines 441–445). Crucially, a good modelling framework makes use of data where available and imputes them where not,

yielding not only good performance for A, but potentially also better performance for B: there is no reason a good model drawing on A & B would perform worse in B than a model of B alone. In any event, just as we would not dismiss MRI in all patients just because it is contra-indicated in some, so we cannot reasonably dismiss an expressive model just because its inputs are not always available. We have expanded on this point in the revised text, with an explanation of how imputation could handle missing data here, even in the context of structured missingness.

Finally, that there may be practical constraints to implementing an approach within current clinical pathways does not vitiate an argument for an alternative approach that *is* practically feasible, even if not currently implemented. Innovation necessarily implies projection into the future, and the future here is not speculative but readily deliverable with current technology.

5. *Further, the anatomical modelling approach is based on forcing diverse anatomy into an average brain volume, mapped onto averaged maps for receptor distribution, gene expression etc. Individual anatomy is more heterogeneous than the model systems, even less so when the extreme complexity of underlying disease (old infarcts, disruption of white matter integrity etc) is considered.*

We entirely agree that stroke is marked by great heterogeneity: this is precisely why we argue it ought to be modelled with commensurately expressive model architectures. Our modelling approach employs lesion segmentation and non-linear registration algorithms optimized for lesion data that are proven to preserve fine anatomical detail even when represented in template space⁶⁻⁸. Note a common anatomical reference frame is inevitably required if meaningful statistical comparisons are to be made across a population of patients. In any event, the detail of a high-resolution voxel-wise lesion map is orders of magnitude greater than either lesion volumes or crude vascular territories, which is the best current trials use. It is true that finer heterogeneity may be resolved with (say) multispectral imaging sensitive to both acute and chronic damage or with even more expressive models. But—crucially—our modelling approach *explicitly takes unmodelled finer heterogeneity into account* by evaluating performance across the full spectrum of “response noise”, i.e. the strength of the association between anatomy and treatment-conditional outcomes. Unmodelled heterogeneity would appear as noise—a more weakly learnable association—and so is included in the parameter space we cover. It would, of course, differ from noise in being learnable by an even more expressive model. But that would be an argument for more, not less, expressive models, and for advancing further in the direction we set out here. In short, unmodelled heterogeneity would not undermine the conclusions we draw here, but would amplify them further.

We should be clear about the role of receptor and gene expression distribution maps here. They are used as empirical priors for the *spatial structure* of plausible patterns of treatment-conditional dependence. We are not interested in any one specific receptor or gene, but in the class effect evaluated across all possible receptors or genes as exemplars of plausible physiological modulation. Again, that there will be finer patterns of treatment-conditional dependence is only an argument for more, not less, expressive models, and the scenario where unmodelled heterogeneity corrupts the treatment-outcome relation is already covered in our analysis.

6. *The text argues that observational data should be considered on the basis of the simulation exercise as having superiority over RCT data. This fails to acknowledge the many serious concerns about observational data, which are often highly biased and misleading even when a seemingly compelling hypothesis can be built around them (aggressive control of hyperglycaemia in ICU patients might be a pertinent example of the dangers of reliance on observational data). Again, the*

manuscript could better recognise the limitations of the current study and place this (currently very strong) recommendation in more appropriate context.

Thank you for this comment. We quantify the comparative impact on fidelity of model expressivity vs non-random treatment allocation: *the risks of confoundedness—both hidden and learnable—are explicitly modelled here*. It would be incorrect to say we are failing to acknowledge bias in treatment allocation: we are explicitly modelling its impact. We entirely agree, however, that there are many ways in which observational data may be corrupted to a greater degree than trial data, and we have modified our manuscript to more clearly express the comparative balance in both risk and benefit (e.g. lines 432–464). We now also clarify that *all* forms of inference here have shortcomings, and clarification of their capabilities requires careful analysis of the kind we have conducted here, with sensitivity to the specific context of use (e.g. lines 454–464).

7. *The Abstract states “randomized controlled trials, [are] based on simple descriptions of presumptively homogeneous populations,” which I think is a considerable over-simplification. Similar comments appear elsewhere in the text.*

Thank you for this comment. We have now clarified that the simplicity we speak of is statistical: the number and expressivity of the variables used to model treatment effects (e.g. line 2: “*typically using simple estimands*”). Primary outcome measures in stroke are near-universally real numbered or categorial scalars. The number of covariates is overwhelmingly less than ~6. This limitation is reinforced by the properties of the statistical models used—general linear models—that lack expressivity to model high-dimensional data or complex non-linear effects.

8. *RCTs involve many stages of investigation, from first-in-man drug trials, through proof of concept, to generalisable efficacy, to real world implementation and may include the full range of interventions from simple molecules through to complex care protocols. In trials that are seeking to establish the effect of an intervention at an early stage (typically phase 2 trials) the risk-benefit balance of the intervention is not sufficiently characterised, and there are ethical and practical barriers to recruiting large numbers of participants. Strategies to reduce heterogeneity are typically employed through a range of means including selection criteria to focus on a (predicted) responder population with low risk of adverse effects, including clinical and brain imaging approaches. The descriptions of populations are typically not simple and extend to a wide variety of demographic features, physiological variables, medical and medication history, stroke features and imaging. The text states that estimands are always simplified and lack nuance, but this is in large part driven by the constraints of sample size and practical barriers to obtaining all the desirable information in all participants.*

Thank you for this comment. Nothing we say disputes that drug development is an elaborate process involving many stages. We are concerned with a specific aspect here: inference to individual-level treatment effects. Though RCTs infer effects at the population level they are used to prescribe treatment at the individual level. Fidelity at the individual level therefore arises in real-world use where the clinician must select between a set of approved agents (or no agent) in any individual patient. Since individual-level fidelity naturally propagates to the population, identifying better models of treatment effects also has consequences for population-level inference in the late stages of clinical evaluation. But this is not our focus here: we report individual-level fidelity, not efficiency at detecting population-level effects.

The number of characteristics modelled in the statistical evaluation of treatment effects is typically very small, and their putative relations are assumed to be simple: it cannot be otherwise, as we say, because the underlying statistical frameworks preclude it. The reviewer concedes this point by saying simplification is driven by “the constraints of sample size and practical barriers to

obtaining all the desirable information in all participants”. It is surely self-contradictory to argue both that we are wrong in claiming estimands are commonly simple, and that estimands *are* commonly simple owing to constraints.

We entirely agree on the practical constraints of sample size—this is precisely why we suggest consideration of adaptation to observational data regimes is necessary, and the challenges of therapeutic inference in an observational setting need to be explored. But even so, though there are logistical and commercial constraints to large Phase 3 trials, they are neither insurmountable nor justify exclusion of discussion of their statistical necessity. Moreover, both post-market surveillance, and real-world individualized selection between multiple approved treatments, offer data at substantial scale. Indeed, the last of these is the most immediately applicable scenario here, as we now re-emphasize in the revision (e.g. lines 454–458).

We respectfully disagree complex models are critically limited by the availability of rich data. As we make clear above, high-resolution brain imaging—CT and/or MRI—is near-universal in stroke, and rich imaging characteristics are never modelled not because they are not available but because of explicit modelling decisions to keep the statistical model simple. Even if MRI were taken to be an essential pre-requisite, it is perfectly feasible to run an MRI-based acute stroke service (e.g. Cheng et al., 2023⁹) given sufficient clinical justification. Studies of this kind are precisely the kind of justification that is needed. We have clarified these points in the manuscript (e.g. lines 432–453).

9. *It may be the case that complex modelling could further assist patient selection for phase 2 trials but this is a hypothesis that requires to be tested, and the constraints over availability of complex anatomical imaging for such an approach need to be recognised as this biases the population and inevitably leads to losses to follow-up.*

We respectfully disagree. We are concerned with the fidelity, at the individual-level, of inferences to treatment effects with high- vs low-dimensional models under randomized vs non-randomized treatment allocation regimes, where response to treatment is sensitive to the anatomy of the brain. The conclusion of our investigation is *not* a hypothesis to be tested by a trial of any specific agent, *for what we have evaluated is the inferential framework itself*. We recognize this can be a difficult idea to convey, especially to non-mathematical audiences, which is why we introduce it in great detail with a vividly illustrative example in Figure 1.

It is plain that inferences here cannot be *assumed* to be indifferent to anatomy given the fundamental nature of stroke and the functional deficits it causes, and we *demonstrate*—across the widest range of real lesions and biologically plausible patterns of treatment responsiveness ever evaluated—that high-dimensional models are superior to low-dimensional models in inferring individual-level treatment effects, even in the presence of non-random treatment allocation. The validity of this conclusion is not tested by the outcome of any treatment, individually or within an actual trial, because treatment outcomes are necessarily counterfactual and lack ground truths. Yes, one might *illustrate* the application of our approach to an actual treatment, but neither success nor failure would be grounds to accept or reject our analysis, whose conclusions are grounded in the mathematical properties of lesions and the functional organization of the brain. Any given trial may fail because the agent does not actually work, and it may succeed with simple statistics because the effect size very large (e.g. thrombectomy in large vessel occlusion): if we do not know the effect size *a priori* we cannot tell how accurately we have estimated it. This is true of *any* statistical analysis here, both established and novel.

Second, as we say above, imaging here is near-universal: what has hitherto been lacking is modelling of its rich features, not the availability of data. We address this in further detail below.

Third, we do not see how loss-to-follow-up is relevant here. We are dealing with the grounding of treatment decisions at the point of investigation and the use of any model—simple or complex—is wholly independent of any follow-up arrangement. All that needs to be ascertained at follow-up is the outcome.

10. *In large phase 3 (or later) trials, heterogeneity is inherent, and is often embraced in the trial design; generalisability of treatment effects that are suggested in early phase trials is important, as is a more realistic estimate of average effect size, and the larger sample sizes that are involved in phase 3 trials allow more reliable subgroup analyses. How complex modelling could assist should be considered more critically. A major limitation of phase 3 trials – particularly in the stroke field, historically - is that sample size is insufficient to allow reliable statistical analysis of subgroups. Unless complex modelling allows sample size to be reduced while retaining robust statistical analyses this approach might not provide any advantage.*

Thank you for this comment. The statistical efficiency of a trial is critically dependent on the unexplained variability of the outcome. Where—as here—substantial variability is introduced by the complex relationship between lesion patterns and outcomes, a better—necessarily more expressive—model of the lesion-outcome relationship will improve statistical efficiency, reducing sample size requirements, and increasing statistical power for the same sample size. But it is precisely because of the challenges of obtaining sufficient sample sizes in Phase 3 that we evaluate the utility of large-scale observational data, even in the presence of non-random treatment allocation bias. We now expand on this aspect in the Discussion (e.g. lines 454–464).

11. *The model data set is based on MRIs that were acquired in a hospitalised acute stroke population. Typically MRI in UK hospitals is undertaken in the subacute period 24–48h (or later) after onset. The main paper gives no detail on the population from which these scans were acquired at all, and the supplementary data presents only an age histogram. There is no information at all that would allow a reader to judge how representative a population has been studied – demographics, medical history, stroke onset and presentation, treatment, timings for these, clinical severity and outcomes are all absent. Selection bias is not acknowledged by the authors but is plain from the flow chart (Fig S3) which illustrates that even from the subset of patients for whom imaging was available to review, >20% of scans were excluded due to technical failure or presence of other pathology. Since tolerance of subacute MRI depends on severity of stroke, many severe patients are excluded, as of course are all those who do not survive for long enough to have such imaging. Intolerance of MRI due to agitation, claustrophobia or contraindications such as ferromagnetic foreign bodies or pacemakers further reduces the representation of the stroke population. The study scans may exhibit heterogeneity of anatomical locations, but this dataset is far from representative of all strokes. The argument that acquisition of complex imaging data will be feasible and superior compared with conventional data is undermined by the figure legend that notes that 16% of these datasets lacked the most basic clinical information (age or sex). The current model has therefore selected imaging data alone, and has neither sought nor compared conventional clinical data at either baseline or at follow-up. Having access to a large number of MRIs does not equate to generalisability.*

Thank you for this comment. The data is drawn from UCLH’s comprehensive stroke service to which all patients with suspected stroke falling within the North Central London area are referred. This is a richly varied population—demographically, socioeconomically, and pathologically—reflecting the composition of one of the most diverse cities on Earth. We present age and sex data

in the supplementary material. All patients with radiologically confirmed ischaemic stroke were eligible for inclusion, and the *only* selection criterion was the availability of MRI of sufficient quality and the absence of confounding major other pathology (e.g. lines 432–453). The clinical policy of the stroke service over the interval of data collection was to perform MRI on *everyone* in whom it was not contra-indicated. There are no grounds for claiming that the dataset is “far from representative” for the only subpopulations relatively undersampled are those in whom MRI is contraindicated: a small minority.

But under-representation of small subpopulations in our analysis cannot conceivably alter the findings, for the following reasons. To the extent to which such subpopulations exhibit distinct, apprehensible complex anatomical patterns of damage, their inclusion would increase the advantage of complex modelling, for their presence would increase successfully modelled complex heterogeneity. To the extent to which such populations exhibit anatomical patterns too complex to be successfully modelled, their introduction would correspond to added “response noise”, which we already model across the entire range. To the extent to which such populations exhibit unusually simple anatomical patterns—deeply implausible given that both vascular anatomy and mechanisms of damage are widely shared—they can only be differentiated from those exhibiting complex patterns by models expressive enough to capture complexity. And such differentiation would be undeniably necessary, for we have already shown that the majority benefit from complex modelling. In short, under-representation of a process-defined minority cannot—logically cannot—substantively undermine our results, and it would be misleading to claim that it can. Moreover, it would be perverse to deny what is shown to be true of the vast majority of patients on the speculative basis that it might not be true of a minority, just as it would be perverse not to use an investigation or treatment on the grounds that it cannot be used in all patients.

Our analysis is focused on rich anatomical features derivable from routine imaging over and above clinical features. That clinical features alone, as conventionally captured, do not provide sufficient expressivity is demonstrated by the widely acknowledged—including by the reviewer in comments above—heterogeneity of observed outcomes. The relevant expressivity of clinical features is severely constrained in two critical ways. First, standard clinical scores such as NIHSS are both highly reductive and have been shown to have very low intrinsic dimensionality¹⁰, conveying limited information. Second, a deficit reductively described at baseline cannot conceivably differentiate between damage to different components of the neural substrate—with mechanistic and therefore potentially therapeutic consequences—that underpins the underlying function. For example, the NIHSS measures language on a 3-point scale: how could that conceivably capture damage to the intricate, anatomically distributed network of language function¹¹, its evolution over time, and its response to component-specific treatment? Yes, the value of modelling clinical features with greater nuance, through more expressive measures than standard scores, should be the subject of exactly the kind of investigation we have conducted here, but that is an argument for extending our approach to the clinical domain, not an argument against our analysis of the imaging domain. Adding more informative features—clinical or other—can only amplify the effect of increasing model expressivity we report here (e.g. lines 448–453).

12. *Ischaemic lesions are dynamic in the first days after stroke, and a DWI lesion at 6h offers very limited insight into the extent or topography of the lesion at 24h or 48h. This is an important caveat to the dataset used here, and should temper the language used to support the applicability of the modelling approach. We do not know whether complex modelling in other circumstances would yield similar (synthetic) results; nor do we know what imaging modalities (or other data) would act as a reliable input to a similar multidimensional modelling exercise. It is likely that each imaging*

modality will have its own limitations and characteristics. Many important acute modalities have limited resolution and anatomical delineation of lesions is much less clear than with DWI.

DWI lesions are indeed dynamic but the claim that DWI imaging “at 6h offers very limited insight into the extent or topography of the lesion at 24h or 48h” is not supported by the studies we are aware of¹²⁻¹⁹. Indeed, longitudinal comparisons between acute DWI and chronic T2 lesions show close correspondence. In any event, variability in the segmented anatomy is explicitly modelled in our analysis as part of “response noise” and shown not to affect the conclusion. Since what we are modelling here is the anatomical pattern of damage, not imaging-specific signal, our conclusions are generalizable to other modalities from which the anatomy can be inferred, though fidelity will, of course vary. We discuss this aspect further in the revised Discussion (e.g. lines 433–437).

13. *It should be further emphasised that there were no clinical outcomes used in the modelling exercise. Again, very broad conclusions are being drawn from an exercise that does not even include basic clinical evaluations that could be understood by patients or families, or clinicians.*

The reason we do not model *actual* clinical outcomes is because we must evaluate *counterfactual* clinical outcomes within an empirically informed semi-synthetic framework that makes it possible to establish a ground truth. This is the fundamental problem of causal inference that no method—neither established nor new—can circumvent. If we were to evaluate a *specific* clinical outcome in the context of a *specific* treatment, we could not—information theoretically could not—distinguish the absence of a real effect from a model’s inability to quantify it correctly. Instead, we explore the widest conceivable range of *possible* outcomes, *possible* treatment effects, and *possible* noise and bias corrupting their inferred relation. This is the only kind of test that can answer the question we have set out to answer (e.g. lines 94–98, 603–609).

We entirely agree with the reviewer that these ideas are not simple to convey, but our first duty to patients and their families is to assure them we are using the best possible strategy for selecting the treatments we offer. We would be in neglect of this duty if we did not examine difficult but crucial questions that lie at the heart of causal inference, especially when much of the intellectual framework underlying recent advances is already accepted in other fields (indeed attracted the Nobel Prize for Economics in 2021).

14. *I would regard the current study as no more than hypothesis-generating. The hypothesis that complex modelling provides more robust insight and predictive value than simpler modelling should be investigated, but a modelling exercise involving simulations based on a highly selected dataset subjected to complex image analysis is not a basis for generalised conclusions.*

We respectfully disagree. First, the population is not “highly selected” but excludes only the few for whom MRI imaging was contraindicated, and whose exclusion, as we have argued above, could not conceivably alter the conclusion. Second, the use of “complex image analysis” is does not limit the generalizability of the findings, indeed their generalizability is supported by coverage of the widest conceivable range of possible parameters across more than 100 million numerical experiments. Most importantly, our conclusion does not *generate* a hypothesis of the superiority of high-dimensional over low-dimensional methods in interventional inference but *tests* precisely that in the only manner the fundamental nature of counterfactual inference allows.

15. *In summary I feel that the present analysis has merit in demonstrating an approach to complex modelling of MRI-defined brain anatomical lesions that might yield better predictive value for individual cases in whom imaging is available compared with simpler modelling, but that the*

requirements for appropriate imaging inputs to the model have not been explored with respect to modality or timing, no clinical outcomes have been assessed, and the complex model has not been compared with the predictive value of simpler models based on conventionally available variables.

Thank you for this comment. We are glad that you recognize merit in our work. As we say above, our analysis relies on anatomical maps derived from MRI but explicitly includes modelling of widely varying degrees of corruption of the relation between phenotypes and outcomes that takes into account differences in imaging quality. We do not evaluate actual clinical outcomes as the analysis necessarily requires counterfactual outcomes, here explored across the full range observable in reality. We explain at length above why clinical variables could not conceivably undermine the results obtained here.

16. *Replication of the findings in another dataset and exploration of the robustness of this approach with respect to other imaging modalities or timings is necessary. The present approach could be evaluated in the context of a phase 2 clinical trial that uses clinical outcomes, and compared against current best standards. The approach is more likely suited to a subacute population and therefore to rehabilitation or regenerative therapy strategies and more work is required to investigate potential applicability to the acute environment.*

While replication is always desirable, this is by far the largest and most comprehensive analysis of its kind—in terms of data scale, diversity and inclusivity, and range of potential lesion-outcome relationships and outcomes. We naturally wish to *apply* the approach—and encourage others to do so—to real world data, but its validity is not testable by coherence with an RCT because, as we illustrate in the example in Figure 1, and as is widely acknowledged in the field of contemporary causal modelling, an RCT is not guaranteed to provide the correct answer. Of course, others should conduct similar investigations, and the same approach should be extended to other sources of rich data with a plausible bearing on outcomes, including high-dimensional characterisation of cognitive and behavioural deficits.

17. *“Our results indicate that complex modelling with richly represented lesion data is critical to individualized prescriptive inference in ischaemic stroke.” – I would suggest “may offer an approach to improve upon” rather than “critical.”*

Thank you for this suggestion. We have changed to “may substantively enhance”, for that is what our results show.

18. *Discussion p17 “this question... requires semi-synthetic simulations,” – I think the question of whether there are subgroups for whom a treatment might be effective can be explored with simulations, but this is only one approach. There are many examples of subgroup analyses (the great majority of which have been misleading) in the conventional trial world, and whether the simulation approach brings any useful insights is at best a testable hypothesis, but would still require confirmation in a conventional RCT.*

Once again, we respectfully disagree. We are not examining a specific subpopulation, a specific agent, or a specific outcome. We are examining the fidelity of different forms of inference in the context of an anatomically sensitive disorder, which stroke is universally accepted to be. The question of which form of inference is better is precisely what we are addressing, so it cannot be—logically cannot be—foreclosed by one answer. The question cannot be tested by an RCT but by the kind of analysis we have conducted here, as acknowledged by the other reviewers, and as accepted in the wider field of causal inference. The belief that RCTs evaluated by simple low-dimensional models are the ultimate arbiter of truth in interventional inference is widespread in

the clinical world but it is incorrect, as shown by the simple example given in Figure 1. Any form of interventional inference—traditional or new—involves assumptions about the structure of the underlying system, and where these assumptions are manifestly violated—as unquestionably here—the only licit form of comparison of inferential performance is through semi-synthetic ground truth simulations. The robustness of that evaluation is a function of the scale and range of data and parameters surveyed, here the largest and widest ever attempted.

Randomization is nonetheless a powerful ingredient of inferential testing, and should be used where possible. We now expand on how the approach presented here can be combined with randomized studies (e.g. lines 376–381). But the success or failure of such extension does not alter the significance of our analysis, for it would be manifestly circular reasoning to claim that the fidelity of RCTs is determined by RCTs themselves.

Thank you again for your detailed thoughts on our revised paper.

References

1. Yang, H., Huang, C., Nie, X., Wang, L., Liu, X., Luo, X., and Liu, L. (2023). IS-Net: Automatic Ischemic Stroke Lesion Segmentation on CT Images. *IEEE Transactions on Radiation and Plasma Medical Sciences* 7, 483–493. <https://doi.org/10.1109/TRPMS.2023.3246496>.
2. Soltanpour, M., Greiner, R., Boulanger, P., and Buck, B. (2021). Improvement of automatic ischemic stroke lesion segmentation in CT perfusion maps using a learned deep neural network. *Computers in Biology and Medicine* 137, 104849. <https://doi.org/10.1016/j.compbiomed.2021.104849>.
3. Mäkelä, T., Öman, O., Hokkinen, L., Wilppu, U., Salli, E., Savolainen, S., and Kangasniemi, M. (2022). Automatic CT Angiography Lesion Segmentation Compared to CT Perfusion in Ischemic Stroke Detection: a Feasibility Study. *J Digit Imaging* 35, 551–563. <https://doi.org/10.1007/s10278-022-00611-0>.
4. Luo, J., Dai, P., He, Z., Huang, Z., Liao, S., and Liu, K. (2024). Deep learning models for ischemic stroke lesion segmentation in medical images: A survey. *Computers in Biology and Medicine* 175, 108509. <https://doi.org/10.1016/j.compbiomed.2024.108509>.
5. Carruthers, R., Straw, I., Ruffle, J.K., Herron, D., Nelson, A., Bzdok, D., Fernandez-Reyes, D., Rees, G., and Nachev, P. (2022). Representational ethical model calibration. *NPJ digital medicine* 5, 1–9.
6. Nachev, P., Coulthard, E., Jager, H.R., Kennard, C., and Husain, M. (2008). Enantiomorphic normalization of focally lesioned brains. *NeuroImage* 39, 1215–1226.
7. Foulon, C., Cerliani, L., Kinkingnehun, S., Levy, R., Rosso, C., Urbanski, M., Volle, E., and Thiebaut de Schotten, M. (2018). Advanced lesion symptom mapping analyses and implementation as BCBtoolkit. *GigaScience* 7, giy004.
8. Foulon, C., Gray, R., Ruffle, J.K., Best, J., Xu, T., Watkins, H., Rondina, J., Pombo, G., Giles, D., Wright, P., et al. (2025). Generalizable automated ischaemic stroke lesion segmentation with vision transformers. Preprint at arXiv, <https://doi.org/10.48550/arXiv.2502.06939> <https://doi.org/10.48550/arXiv.2502.06939>.
9. Althaus, K., Dreyhaupt, J., Hyrenbach, S., Pinkhardt, E.H., Kassubek, J., and Ludolph, A.C. (2021). MRI as a first-line imaging modality in acute ischemic stroke: a sustainable concept. *Ther Adv Neurol Disord* 14, 17562864211030363. <https://doi.org/10.1177/17562864211030363>.
10. Cheng, B., Chen, J., Königsberg, A., Mayer, C., Rimmele, L., Patil, K.R., Gerloff, C., Thomalla, G., and Eickhoff, S.B. (2023). Mapping the deficit dimension structure of the National Institutes of Health Stroke Scale. *EBioMedicine* 87.
11. Hertrich, I., Dietrich, S., and Ackermann, H. (2020). The Margins of the Language Network in the Brain. *Front. Commun.* 5. <https://doi.org/10.3389/fcomm.2020.519955>.
12. Löubld, K.-O., Baird, A.E., Schlaug, G., Benfield, A., Siewert, B., Voetsch, B., Connor, A., Burzynski, C., Edelman, R.R., and Warach, S. (1997). Ischemic lesion volumes in acute stroke by diffusion-weighted magnetic resonance imaging correlate with clinical outcome. *Annals of Neurology* 42, 164–170. <https://doi.org/10.1002/ana.410420206>.
13. Warach, S., Dashe, J.F., and Edelman, R.R. (1996). Clinical outcome in ischemic stroke predicted by early diffusion-weighted and perfusion magnetic resonance imaging: a preliminary analysis. *J. Cereb. Blood Flow Metab.* 16, 53–59. <https://doi.org/10.1097/00004647-199601000-00006>.
14. Barber, P.A., Darby, D.G., Desmond, P.M., Yang, Q., Gerraty, R.P., Jolley, D., Donnan, G.A., Tress, B.M., and Davis, S.M. (1998). Prediction of stroke outcome with echoplanar perfusion- and diffusion-weighted MRI. *Neurology* 51, 418–426.
15. Wheeler, H.M., Mlynash, M., Inoue, M., Tipirneni, A., Liggins, J., Zaharchuk, G., Straka, M., Kemp, S., Bammer, R., Lansberg, M.G., et al. (2013). Early Diffusion-Weighted Imaging and Perfusion-Weighted Imaging Lesion Volumes Forecast Final Infarct Size in DEFUSE 2. *Stroke* 44, 681–685. <https://doi.org/10.1161/STROKEAHA.111.000135>.
16. Ritzl, A., Meisel, S., Wittsack, H.-J., Fink, G.R., Siebler, M., Mödler, U., and Seitz, R.J. (2004). Development of brain infarct volume as assessed by magnetic resonance imaging (MRI): Follow-up of diffusion-weighted MRI lesions. *Journal of Magnetic Resonance Imaging* 20, 201–207. <https://doi.org/10.1002/jmri.20096>.
17. Tong, D.C., Yenari, M.A., Albers, G.W., O'Brien, M., Marks, M.P., and Moseley, M.E. (1998). Correlation of perfusion- and diffusion-weighted MRI with NIHSS score in acute (<6.5 hour) ischemic stroke. *Neurology* 50, 864–870.
18. Filippi, M., Horsfield, M.A., Bressi, A., Martinelli, V., Baratti, C., Reganati, P., Campi, A., Miller, D.H., and Comi, G. (1995). Intra- and inter-observer agreement of brain MRI lesion volume measurements in multiple sclerosis: A comparison of techniques. *Brain* 118, 1593.
19. Thijs, V.N., Lansberg, M.G., Beaulieu, C., Marks, M.P., Moseley, M.E., and Albers, G.W. (2000). Is early ischemic lesion volume on diffusion-weighted imaging an independent predictor of stroke outcome? A multivariable analysis. *Stroke* 31, 2597–2602.

Reviewer #4 (Remarks to the Author)

Based largely on the rebuttal text, my understanding is that the authors position includes the following:

We are grateful for the reviewer's further thoughts. The reviewer's representation of our position suggests several misunderstandings. We have carefully revised the manuscript to minimize the risk of being misunderstood by all readers—reviewers and the wider community—and precise, specific, detailed, logical, evidentially grounded argument seems to us to offer the best prospect of success in achieving this.

• Their existing data are sufficiently robust, and no amount of additional data would, or indeed could, mathematically, modify their conclusions. Concerns that bias of MRI acquisition to younger, fitter patients; the supposed stability of lesions over time (not what is observed clinically in the acute stroke setting); and reliance on single centre data; are essentially dismissed.

This is amongst the largest stroke studies ever conducted, and by far the largest to include expressive modelling of high-resolution representations of ischaemic brain lesions. As elaborated in detail in the paper and the previous rebuttal, it includes the vast majority of patients consecutively managed at a large, fully inclusive stroke centre serving one of the most diverse concentrated populations in the world. The conclusions of our study rest on the characterization of structured heterogeneity in ischaemic lesion patterns, statistically evaluated by out-of-sample model comparison. This heterogeneity cannot suddenly vanish with the addition of even more patients, anymore than a distinction between MCA and PCA territory ischaemic stroke could vanish with the

addition of ACA territory stroke. As we say in the previous rebuttal, the shortcomings the reviewer identifies—the desire for an even wider, more diverse population from multiple centres—can only *increase* heterogeneity, not reduce it. On the subject of the stability of lesions over time, the reviewer’s clinical intuition may be correct, but we can only reasonably adduce evidence from objective, empirical studies such as those we cite.

Please see our statements in lines 366–371: “As in any study, there will nonetheless be minority subpopulations with comparative undersampling for whom the generalizability of the findings cannot be assured. But where material heterogeneities have been identified in the majority population, the existence of undercharacterized subpopulations does not remove the need to account for them but rather provides grounds for insisting on better coverage. And if such subpopulations are defined by complex characteristics, including lesion anatomy, their existence strengthens the case for more expressive models.”

• *The conclusions cannot be validated or tested since clinical trials are inherently unable to identify true treatment effects*

As we explain at some length in the paper and in the previous rebuttal, and as acknowledged by the other reviewers and widely established in the field of causal modelling, interventional inference is fundamentally counterfactual and therefore always lacks a ground truth. We cannot know the *potential* outcome of treating a patient alternatively because we only get to treat each patient once. Consequently, we cannot know the true effect size of a treatment in a heterogeneous population because subpopulation-specific effects may diverge. Moreover, one cannot conduct a RCT to prove that RCTs are better than another form of inference any more than one could conduct a *t*-test to show that *t*-tests are better than another test. The validity of any form of inference here depends on its foundational assumptions, and where—as here—the assumptions are manifestly violated, the only validation we can perform is of the kind we conduct in this study. We have never said that a RCT cannot identify a true treatment effect—in the setting of treatment response homogeneity it obviously should—only that RCTs can neither be assumed to be infallible nor could possibly be the mechanism for validating the form of inference they presuppose.

To be absolutely clear that our points are not misinterpreted in this way, we have modified our statements to ensure that we are referring specifically to propagating cohort-level simple estimands to an individual, where the context is heterogeneous, as in lines 96–98: “we cannot rely on empirical evidence of treatment efficacy from standard RCTs, for we are concerned with the setting of heterogeneity where their assumptions are definitionally violated”.

- *All measures of clinical outcome are unreliable and should not be considered necessary to model*

This is a misrepresentation of what we have said. We have not said that “all measures of clinical outcome are unreliable” but that “clinical features alone, as conventionally captured, do not provide sufficient expressivity”. No-one could possibly argue that death (say) is unreliable, and no-one reading our text in good faith could conclude this is what we assert. Equally, we have not said that clinical features should not be modelled—quite the opposite, as emphasized in bold in our previous answer reproduced below—but that the absence of clinical feature modelling does not substantively undermine our conclusions.

“Our analysis is focused on rich anatomical features derivable from routine imaging over and above clinical features. **That clinical features alone, as conventionally captured, do not provide sufficient expressivity is demonstrated by the widely acknowledged—including by the reviewer in comments above—heterogeneity of observed outcomes.** The relevant expressivity of clinical features is severely constrained in two critical ways. First, standard clinical scores such as NIHSS are both highly reductive and have been shown to have very low intrinsic dimensionality, conveying limited information. Second, a deficit reductively described at baseline cannot conceivably differentiate between damage to different components of the neural substrate—with mechanistic and therefore potentially therapeutic consequences—that underpins the underlying function. For example, the NIHSS measures language on a 3-point scale: how could that conceivably capture damage to the intricate, anatomically distributed network of language function, its evolution over time, and its response to component-specific treatment? **Yes, the value of modelling clinical features with greater nuance, through more expressive measures than standard scores, should be the subject of exactly the kind of investigation we have conducted here, but that is an argument for extending our approach to the clinical domain, not an argument against our analysis of the imaging domain. Adding more informative features—clinical or other—can only amplify the effect of increasing model expressivity we report here.**”

We go on to say what we do not have a better way of expressing and hence reproduce below:

“The reason we do not model actual clinical outcomes is because we must evaluate counterfactual clinical outcomes within an empirically informed semi-synthetic framework that makes it possible to establish a ground truth. This is the fundamental problem of causal inference that no method—neither established nor new—can circumvent. If we were to evaluate a specific clinical outcome in the context of a specific

treatment, we could not—information theoretically could not—distinguish the absence of a real effect from a model’s inability to quantify it correctly. Instead, we explore the widest conceivable range of possible outcomes, possible treatment effects, and possible noise and bias corrupting their inferred relation. This is the only kind of test that can answer the question we have set out to answer. We already devote a great deal of the introduction to explaining this point, but have now expanded on it in greater detail.”

• Any imaging modality that yields structural imaging data can be used to derive models robustly and without being tested (despite inherently lower resolution, poorer discrimination of acute from chronic changes, etc)

This is a misrepresentation. We say:

“DWI lesions are indeed dynamic but the claim that DWI imaging “at 6h offers very limited insight into the extent or topography of the lesion at 24h or 48h” is not supported by the studies we are aware of. Indeed, longitudinal comparisons between acute DWI and chronic T2 lesions show close correspondence. **In any event, variability in the segmented anatomy is explicitly modelled in our analysis as part of “response noise” and shown not to affect the conclusion. Since what we are modelling here is the anatomical pattern of damage, not imaging-specific signal, our conclusions are generalizable to other modalities from which the anatomy can be inferred, though fidelity will, of course vary.** We discuss this aspect further in the revised Discussion.”

We do not say that “any imaging modality that yields structural imaging data can be used to derive models robustly” but that our conclusions are generalizable to other modalities from which the lesion anatomy can be inferred because we explicitly model variability in the quality of segmented anatomy. Even then, we clarify that fidelity will, of course, vary. But that an approach shown to be of value to patients may be dependent on imaging already widely and routinely deployed in clinical practice is no argument against using it, still less writing about it: it is an argument for making imaging more widely available, which is, in fact, the natural trend in stroke care worldwide.

The text fundamentally starts from the position that treatment effects of interventions in stroke cannot be considered reliable because stroke is heterogeneous. That the clinically-defined diagnosis of stroke reflects a wide range of underlying pathologies is not at issue: the apparently dogmatic approach of the authors however appears to disregard more than 30 years of substantial progress in treatment that has been successful largely because, in recognition of this heterogeneity, clinical trials selected participants based on more precise phenotypic characterisation of patients (much of it based on imaging).

The reviewer's statement is a self-contradiction. It cannot be both true that heterogeneity is not a problem and that substantial progress has depended on recognizing it. If we take the latter statement as the reviewer's position, then the question is only the requisite expressivity with which heterogeneity is modelled. Current clinical trials do not use rich representations of imaging, indeed our baseline for comparison—arterial territories—is more expressive than anything in routine use. If more precise characterization has been essential to progress, there is no reason not to extend it to the scale of the functional organization of the brain that we already know is directly expressive of the consequences of focal brain injury, in stroke and elsewhere.

The modelling approach uses only brain imaging data, with no other clinical information, and the model outcomes are based on anatomical correlates of imaging. Under these specific conditions the conclusion is that a complex model of anatomy outperforms a simple randomisation scheme: in itself this is unsurprising, although it is reassuring to see this hypothesis confirmed in a large dataset. Qualitatively it reaches the same conclusion as previous, much simpler, critiques of clinical trial design in stroke that have addressed the issue of heterogeneity and potential use of imaging to more precisely phenotype trial participants.

No-one has previously addressed the question of the requisite expressivity of lesion representations in interventional inference in ischaemic stroke, or the impact of the data and treatment allocation regimes, but if others have reached the same conclusion, then it is with evidence or articulation that have not been sufficiently persuasive across the domain (since current practice remains simple in its approach to phenotyping) or for the present reviewer as his or her position is expressed in the preceding paragraphs.

We have already addressed the issue of clinical information above. Anatomy is not a mere correlate in stroke: it strongly determines the manifestations of focal brain injury and the brain's response to it in recovery, hence its central position in any model of stroke.

The extrapolation by the authors is very far-reaching. The text barely acknowledges the very considerable practical barriers to obtaining complex data in an emergency setting, assumes that inherently different imaging modalities will be readily adapted to the model scheme, and dismisses the limited correspondence of clinically meaningful outcomes such as disability to brain anatomy. There is no clear practical suggestion of how complex modelling might be reconciled with clinical trials, or consider whether there are constraints on how it might be applied. Indeed the text continues to express the view that observational data can be considered superior to randomised trials.

We have addressed all these points at length in the preceding rebuttal: since the reviewer does not introduce any new criticisms or explain in what way our answers fall short, we

can only refer to our previous answers. To recap briefly, there is no major obstacle to feasibility—automated analysis of high-resolution stroke imaging is now not only possible, but supported by commercial products in increasingly widespread use—and our focus here is in any event on establishing empirical necessity, which is a preliminary to product development. The “limited correspondence of clinically meaningful outcomes such as disability to brain anatomy” is, as we have already explained, a self-fulfilling prophecy if both clinical outcomes and brain anatomy are reductively modelled, which is the current norm, and radically at odds with what we know about the organization of the brain from both correlative and disruptive functional anatomical studies.

We do not express the view that “observational data can be considered superior to randomised trials”. We explicitly quantify the relative importance of randomization vs model expressivity under data regimes that compel a compromise between them, and present the results of this analysis. We find that model expressivity has a greater impact on fidelity than confoundedness, favouring the regime that is best able to support it. This does not mean that observational data is superior to randomized trials simpliciter: the ideal inference will always maximise both data scale and unconfoundness. The results are not a view or an opinion: they are the outcome of the analysis.

Observational data has the undeniable benefit of being able to capture larger cohort sizes of a population, including time-sensitive acute disease data, of scales that RCTs may fail to attain (at least without vast time and resources), hence we seek to investigate whether observational data can be used to improve the healthcare decisions; while acknowledging the gold-standard quality of randomization in accounting for unobserved confounders (lines 69-72: “Randomization remains a powerful strategy, the best available means of protection from hidden confounding. But in the setting of unknown heterogeneity defined by many interacting factors, the necessary data scale becomes infeasibly large, forcing reliance on routine clinical data streams with limited control over randomization.”).

These are very big, generalised claims from a computer modelling exercise. Some recognition of the limitations and much more cautious conclusions would, in my view, remain appropriate.

This is not a “computer modelling exercise” but a comprehensive analysis of a large, diverse, highly inclusive set of data derived from real patients, whose treatments and therefore outcomes are shown to be potentially far more individually optimizable than the current approach to treatment identification and selection in stroke allows. Our claims are confined to the conclusions the data and its analysis compel. We have expanded the limitations section (lines 356-395) and clarified the exposition throughout to ensure correct interpretation.